# TWLA: Achieving Ternary Weights and Low-Bit Activations for LLMs via Post-Training Quantization

Zhixiong Zhao[1][*][†]   Zukang Xu[1][*]   Zhixuan Chen[1]   Xing Hu[1]   Zhe Jiang[2]   Dawei Yang[1][‡]

## Abstract

Large language models (LLMs) exhibit exceptional general language processing capabilities, but their memory and compute costs hinder deployment. Ternarization has emerged as a promising compression technique, offering significant reductions in model size and inference complexity. However, existing methods struggle with heavy-tailed activation distributions and therefore keep activations in high precision, fundamentally limiting end-to-end inference acceleration. To overcome this limitation, we propose **TWLA**, a post-training quantization (PTQ) framework that achieves 1.58-bit weight compression and 4-bit activation quantization while maintaining high accuracy. TWLA comprises three components: (1) Euclidean-to-Manifold Asymmetric Ternary Quantizer (E2M-ATQ) minimizes layer-output error under weight ternarization via a two-stage optimization from Euclidean initialization to manifold relocation; (2) Kronecker Orthogonal Tri-Modal Shaping (KOTMS) applies a Kronecker-structured orthogonal rotation to reshape weights into ternary-friendly tri-modal distributions, while the shared rotation statistically suppresses activation outliers; and (3) Inter-Layer Aware Activation Mixed Precision (ILA-AMP) explicitly introduces adjacent-layer second-order interaction costs in bit allocation and jointly optimizes for the layer-wise disparity of activation quantization gains induced by the shared orthogonal transform, preventing cascades triggered by a few weak layers. Extensive experiments demonstrate that TWLA maintains high accuracy under **W1.58A4**, while delivering significant inference acceleration. The code is available at TWLA.

[*]Equal contribution [†]This work was conducted during his internship at Houmo AI. [‡]Corresponding author. [1]Houmo AI, China [2]Southeast University, Nanjing, China. Correspondence to: Dawei Yang <dawei.yang@houmo.ai>.

*Proceedings of the 43rd International Conference on Machine Learning*, Seoul, South Korea. PMLR 306, 2026. Copyright 2026 by the author(s).

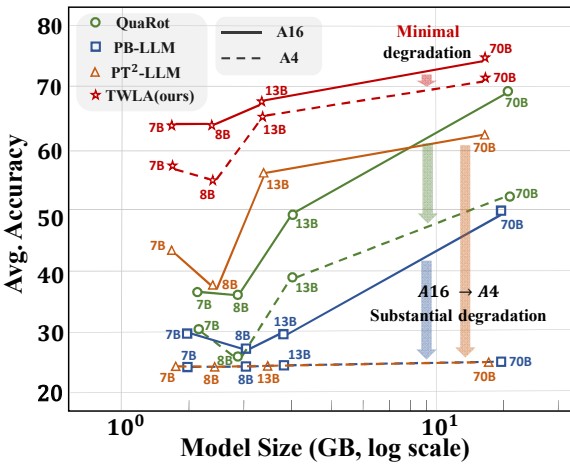

*Figure 1.* LLaMA-family performance on seven zero-shot tasks. TWLA remains robust under both weight-only and weight–activation quantization (at equal memory cost), while other methods degrade substantially with 4-bit activation quantization.

## 1. Introduction

In recent years, Large Language Models (LLMs) have achieved outstanding results across a wide range of diverse domains (Zhu et al., 2025). However, this strong capability largely comes from scaling up model size. Many state-of-the-art models contain billions, or even hundreds of billions, of parameters. This scale creates heavy demands on memory and computation during inference. For example, DeepSeek-R1-671B (Guo et al., 2025) has hundreds of billions of parameters. With FP16 inference, storing the weights alone can require no less than 1 TB of memory. Such a high resource requirement makes deployment on edge devices and other resource-limited platforms difficult (Zhao et al., 2025). Therefore, reducing inference cost while preserving model quality has become a central challenge for practical and sustainable LLM deployment (Li et al., 2024a; 2025).

Quantization has become a core technique for model compression (Zhao et al., 2026a; Xu et al., 2026; Gul et al., 2026) and ternarization is a representative option. It constrains weights to $\{-1, 0, +1\}$, which enables high compression and reduces compute complexity. Similar to binarization (Zhao et al., 2026b; Liu et al., 2025), ternarization can replace most floating-point multiplications with

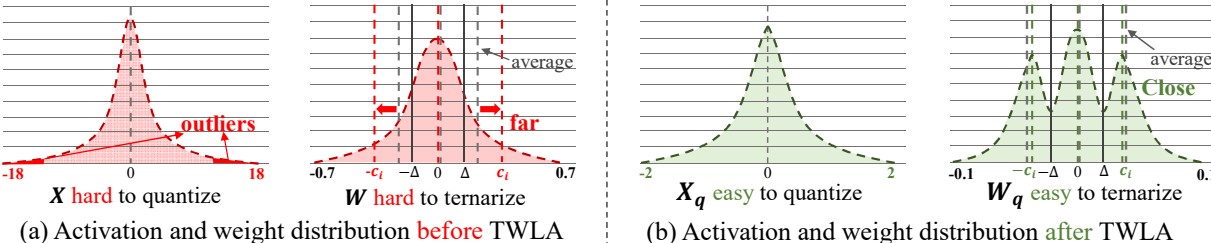

(a) Activation and weight distribution before TWLA     (b) Activation and weight distribution after TWLA

*Figure 2.* (a) Before applying TWLA, activation outliers hinder low-bit quantization and unimodal weights are misaligned with ternary codebooks. (b) After applying TWLA, activations are smoothed and weights are transformed into symmetric tri-modal forms.

additions and simple branching to lower inference costs, while offering stronger representational power and a superior accuracy-efficiency trade-off. Recent studies, such as TernaryLLM (Chen et al., 2024) and PT$^2$-LLM (Yan et al., 2025), mitigate the accuracy loss through distillation or iterative optimization. However, existing ternarization methods primarily focus on weight-only schemes and lack systematic modeling of activation quantization, leading to severe accuracy degradation at low bit-widths (see Fig. 1). To maintain performance, they typically retain activations in full precision and dequantize ternary weights during inference, which fundamentally limits end-to-end acceleration. BitNet v2 (Wang et al., 2025) demonstrates joint deployment of ternary weights with 4-bit activations, but it relies on expensive quantization-aware training (QAT), which requires a reported training cost exceeding $10^4$ GPU hours. These limitations highlight an urgent need for a practical post-training quantization (PTQ) approach. It should jointly support ternary weights and low-bit activations, and deliver efficient end-to-end inference.

We revisit the statistical properties of weights and activations in LLMs, as shown in Fig. 2(a). Empirically, per-channel weight distributions are often close to a unimodal Gaussian. This shape mismatches the ternary codebook $\{-1, 0, +1\}$, which leads to large quantization error when weights are projected into the ternary space. In contrast, a tri-modal distribution is more aligned with ternary representation, and it should reduce approximation error in principle. Meanwhile, activations exhibit heavy tails and extreme outliers. Under low-bit quantization, this heavy-tailed behavior often dominates the distortion. This contrast motivates our central question. *Can we, within a PTQ framework, ❶ reshape weight distributions toward a ternary-friendly tri-modal form, and ❷ suppress activation outliers to weaken heavy tails, so that ternary weights and low-bit activations can work well together?*

To this end, we propose **TWLA** (**T**ernarized **W**eights and **L**ow-bit **A**ctivations), a PTQ framework for efficient inference, which consists of three tightly coupled modules. First, we propose **E2M-ATQ**, a Euclidean-to-Manifold asymmetric ternary quantizer, to improve layer-output calibratability

for weight ternarization. Second, we introduce **KOTMS**, a Kronecker-structured orthogonal rotation optimized with a Cayley parameterization, which reshapes weights toward a ternary-friendly tri-modal distribution (see Fig. 2(b)); by orthogonal equivalence, the same rotation enables shared orthogonal mixing of activations, statistically shrinking outliers and stabilizing the dynamic range for low-bit activation quantization. Notably, since the activation benefits arise from a shared rotation, the quantizability gains are inherently heterogeneous across layers; consequently, layers with smaller gains can still become bottlenecks for low-bit activation quantization. To mitigate this effect, we further propose **ILA-AMP**, an inter-layer aware activation mixed-precision strategy. It introduces a second-order interaction cost between adjacent layers for the first time and unifies per-layer sensitivity with inter-layer coupling induced by distribution shift into a single quadratic surrogate objective. This formulation explicitly incorporates the layer-wise disparity of activation gains caused by the shared orthogonal matrix into the optimization, preventing cascades initiated by a few weak layers. Through this coordinated design, TWLA enables stable and efficient deployment of ternary weights with low-bit activations for end-to-end inference.

To summarize, our main contributions are:

- We first identify the key bottlenecks of LLMs under W1.58A4: limited ternary-quantizer calibratability and a ternary codebook–distribution mismatch for weights, and heavy-tailed distributions for activations.

- We propose TWLA, a retraining-free PTQ framework that achieves 1.58-bit weights together with low-bit activation quantization (e.g., 4-bit).

- Extensive experiments show that TWLA outperforms SOTA 2-bit and sub-2-bit methods, advancing LLMs to the W1.58A4 regime for PTQ.

## 2. Related Work

### 2.1. Ternarization for Large Language Models.

Ternarization constrains parameters to $\{-1, 0, +1\}$, reducing storage and simplifying arithmetic for efficient deployment. Early work such as TWN (Li et al., 2022) and TTQ (Zhu et al., 2017) established scale-aware ternary quantization and learned scaling factors, and later studies extended ternarization to activations for fully ternary networks (Wang et al., 2018; Alemdar et al., 2017). More recent efforts scale ternarization to Transformer and LLM settings, often with distillation or end-to-end training, e.g., TernaryBERT (Zhang et al., 2020), BitNet v2 (Wang et al., 2025) and RobuQ (Yang et al., 2025c). However, many of these methods rely on the training from scratch or QAT, which is costly and less transferable to arbitrary pretrained models. To reduce overhead, PTQ-based ternarization has been explored, such as PT²-LLM (Yan et al., 2025). However, existing PTQ approaches are largely weight-only, typically keeping activations in full precision and dequantizing weights during inference. As a result, jointly suppressing activation outliers and enabling low-bit activations together with ternary weights under PTQ remains an open challenge.

### 2.2. Mixed-Precision Quantization.

Mixed-Precision Quantization (MPQ) assigns different bit-widths to weights and/or activations to improve the accuracy–efficiency trade-off under a fixed budget. Although extensively explored in vision, MPQ for LLMs presents unique challenges. Recent works like APTQ (Guan et al., 2024) extend MPQ to Transformer blocks via Hessian-trace criteria. Under extreme low-bit regimes (e.g., 2-bit), coarse layer/block allocation often becomes insufficient, motivating finer-grained mixed precision and stronger outlier handling. Representative weight-focused methods include SpQR (Dettmers et al., 2023), PB-LLM (Shang et al., 2023), and LLM-MQ (Li et al., 2023), as well as TreeQ (Yang et al., 2025b) and SliM-LLM (Huang et al., 2024). Beyond weight-only designs, ResQ (Saxena et al., 2024) allocates higher precision to high-variance activation subspaces and applies rotations to suppress activation outliers. Despite these advances, most MPQ methods still assume independent layer-wise sensitivity, while activation quantization can shift distributions and couple errors across layers. Modeling such cross-layer interactions remains an important open problem for MPQ in LLMs.

## 3. Method

**Overview.** Fig. 3 illustrates the overall workflow of TWLA. We first review the standard ternarization formulation and basic notations in Sec. 3.1. Building on this foundation, Sec. 3.2 introduces E2M-ATQ, a training-free two-

stage procedure to minimize layer output error under weight ternarization. Sec. 3.3 then presents KOTMS, which applies a lightweight Kronecker-structured orthogonal rotation to reshape weights into ternary-friendly tri-modal distributions while suppressing activation outliers through shared rotation. Finally, Sec. 3.4 proposes ILA-AMP, which models second-order interaction costs between adjacent layers to allocate activation bits across layers, thereby preventing accuracy cascades caused by a few weak layers. The complete pseudocode is provided in Appendix A, and weights/activations distribution visualizations are available in Appendix D.

### 3.1. Preliminaries

**Weight Ternarization.** Ternarization converts floating-point weights in a model into the ternary set $\{-1, 0, +1\}$. Specifically, given a full-precision weight matrix $\mathbf{W} \in \mathbb{R}^{n \times m}$, TWN (Li et al., 2022) employs a row-wise threshold $\boldsymbol{\Delta} \in \mathbb{R}^{n \times 1}$ to map each element $W_{ij}$ to a ternary codebook $\mathbf{T} \in \{-1, 0, +1\}^{n \times m}$ according to

$$
T_{ij} = \begin{cases} +1, & W_{ij} > \Delta_i, \\ 0, & |W_{ij}| \le \Delta_i, \\ -1, & W_{ij} < -\Delta_i, \end{cases} \quad \Delta_i \approx \frac{0.75}{m} \sum_{j=1}^{m} |W_{ij}|. \tag{1}
$$

To improve reconstruction accuracy, TWN introduces a row-wise scaling factor $\boldsymbol{\alpha} \in \mathbb{R}^{n \times 1}$ to recover the magnitude of the original weights after the ternary matrix $\mathbf{T}$ is determined. For the $i$-th row, the optimal scaling factor $\alpha_i$ is obtained by minimizing the least-squares reconstruction error:

$$
\alpha_i = \arg\min_{\alpha} \|\mathbf{W}_{i,:} - \alpha \, \mathbf{T}_{i,:}\|_2^2 = \frac{\sum_{j=1}^{m} T_{ij} W_{ij}}{\sum_{j=1}^{m} |T_{ij}|}. \tag{2}
$$

Accordingly, the quantized weight matrix can be expressed as $\hat{\mathbf{W}} = \boldsymbol{\alpha}\mathbf{T}$. This ternarization scheme provides a practical solution for initializing ternary parameters in PTQ settings.

### 3.2. Euclidean-to-Manifold Asymmetric Ternary Quantizer

Pretrained LLM weights are often biased with non-zero row-wise means, which violates the common symmetry assumption in low-bit quantization. To capture such bias, we adopt an asymmetric ternary parameterization (e.g., (Yan et al., 2025)) that represents a full-precision weight matrix $\mathbf{W} \in \mathbb{R}^{n \times m}$ as quantized weight matrix $\bar{\mathbf{W}} \in \mathbb{R}^{n \times m}$:

$$
\bar{\mathbf{W}} = \boldsymbol{\mu}\mathbf{1}^{\top} + \operatorname{diag}(\boldsymbol{\alpha})\,\mathbf{T}, \qquad \mathbf{T} \in \{-1, 0, 1\}^{n \times m}, \tag{3}
$$

where $\boldsymbol{\mu}, \boldsymbol{\alpha} \in \mathbb{R}^n$ denote the row-wise shift and scale, and $\mathbf{1} \in \mathbb{R}^m$ is the all-one vector. We initialize $\boldsymbol{\mu}$ by the row mean $\boldsymbol{\mu} = \frac{1}{m}\sum_{j=1}^{m} \mathbf{W}_{:j}$. The discrete ternary codebook induces a stratified feasible set: with $\mathbf{T}$ fixed, the quantized

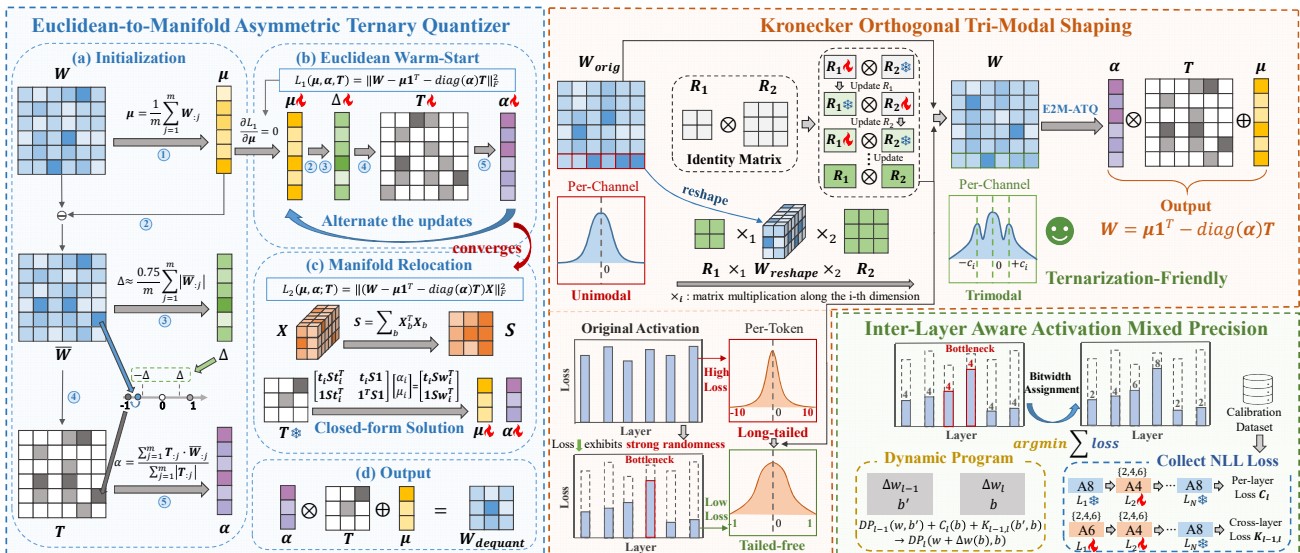

*Figure 3.* Overview of TWLA (initialization follows the same design as PT²-LLM (Yan et al., 2025)). Euclidean-to-Manifold Asymmetric Ternary Quantizer (E2M-ATQ): minimizes layer-output error under weight ternarization via a two-stage optimization. Kronecker Orthogonal Tri-Modal Shaping (KOTMS): reshapes weight into ternary-friendly tri-modal distribution. Inter-Layer Aware Activation Mixed Precision (ILA-AMP): prevents cascades triggered by weak layers.

weights vary only within the corresponding affine stratum and optimizing the continuous parameters $(\boldsymbol{\mu}, \boldsymbol{\alpha})$ often admits closed-form updates, whereas allowing $\mathbf{T}$ to change entails discrete search and combinatorial complexity. However, under output-alignment objectives, calibration-induced inter-column correlations globally couple $\mathbf{T}$, turning an otherwise element-wise thresholding step into a difficult combinatorial optimization. We therefore adopt a two-stage pipeline: we first obtain a stable ternary pattern and a reliable initialization in the Euclidean weight domain, and then freeze $\mathbf{T}$ as a structural prior and relocate $(\boldsymbol{\mu}, \boldsymbol{\alpha})$ on the metric manifold defined by the calibration second moment via a metric-consistent closed-form update, thereby aligning the layer output within the selected stratum.

**Euclidean warm-start under Frobenius geometry.** In the first stage, we obtain a stable ternary pattern by minimizing the weight-domain reconstruction error

$$L_1(\boldsymbol{\mu}, \boldsymbol{\alpha}, \mathbf{T}) = \left\| \mathbf{W} - \boldsymbol{\mu}\mathbf{1}^\top - \mathrm{diag}(\boldsymbol{\alpha})\,\mathbf{T} \right\|_F^2. \quad (4)$$

Let $\mathbf{E} = \mathbf{W} - \bar{\mathbf{W}}$ denote the residual error, where the mean of $\mathbf{E}$ is not always zero due to inevitable errors during the ternarization process. Following (Li et al., 2024b), we apply residual-mean correction:

$$\boldsymbol{\mu} \leftarrow \boldsymbol{\mu} + \frac{1}{m}\mathbf{E}\mathbf{1}. \quad (5)$$

After correcting $\boldsymbol{\mu}$, we sequentially update $\boldsymbol{\alpha}$ and $\mathbf{T}$ according to Eq. 2 and Eq. 1. This update scheme naturally extends to an iterative algorithm. Specifically, at each iteration we

update $\boldsymbol{\mu}$, then $\boldsymbol{\alpha}$, and finally $\mathbf{T}$, so that each variable is optimal under the objective in Eq. 4 given the others. (Appendix B.1 proves the objective decreases monotonically throughout the iterations). Upon convergence, we obtain a stable ternary pattern $\mathbf{T}^{(0)}$ and a Euclidean initialization $(\boldsymbol{\mu}^{(0)}, \boldsymbol{\alpha}^{(0)})$ for the subsequent manifold relocation stage.

**Manifold relocation under calibration-induced metric.** The Euclidean reconstruction error in the weight domain may not faithfully reflect forward output deviation. In this stage, we relocate the continuous parameters $(\boldsymbol{\mu}, \boldsymbol{\alpha})$ on the metric manifold induced by calibration activations, while freezing the discrete stratum as $\mathbf{T} = \mathbf{T}^{(0)}$. Given calibration activations $\mathbf{X}$, we minimize the layer output error

$$L_2(\boldsymbol{\mu}, \boldsymbol{\alpha}; \mathbf{T}) = \left\| \left( \mathbf{W} - \boldsymbol{\mu}\mathbf{1}^\top - \mathrm{diag}(\boldsymbol{\alpha})\,\mathbf{T} \right) \mathbf{X} \right\|_F^2. \quad (6)$$

To avoid repeatedly multiplying high-dimensional $\mathbf{X}$, we precompute the activation second moment (Li et al., 2024b) $\mathbf{S} = \sum_b \mathbf{X}_b^\top \mathbf{X}_b \in \mathbb{R}^{m \times m}$, which allows us to rewrite the $L_2$ reconstruction error $L_2 = \|\mathbf{W}\mathbf{X} - \mathbf{W}_c\mathbf{X}\|_F^2$ as the following equivalent quadratic form:

$$L_2(\boldsymbol{\mu}, \boldsymbol{\alpha}; \mathbf{T}) = \mathrm{Tr}\left( \mathbf{E}\mathbf{S}\mathbf{E}^\top \right), \mathbf{E} = \mathbf{W} - \boldsymbol{\mu}\mathbf{1}^\top - \mathrm{diag}(\boldsymbol{\alpha})\,\mathbf{T}. \quad (7)$$

Equivalently, minimizing Eq. 7 amounts to projecting $\mathbf{W}$ onto the affine constraint set induced by a fixed $\mathbf{T}$. Since $\mathbf{S}$ is typically dense, optimizing over $\mathbf{T}$ would introduce global coupling and yield a challenging combinatorial problem; we thus fix $\mathbf{T} = \mathbf{T}^{(0)}$ and optimize only $(\boldsymbol{\mu}, \boldsymbol{\alpha})$ for relocation. Under this constraint, the objective decouples across rows:

for each row $i$ with $(\mathbf{w}_i, \mathbf{t}_i)$, imposing $\partial L_2/\partial \mu_i = 0$ and $\partial L_2/\partial \alpha_i = 0$ gives a $2 \times 2$ linear system.

$$\begin{bmatrix} \mathbf{t}_i \mathbf{S} \mathbf{t}_i^\top & \mathbf{t}_i \mathbf{S} \mathbf{1} \\ \mathbf{1}^\top \mathbf{S} \mathbf{t}_i^\top & \mathbf{1}^\top \mathbf{S} \mathbf{1} \end{bmatrix} \begin{bmatrix} \alpha_i \\ \mu_i \end{bmatrix} = \begin{bmatrix} \mathbf{t}_i \mathbf{S} \mathbf{w}_i^\top \\ \mathbf{1}^\top \mathbf{S} \mathbf{w}_i^\top \end{bmatrix}. \qquad (8)$$

The unique closed-form solution is

$$\alpha_i^* = \frac{(\mathbf{t}_i \mathbf{S} \mathbf{w}_i^\top)(\mathbf{1}^\top \mathbf{S} \mathbf{1}) - (\mathbf{t}_i \mathbf{S} \mathbf{1})(\mathbf{1}^\top \mathbf{S} \mathbf{w}_i^\top)}{(\mathbf{t}_i \mathbf{S} \mathbf{t}_i^\top)(\mathbf{1}^\top \mathbf{S} \mathbf{1}) - (\mathbf{t}_i \mathbf{S} \mathbf{1})(\mathbf{1}^\top \mathbf{S} \mathbf{t}_i^\top)},$$
$$\mu_i^* = \frac{(\mathbf{t}_i \mathbf{S} \mathbf{t}_i^\top)(\mathbf{1}^\top \mathbf{S} \mathbf{w}_i^\top) - (\mathbf{1}^\top \mathbf{S} \mathbf{t}_i^\top)(\mathbf{t}_i \mathbf{S} \mathbf{w}_i^\top)}{(\mathbf{t}_i \mathbf{S} \mathbf{t}_i^\top)(\mathbf{1}^\top \mathbf{S} \mathbf{1}) - (\mathbf{t}_i \mathbf{S} \mathbf{1})(\mathbf{1}^\top \mathbf{S} \mathbf{t}_i^\top)}. \qquad (9)$$

The full derivation is provided in Appendix B.2. After the two-stage Euclidean-to-Manifold (E2M) procedure, the resulting quantized weights under E2M-ATQ are

$$\bar{\mathbf{W}} = \boldsymbol{\mu}^* \mathbf{1}^\top + \mathrm{diag}(\boldsymbol{\alpha}^*)\, \mathbf{T}^{(0)}. \qquad (10)$$

### 3.3. Kronecker Orthogonal Tri-Modal Shaping

In the previous section, E2M-ATQ improves the continuous shift and scale parameters for a fixed ternary pattern. However, such parameter relocation does not change the geometric mismatch between pretrained weights and the ternary codebook. The asymmetric ternary levels $\{-\alpha_i + \mu_i,\ \mu_i,\ +\alpha_i + \mu_i\}$ are most effective when each channel contains three well-separated attraction regions, while pretrained LLM weights are usually concentrated around a near-unimodal shape (Ye et al., 2025). Consequently, many entries remain close to ternary decision boundaries, making the final hard projection sensitive to small perturbations and increasing the ternarization error.

To reduce this mismatch before hard ternary projection, KOTMS introduces a structured orthogonal coordinate transformation and optimizes it with a ternary-codebook shaping loss. The orthogonal transformation preserves the full-precision mapping through its inverse rotation, while the shaping loss encourages the transformed weights to become more compatible with the asymmetric ternary levels used by E2M-ATQ. In this way, KOTMS complements E2M-ATQ: E2M-ATQ calibrates the ternary parameters within a selected discrete pattern, whereas KOTMS improves the coordinate system in which the ternary projection is performed.

**Codebook-Aligned Shaping Objective.** We use a symmetric three-component Gaussian mixture as a smooth surrogate for ternary-codebook alignment. The three modes serve as adaptive row-wise anchors corresponding to the negative, zero, and positive ternary regions. For the $i$-th row $\mathbf{w}_i$, let $\mathbf{R}$ denote a learnable orthogonal transform and define

$$\mathbf{z}_i = \mathbf{w}_i \mathbf{R}, \qquad z_{ij} = (\mathbf{z}_i)_j. \qquad (11)$$

We assign three anchors $\{-c_i, 0, +c_i\}$ to each row and minimize the following negative log-likelihood:

$$\mathcal{L}_{\mathrm{TriGMM}} = -\frac{1}{nm} \sum_{i=1}^n \sum_{j=1}^m \log \Big[ \pi_+ \phi(z_{ij}; +c_i, \sigma_i^2)$$
$$+ \pi_0 \phi(z_{ij}; 0, \sigma_i^2) + \pi_- \phi(z_{ij}; -c_i, \sigma_i^2) \Big], \qquad (12)$$

where $\phi(\cdot; c, \sigma^2)$ denotes the Gaussian density, $c_i = \frac{1}{m} \sum_{j=1}^m |z_{ij}|$, $\sigma_i = \mathrm{std}(\{z_{ij}\}_{j=1}^m)$, and $\pi_+ = \pi_- = \frac{1-\pi_0}{2}$ with $\pi_0 \in (0, 1)$. This objective gives higher likelihood to values close to the ternary-aligned anchors and lower likelihood to values far from them. Therefore, minimizing $\mathcal{L}_{\mathrm{TriGMM}}$ provides a differentiable approximation to moving transformed entries toward $\{-c_i, 0, +c_i\}$, which is consistent with the subsequent hard ternary projection $T_{ij} \leftarrow \Pi_{\{-1,0,1\}}(\cdot)$.

To avoid degenerate solutions, such as $c_i \to 0$ or excessive concentration around the zero component, we further regularize the zero-mode responsibility:

$$\mathcal{L}_{\mathrm{shape}} = \mathcal{L}_{\mathrm{TriGMM}} + \beta \mathcal{L}_{\mathrm{zero}}, \qquad (13)$$

where $\mathcal{L}_{\mathrm{zero}}$ controls the mass assigned to the central mode. Details of this regularizer and the soft-projection interpretation are provided in Appendix B.5 and Appendix B.6. The resulting objective supplies the optimization signal for learning the orthogonal transform used in KOTMS.

**Kronecker-Structured Orthogonal Transform.** A dense orthogonal matrix $\mathbf{R} \in \mathbb{R}^{m \times m}$ is impractical for LLM layers because it requires large storage and expensive matrix multiplication. KOTMS therefore restricts the transform to a Kronecker-structured orthogonal family (Gu et al., 2026; Xiao et al., 2025):

$$\mathbf{R} = \mathbf{R}_1 \otimes \mathbf{R}_2, \quad \mathbf{R}_1 \in \mathcal{O}(n_1),\ \mathbf{R}_2 \in \mathcal{O}(n_2),\ n_1 n_2 = m. \qquad (14)$$

This parameterization keeps the transformation exactly invertible: $\mathbf{R}^{-1} = \mathbf{R}^\top = \mathbf{R}_1^\top \otimes \mathbf{R}_2^\top$. Thus, the inverse rotation can be consistently applied to the activation side, preserving the full-precision function while allowing the weight coordinates to be shaped for ternary projection.

For implementation, a row vector $\mathbf{v} \in \mathbb{R}^{1 \times m}$ is reshaped into $\mathbf{V}_{\mathrm{mat}} \in \mathbb{R}^{n_1 \times n_2}$. The product with the Kronecker transform can then be evaluated as

$$\mathbf{v} \mathbf{R} = \mathrm{vec}\big(\mathbf{R}_2^\top \mathbf{V}_{\mathrm{mat}} \mathbf{R}_1\big)^\top. \qquad (15)$$

As a result, KOTMS stores and applies two small orthogonal factors instead of a full dense rotation matrix. When $n_1$ and $n_2$ are chosen with comparable sizes, the application is dominated by two compact matrix multiplications. This

*Table 1.* Comparison of perplexity on WikiText2 and averaged accuracy on seven Zero-Shot tasks (Arc-Challenge, Arc-Easy , HellaSwag , LAMBADA-openai, LAMBADA-standard, PIQA, and WinoGrande). MP indicates that mixed-precision is adopted for weights and/or activations. Full results are in the Appendix C.1

| Method | #Bits (W) | #Bits (A) | LLaMA2-7B 0-shot[7] Avg.(↑) | Wiki (↓) | LLaMA2-13B 0-shot[7] Avg.(↑) | Wiki (↓) | LLaMA2-70B 0-shot[7] Avg.(↑) | Wiki (↓) | LLaMA3-8B 0-shot[7] Avg.(↑) | Wiki (↓) | Qwen3-8B 0-shot[7] Avg.(↑) | Wiki (↓) | Qwen3-14B 0-shot[7] Avg.(↑) | Wiki (↓) | Qwen3-32B 0-shot[7] Avg.(↑) | Wiki (↓) |
|---|---|---|---|---|---|---|---|---|---|---|---|---|---|---|---|---|
| FP16 | 16 | 16 | 69.49 | 5.47 | 72.19 | 4.88 | 76.71 | 3.32 | 72.51 | 6.14 | 69.09 | 9.00 | 72.47 | 8.64 | 72.42 | 7.61 |
| GPTQ | 2 | | 29.09 | 47.13 | 27.21 | 182.11 | 45.59 | 24.53 | 26.38 | 154.38 | 29.25 | 61.74 | 28.61 | 84.25 | 38.18 | 32.95 |
| QuaRot | 2 | | 36.30 | 19.97 | 49.30 | 10.36 | 68.79 | 5.56 | 35.66 | 23.05 | – | – | – | – | – | – |
| SliM-LLM | 2MP | 16 | 48.96 | 16.03 | 52.00 | 9.06 | – | – | 27.49 | 38.09 | 39.13 | 27.10 | 48.10 | 15.19 | 64.79 | 12.12 |
| PB-LLM | 1.7 | | 29.01 | 41.04 | 26.29 | 335.22 | 50.88 | 12.02 | 29.32 | 42.59 | 38.43 | 26.20 | 44.80 | 22.90 | 63.92 | 13.17 |
| PT²-LLM | 1.58 | | 42.82 | 11.56 | 56.54 | 9.19 | 62.47 | 6.27 | 39.04 | 32.19 | – | – | 46.35 | 16.48 | – | – |
| **TWLA** | **1.58** | | **62.91** | **6.97** | **67.70** | **5.79** | **73.60** | **4.13** | **62.98** | **9.39** | **62.05** | **12.52** | **68.48** | **10.42** | **69.54** | **8.94** |
| GPTQ | 2 | 6 | 27.94 | 120.21 | 25.90 | 2e3 | 35.89 | 287.83 | 25.69 | 386.19 | 26.65 | 125.31 | 26.76 | 200.91 | 31.15 | 125.47 |
| QuaRot | 2 | 6 | 35.22 | 21.25 | 47.95 | 10.62 | 67.97 | 5.63 | 35.51 | 24.37 | – | – | – | – | – | – |
| ResQ | 2.3MP | 6.1MP | 51.18 | 9.93 | 56.49 | 8.62 | 57.81 | 5.38 | 47.57 | 16.93 | – | – | – | – | – | – |
| SliM-LLM | 2MP | 6 | 37.06 | 565.93 | 43.32 | 11.32 | – | – | 27.15 | 233.82 | 31.41 | 67.61 | 37.46 | 28.91 | 52.43 | 20.88 |
| PB-LLM | 1.7 | 6 | 30.60 | 44.36 | 26.61 | 767.99 | 40.89 | 23.66 | 30.28 | 55.75 | 31.67 | 66.83 | 37.28 | 33.15 | 51.45 | 21.05 |
| PT²-LLM | 1.58 | 6 | 38.61 | 22.12 | 50.82 | 16.33 | 48.96 | 13.78 | 30.81 | 69.83 | – | – | 36.00 | 30.90 | – | – |
| **TWLA** | **1.58** | **6MP** | **61.98** | **7.14** | **66.88** | **5.90** | **73.45** | **4.21** | **61.44** | **9.76** | **59.03** | **12.83** | **67.29** | **10.69** | **68.13** | **9.00** |
| GPTQ | 2 | 4 | 25.80 | 2e4 | 25.35 | 7e3 | 25.92 | 2e4 | 25.64 | 2e4 | 25.29 | 1e4 | 25.79 | 3e4 | 24.85 | 2e7 |
| QuaRot | 2 | 4 | 30.83 | 37.79 | 39.42 | 14.49 | 53.64 | 6.77 | 27.69 | 119.90 | – | – | – | – | – | – |
| ResQ | 2.3MP | 4.2MP | 45.23 | 11.97 | 51.84 | 9.29 | 56.43 | 7.92 | 39.12 | 21.95 | – | – | – | – | – | – |
| SliM-LLM | 2MP | 4 | 25.67 | 4e3 | 26.44 | 1e3 | – | – | 25.60 | 1e4 | 19.18 | 2e4 | 25.62 | 1e4 | 26.31 | 6e3 |
| PB-LLM | 1.7 | 4 | 26.46 | 458.08 | 25.54 | 3e3 | 26.18 | 2e3 | 26.29 | 684.28 | 26.26 | 5e3 | 29.23 | 914.41 | 26.36 | 1e4 |
| PT²-LLM | 1.58 | 4 | 27.31 | 341.08 | 29.07 | 2e3 | 26.19 | 1e3 | 25.81 | 720.21 | – | – | 25.53 | 5e3 | – | – |
| **TWLA** | **1.58** | **4MP** | **58.00** | **8.31** | **64.30** | **6.68** | **71.10** | **4.77** | **55.23** | **12.83** | **50.42** | **16.15** | **62.00** | **12.84** | **65.25** | **9.71** |

makes the transform suitable for PTQ calibration and for deployment-time activation rotation with limited overhead. In addition, the shared orthogonal mixing disperses concentrated activation directions and helps reduce heavy-tailed activation outliers, which facilitates low-bit activation quantization; empirical evidence is provided in Appendix B.4.

**Orthogonality-Preserving Optimization.** We optimize the Kronecker factors under the shaping objective in Eq. equation 12. To maintain strict orthogonality during optimization, each factor is parameterized by a Cayley transform. Specifically, for $k \in \{1, 2\}$, we introduce a free matrix $\mathbf{S}_k \in \mathbb{R}^{n_k \times n_k}$ and construct a skew-symmetric generator

$$\mathbf{A}_k = \mathbf{S}_k - \mathbf{S}_k^\top. \quad (16)$$

The corresponding orthogonal factor is defined as

$$\mathbf{R}_k = \text{cayley}(\mathbf{A}_k) = (\mathbf{I}+\mathbf{A}_k)^{-1}(\mathbf{I}-\mathbf{A}_k), \qquad k \in \{1, 2\}. \quad (17)$$

This guarantees $\mathbf{R}_k^\top \mathbf{R}_k = \mathbf{I}$ throughout optimization and avoids explicit projection steps. During calibration, gradients of $\mathcal{L}_{\text{shape}}$ update the free matrices $\mathbf{S}_1$ and $\mathbf{S}_2$, and the resulting orthogonal factors define the final KOTMS rotation.

### 3.4. Inter-Layer Aware Activation Mixed Precision

**Discussion.** Although KOTMS can statistically mitigate activation outliers, it is important to note that its learning objective is defined purely in the weight domain. Consequently, the improvement in activation quantizability is largely an indirect by-product of applying the same orthogonal mixing, and thus can be highly uneven across layers (See Appendix D.2 for a detailed distributional visualization.). As a result, a small number of less-benefited layers may become the primary bottleneck for low-bit activation quantization. Motivated by this observation, we allocate activation bitwidths across layers under a global budget constraint.

Consider a network with $L$ layers. Let $b_\ell$ denote the activation bitwidth assigned to layer $\ell$, with candidates $\mathcal{B} = \{2, 4, 6, 8\}$, and let $\mathbf{b} = (b_1, \ldots, b_L)$ denote a layer-wise configuration. We adopt the per-token negative log-likelihood (NLL) on a validation set as the value function:

$$v_{\text{NLL}}(\mathbf{b}) = \mathbb{E}_{(\mathbf{x},t)\sim\mathcal{D}}\Big[ -\log p_{\boldsymbol{\theta}}(x_{t+1} \mid \mathbf{x}_{\leq t}; \mathbf{b}) \Big]. \quad (18)$$

Here, $\mathcal{D}$ is the validation set and $p_{\boldsymbol{\theta}}(\cdot; \mathbf{b})$ is the activation-quantized model under assignment $\mathbf{b}$. Most MPQ methods assume layer-wise independent sensitivity, i.e., quantization effects are approximately additive across layers. This is often invalid for LLMs: quantizing layer $\ell$ changes its output distribution, shifts the input statistics of layer $\ell!+!1$, and

perturbs the downstream quantizer. These shifts accumulate and amplify, inducing cross-layer coupling that is especially severe at 2–4 bits and can trigger accuracy collapse. We therefore add explicit adjacent-layer interaction terms to model local error propagation along the stack.

Directly minimizing $v_{\mathrm{NLL}}(\mathbf{b})$ over $\mathcal{B}^L$ is intractable. We instead build a DP-friendly second-order surrogate that approximates the NLL change via (i) per-layer costs $C_\ell(b)$ and (ii) adjacent interaction costs $K_{\ell-1,\ell}(b', b)$. We use $b_{\max} = 8$ as a reference: start from uniform 8-bit quantization and then lower selected layers around this baseline, yielding controlled perturbations and lower-variance sensitivity estimates. The first-order cost is defined as

$$C_\ell(b) = v_{\mathrm{NLL}}(b_\ell = b,\ b_{k\neq\ell} = b_{\max}) - v_{\mathrm{NLL}}(\mathbf{b}_{\max}), \tag{19}$$

and the adjacent interaction cost as

$$K_{\ell-1,\ell}(b', b) = v_{\mathrm{NLL}}\Big(b_{\ell-1} = b',\ b_\ell = b,\ b_{k\notin\{\ell-1,\ell\}} = b_{\max}\Big)$$
$$-v_{\mathrm{NLL}}(\mathbf{b}_{\max}) - C_{\ell-1}(b') - C_\ell(b), \tag{20}$$

When $K_{\ell-1,\ell}(b', b) > 0$, it indicates coupling amplification caused by error propagation between adjacent layers. We provide further justification for modeling interactions only between adjacent layers in Appendix B.7.

Then we formulate mixed-precision activation allocation as a budget-constrained minimization problem with adjacent second-order terms:

$$\min_{\{b_\ell\}} \sum_{\ell=1}^{L} C_\ell(b_\ell) + \sum_{\ell=2}^{L} K_{\ell-1,\ell}(b_{\ell-1}, b_\ell),\ \text{s.t.} \sum_{\ell=1}^{L} b_\ell \leq B, \tag{21}$$

where $B$ is the global bit budget. Since $K_{\ell-1,\ell}$ is defined only on adjacent layer pairs, the surrogate objective has a chain structure. We therefore solve the global bit allocation exactly via dynamic programming under the budget constraint $c \leq B$, and recover the optimal assignment $\{b_\ell^*\}_{\ell=1}^{L}$ by backtracking. Full details are deferred to Appendix B.8.

## 4. Experiments

### 4.1. Experiment setup

**Models and Datasets.** We conduct experiments on the LLaMA families (Touvron et al., 2023) and the Qwen3 family (Yang et al., 2025a). In addition, we evaluate instruction-tuned variants to further demonstrate the effectiveness of our method. Beyond standard perplexity evaluation on Wiki-Text2 (Merity et al., 2016) and C4 (Raffel et al., 2023), we assess TWLA on a broad set of zero-shot tasks, including ARC-Challenge and ARC-Easy (Clark et al., 2018), HellaSwag (Zellers et al., 2019), LAMBADA-openai and LAMBADA-standard (Paperno et al., 2016), PIQA (Bisk et al., 2019), and WinoGrande (Sakaguchi et al., 2019).

*Table 2.* Results on Qwen3-32B-Instruct across three challenging benchmarks: MMLU, HumanEval, and GSM8K. Best and second-best results are marked in **bold** and underlined, respectively.

| Model | Method | #Bits(W) | #Bits(A) | MMLU | HumanEval | GSM8K |
|---|---|---|---|---|---|---|
| | FP16 | 16 | 16 | 80.75 | 50.41 | 66.55 |
| | GPTQ | 3 | | 73.97 | 41.00 | 53.33 |
| | GPTQ | 2 | | 27.10 | 0 | 1.23 |
| | SliM-LLM | 2MP | 16 | 55.38 | 23.74 | 35.21 |
| Qwen3-32B | PB-LLM | 1.7 | | 48.21 | 12.20 | 24.33 |
| -Instruct | **TWLA** | 1.58 | | **75.59** | **42.99** | **55.72** |
| | GPTQ | 3 | 4 | 23.28 | 0 | 1.02 |
| | GPTQ | 2 | 4 | 23.20 | 0 | 0.13 |
| | SliM-LLM | 2MP | 4 | 23.71 | 0 | 0.97 |
| | PB-LLM | 1.7 | 4 | 23.12 | 0 | 1.33 |
| | **TWLA** | 1.58 | 4MP | **70.21** | **37.58** | **48.67** |

We further evaluate TWLA on more challenging reasoning benchmarks, including the multi-domain knowledge task MMLU (Hendrycks et al., 2021), the mathematical reasoning benchmark GSM8K (Cobbe et al., 2021), and the code generation benchmark HumanEval (Chen et al., 2021).

**Baseline Methods.** We compare TWLA with diverse PTQ baselines in the 2-bit and sub-2-bit regimes, covering both weight-only and weight–activation quantization. For weight-only PTQ, we include SliM-LLM (Huang et al., 2024) as a strong mixed-precision baseline, PB-LLM (Shang et al., 2023) for sub-2-bit quantization (with an average weight bit-width close to ours), and PT$^2$-LLM (Yan et al., 2025) as a representative ternarization method with the same average weight bit-width as TWLA. For weight–activation PTQ, we benchmark ResQ (Saxena et al., 2024), which applies mixed-precision activation quantization with an average of about 4.25 bits. We further report GPTQ (Frantar et al., 2023) and QuaRot (Ashkboos et al., 2024) as widely adopted PTQ baselines for completeness.

**Implementation Details.** All experiments are conducted on NVIDIA A6000 GPUs. For E2M-ATQ, we perform 15 iterations to ensure the convergence of the ternarization parameters. Following BWLA (Zhao et al., 2026b), we select 128 calibration samples from WikiText2 (Merity et al., 2016), each with a sequence length of 2048. Based on this sample set, we optimize the parameters in KOTMS for 100 iterations with a fixed learning rate of 0.01, and the same sample set is also employed as the calibration data for the ILA-AMP.

### 4.2. Main Results

**Comparison Results.** We systematically evaluate ternarization across the LLaMA and Qwen3 families under different activation bit-widths, where all non-ternary PTQ baselines use 2-bit weights. As shown in Table 1, TWLA achieves the best performance consistently across model

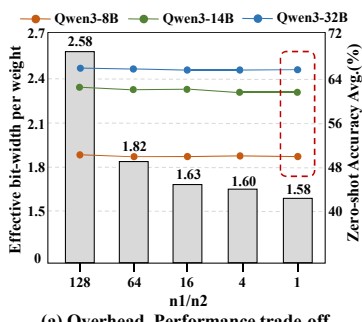
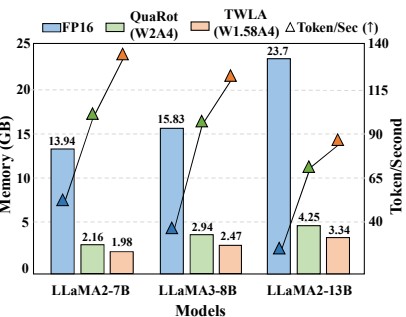
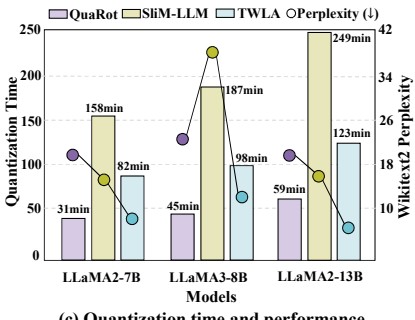

(a) Overhead–Performance trade-off     (b) Speed up and memory savings     (c) Quantization time and performance

*Figure 4.* (a) Ablation of the Overhead–Performance trade-off for KOTMS. (b) Comparison of throughput and memory consumption across FP16, QuaRot, and TWLA. (c) Comparison of quantization time and perplexity across QuaRot, SliM-LLM, and TWLA.

*Table 3.* Impact of different components in TWLA.

| #Bits (A) | E2M-ATQ | KOTMS | ILA-AMP | LLaMA2-13B C4 (↓) | LLaMA2-13B MMLU (↑) | Qwen3-14B C4 (↓) | Qwen3-14B MMLU (↑) |
|---|---|---|---|---|---|---|---|
| 16 | ✕ | ✕ | - | 6e3 | 23.01 | 1e4 | 23.22 |
|    | ✓ | ✕ | - | 18.32 | 30.15 | 23.64 | 57.64 |
|    | ✕ | ✓ | - | 57.98 | 24.12 | 102.12 | 25.02 |
|    | ✓ | ✓ | - | **8.64** | **44.86** | **17.26** | **69.15** |
| 4 | ✓ | ✕ | ✕ | 3e3 | 23.76 | 2e3 | 24.12 |
|   | ✓ | ✓ | ✕ | 25.03 | 27.52 | 31.78 | 47.60 |
|   | ✓ | ✕ | ✓ | 55.86 | 24.01 | 390.95 | 25.33 |
|   | ✓ | ✓ | ✓ | **10.07** | **38.17** | **21.00** | **60.82** |

scales. Under A16 setting, TWLA operates at the lowest average weight precision (1.58 bits) yet attains both the lowest WikiText2 perplexity and the highest average accuracy, clearly outperforming 2-bit baselines such as GPTQ, QuaRot, and SliM-LLM, and delivering substantial gains over the current SOTA ternarization method PT$^2$-LLM (e.g., on LLaMA-3-8B, average accuracy improves from 39.04 to 62.98 while WikiText2 PPL drops by 70%). Under the more challenging A4 setting, prior methods generally suffer perplexity explosion, whereas TWLA remains stable and further surpasses ResQ, which also employs mixed-precision activations; on LLaMA-2-70B, TWLA raises average accuracy from 56.43 to 71.10 (exceeding 92% of FP16) while reducing memory consumption by over 30% relative to ResQ. Full expanded results are provided in Appendix C.1.

**Experiments of Instruction-tuned Models.** Instruction tuning substantially improves the practical utility of LLMs, but it also makes PTQ considerably more challenging. We evaluate Qwen3-32B-Instruct on three challenging benchmarks (Table 2). With FP16 activations, TWLA retains roughly 85% of FP16 performance, recovering most of the reasoning capability of the model. Notably, TWLA even outperforms 3-bit GPTQ while using roughly half the memory, yielding a favorable accuracy–efficiency trade-off. When activations are further quantized to 4-bit, existing methods largely collapse: MMLU accuracy drops to near

random-guessing level (around 25%), while HumanEval and GSM8K fall to almost zero. In contrast, TWLA preserves about 90% of its FP16-activation performance, indicating strong robustness and high quantization quality in the low-bit activation regime.

### 4.3. Ablation Studies

TWLA consists of three modules (E2M-ATQ, KOTMS, and ILA-AMP) designed to reduce quantization error. Our ablations include (i) evaluating the individual contribution of each module and (ii) ablating the orthogonal matrix size in KOTMS to examine the overhead–performance trade-off. We provide calibration-data ablations in Appendix C.2 (seed, size, and type), and report additional ILA-AMP analyses on sensitivity metrics, layer-wise bit allocations, and interaction-order cost–benefit trade-offs in Appendix C.3.

**Modular Sensitivity Study.** We report C4 perplexity and MMLU accuracy to isolate the effects of each component (Table 3). Without activation quantization, either E2M-ATQ or KOTMS alone improves both metrics, with larger gains from E2M-ATQ; combining them performs best (e.g., Qwen3-14B gains ∼45% MMLU). Under 4-bit activations, using only E2M-ATQ+KOTMS degrades accuracy, whereas adding ILA-AMP recovers a further ∼12% MMLU on LLaMA2-13B and Qwen3-14B, bringing results close to the no-activation-quantization setting. These results highlight the strong synergy among E2M-ATQ, KOTMS, and ILA-AMP, and underscore the necessity of using them together.

**Overhead–Performance Trade-off.** We ablate the orthogonal matrix size in KOTMS to quantify the overhead–performance trade-off (Fig. 4(a)). By varying the Kronecker dimensions $n_1$ and $n_2$ to control the structure of the orthogonal transform, we find that making them more balanced (smaller $n_1/n_2$) sharply reduces KOTMS's effective weight-bit overhead with only minor accuracy loss. On the Qwen3 family, higher symmetry reduces the OKT overhead from roughly 1 bit to a negligible level (<0.01 bit), with the average accuracy dropping by < 2%. We therefore choose $n_1$

and $n_2$ to be as close as possible in all experiments.

### 4.4. Efficiency Analysis of TWLA

**Speedup and Memory Savings.** We evaluate the inference efficiency of TWLA on the LLaMA family with `llama.cpp`[1] on NVIDIA RTX A6000 GPUs, leveraging its native ternary operators for optimized low-bit kernels. We compare FP16, QuaRot (W2A4), and TWLA (W1.58A4) in throughput and memory. Under batch size 4 with 128-token prefill and 256-token decoding, Fig. 3(b) shows that ternary weights further reduce memory and improve speed beyond 2-bit weights. On LLaMA2-13B, QuaRot improves throughput from 23.70 tokens/s (FP16) to 66.42 tokens/s, while TWLA further increases it to 86.35 tokens/s, yielding a 3.64× speedup over FP16 and a 1.3× speedup over QuaRot. Meanwhile, the parameter memory is reduced from 23.7GB to 3.34GB by packing five ternary values into one 8-bit integer (effective ∼1.6 bits/weight), saving > 80% of the storage. These results highlight the substantial efficiency gains brought by TWLA.

**Quantization Time Comparison.** As shown in Fig. 3(c), TWLA exhibits a clear performance–efficiency trade-off. Compressing LLaMA2-7B takes only 82 minutes, substantially faster than SliM-LLM (158 minutes). Although TWLA is slightly slower than QuaRot, the accuracy gain it delivers (an average reduction of about 12 points in Wiki-Text2 perplexity across the LLaMA family) makes its overhead moderate and practically acceptable.

## 5. Conclusion

In this work, we propose TWLA, a PTQ framework that achieves 1.58-bit weight compression together with 4-bit activation quantization while maintaining high accuracy. TWLA targets the key bottlenecks in this extreme quantization regime by systematically mitigating the mismatch between weight distributions and ternary codebooks and suppressing heavy-tailed activation outliers, thereby achieving a strong balance between efficiency and accuracy in a highly challenging setting. Extensive experiments show that TWLA provides a practical PTQ solution for ternary weights with low-bit activations and offers a scalable path toward efficient LLM inference.

---

[1]https://github.com/ggml-org/llama.cpp

## Impact Statement

TWLA is a post-training quantization (PTQ) framework that enables accurate inference with ternary weights and low-bit activations, substantially reducing memory footprint and improving throughput for large language models. This capability can lower the cost and energy required to deploy LLMs, and may broaden access to LLM-based applications in resource-constrained environments (e.g., on-device or edge deployment), potentially benefiting privacy-sensitive and latency-critical use cases. Although our approach aims to make LLMs more accessible and widely used, it does not address the potential risks of misuse for malicious purposes. To mitigate these risks, a strong commitment to user data protection, clear ethical guidelines, and transparency mechanisms is essential.

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

## Appendix Overview

## A. Pseudocode of TWLA

For completeness, this section provides the detailed pseudocode of our TWLA framework. TWLA is a unified post-training quantization (PTQ) pipeline for large language models, aiming to jointly enable 1.58-bit ternary weights and low-bit activations (e.g., 2–6-bit mixed precision) while maintaining competitive accuracy.

As described in Section 3, TWLA consists of three key modules, highlighted in the pseudocode via color-coded comment blocks: (1) KOTMS (Kronecker Orthogonal Tri-Modal Shaping): For each layer, we construct a structured rotation matrix $R_\ell = R_{1,\ell} \otimes R_{2,\ell}$ from two lightweight Kronecker-structured orthogonal factors $R_{1,\ell} \in \mathbb{R}^{m_1 \times m_1}$ and $R_{2,\ell} \in \mathbb{R}^{m_2 \times m_2}$. To obtain a suitable decomposition, we first factorize the rotation dimension into $(m_1, m_2)$ using Algorithm 1, enabling efficient construction of the orthogonal Kronecker structure. KOTMS performs function-preserving orthogonal mixing to reshape the typically unimodal, near-zero-concentrated and heavy-tailed weights into a tri-modal form that better matches ternary codebooks, and it statistically suppresses activation heavy tails and outliers through the same orthogonal mixing. The full optimization procedure is summarized in Algorithm 2.

---

**Algorithm 1** Kronecker Dimension Factorization (for KOTMS)

---

**Require:** Hidden dimension $d$ to be rotated (typically $d = m$ for $W \in \mathbb{R}^{n \times m}$), $d \geq 1$
**Ensure:** Factor pair $(d_1, d_2)$ such that $d = d_1 d_2$ and $d_1 \approx d_2$
1: $a \leftarrow \lfloor \sqrt{d} \rfloor$      // start near $\sqrt{d}$
2: **while** $a > 1$ **and** $d \bmod a \neq 0$ **do**
3:      $a \leftarrow a - 1$      // search closest divisor below $\sqrt{d}$
4: **end while**
5: $d_2 \leftarrow a$
6: $d_1 \leftarrow d/d_2$
7: **return** $(d_1, d_2)$

---

(2) E2M-ATQ (Euclidean-to-Manifold Asymmetric Ternary Quantizer): After KOTMS shaping, we ternarize each layer using a two-stage Euclidean-to-Manifold strategy. In Stage I, we alternate updates of the discrete ternary pattern and continuous row-wise parameters under Frobenius (Euclidean) geometry to obtain a stable ternary structural prior. In Stage

---

**Algorithm 2** KOTMS: Kronecker Orthogonal Tri-Modal Shaping

---

**Require:** Weight matrix $W \in \mathbb{R}^{n \times m}$; target sparsity $\rho$; prior $\pi_0$; penalty $\beta$
**Require:** Steps $T_{\text{kotms}}$; step size $\eta$; numerical floor $\sigma_{\min}$
**Ensure:** Kronecker orthogonal factors $(R_1, R_2)$ and implicit rotation $R = R_1 \otimes R_2$
1:  $\pi_+ \leftarrow (1 - \pi_0)/2, \quad \pi_- \leftarrow (1 - \pi_0)/2$
2:  $(m_1, m_2) \leftarrow \text{KRONECKERFACTORIZE}(m)$            // see Alg. 1
3:  Initialize $S_1 \in \mathbb{R}^{m_1 \times m_1}$, $S_2 \in \mathbb{R}^{m_2 \times m_2}$ (e.g., zeros)
    **Orthogonal Kronecker rotation via Cayley parameterization**
4:  **for** $t = 1$ **to** $T_{\text{kotms}}$ **do**
5:     $A_1 \leftarrow S_1 - S_1^\top, \quad A_2 \leftarrow S_2 - S_2^\top$
6:     $R_1 \leftarrow (I + A_1)^{-1}(I - A_1), \quad R_2 \leftarrow (I + A_2)^{-1}(I - A_2)$
7:     $R \leftarrow R_1 \otimes R_2, \quad Z \leftarrow WR$
8:     Tri-modal shaping loss (row-wise TriGMM + zero-mass regularizer)
9:     **for** $i = 1, 2, \ldots, n$ **do**
10:       $c_i \leftarrow \frac{1}{m} \sum_{j=1}^{m} |z_{ij}|, \quad \sigma_i \leftarrow \max(\text{std}(\{z_{ij}\}_{j=1}^m), \sigma_{\min})$
11:       Compute responsibilities $\{r_{ij+}, r_{ij0}, r_{ij-}\}_{j=1}^m$ using $(\pi_+, \pi_0, \pi_-)$ and means $(+c_i, 0, -c_i)$
12:       $\bar{r}_{i0} \leftarrow \frac{1}{m} \sum_{j=1}^{m} r_{ij0}$
13:    **end for**
14:    $L_{\text{shape}} \leftarrow L_{\text{TriGMM}}(Z) + \beta \cdot \frac{1}{n} \sum_i (\bar{r}_{i0} - \rho)^2$
15:    $(S_1, S_2) \leftarrow (S_1, S_2) - \eta \nabla L_{\text{shape}}$
16: **end for**
17: **return** $(R_1, R_2)$

---

II, we fix the ternary pattern and relocate the row-wise shift/scale parameters under the calibration-induced metric defined by the activation second moment, yielding a metric-consistent closed-form update that directly aligns the forward output error. The pseudocode is provided in Algorithm 3.

(3) ILA-AMP (Inter-Layer Aware Activation Mixed Precision): Although KOTMS can indirectly improve activation quantizability, the gain is often layer-dependent and may propagate through deep Transformers, causing a few low-bit layers to become bottlenecks and trigger cascading degradation. To address this, we treat the activation bitwidth as a controllable degree of freedom and allocate it across layers under a global budget. ILA-AMP constructs an efficient second-order surrogate based on the average validation NLL, including unary per-layer costs and adjacent pairwise interaction costs to capture local coupling amplification induced by distribution shifts and error propagation. Thanks to the chain-structured objective, the optimal mixed-precision configuration can be obtained exactly via dynamic programming. The procedure is summarized in Algorithm 4.

Overall, Algorithm 5 presents the simplified end-to-end TWLA pipeline: we first learn and fold Kronecker orthogonal rotations per layer (KOTMS), then perform E2M-ATQ ternarization in the rotated coordinate system, and finally solve the budget-constrained inter-layer-aware mixed-precision activation assignment with ILA-AMP. The resulting pseudocode faithfully reflects the PTQ pipeline used in all our experiments and can be directly applied to large language models at different scales.

## B. Detailed Proofs and Derivations

### B.1. Monotonic Decrease of the Euclidean Initialization Updates

This appendix justifies the statement in the main text that the Euclidean initialization updates follow a coordinate-descent scheme and yield a monotonically non-increasing objective (cf. Eq. (2) in the main paper).

#### B.1.1. ROW-WISE OBJECTIVE

We analyze a single row for clarity; the extension to the full matrix follows by summing over rows. Let $\mathbf{w} \in \mathbb{R}^m$ be a row of the weight matrix, and let $\mu \in \mathbb{R}$, $\alpha \in \mathbb{R}$, and $\mathbf{t} \in \{-1, 0, 1\}^m$ be the row-wise shift, scale, and ternary pattern, respectively.

---

**Algorithm 3** E2M-ATQ: Euclidean Warm-start → Manifold Relocation

---

**Require:** Weight matrix $W \in \mathbb{R}^{n \times m}$; calibration inputs $X$ for this layer
**Require:** Euclidean iters $T_{\mathrm{euc}}$; TWN-style threshold rule (e.g., $\Delta$); metric floor (optional)
**Ensure:** Ternary codebook $T^{(0)} \in \{-1, 0, +1\}^{n \times m}$, row-wise $(\mu, \alpha)$, ternary-dequantized $\bar{W}$

    **Stage I: Euclidean warm-start (stabilize discrete pattern)**
1:  $\mu \leftarrow \mathrm{row\_mean}(W)$
2:  Initialize $T \leftarrow \text{\textsc{TernaryThreshold}}(W - \mu, \Delta)$
3:  $\alpha \leftarrow \text{\textsc{SupportAwareLS}}(W, \mu, T)$
4:  **for** $t = 1$ **to** $T_{\mathrm{euc}}$ **do**
5:     $R \leftarrow W - \mu \mathbf{1}^\top - \mathrm{diag}(\alpha)T$           // residual
6:     $\mu \leftarrow \mu + \frac{1}{m}R\mathbf{1}$           // residual-mean correction
7:     $\alpha \leftarrow \text{\textsc{SupportAwareLS}}(W, \mu, T)$
8:     Update $\Delta$ and $T \leftarrow \text{\textsc{TernaryThreshold}}(W - \mu, \Delta)$
9:  **end for**
10: $T^{(0)} \leftarrow T$           // freeze stratum
    **Stage II: Manifold relocation under calibration-induced metric**
11: $S \leftarrow \sum_b X_b^\top X_b$           // second moment
12: **for** $i = 1, 2, \ldots, n$ **do**
13:     $t_i \leftarrow T_{i:}^{(0)}, \quad w_i \leftarrow W_{i:}$
14:     Form $2 \times 2$ system: $A_i = \begin{bmatrix} t_i S t_i^\top & t_i S \mathbf{1} \\ \mathbf{1}^\top S t_i^\top & \mathbf{1}^\top S \mathbf{1} \end{bmatrix}, \; b_i = \begin{bmatrix} t_i S w_i^\top \\ \mathbf{1}^\top S w_i^\top \end{bmatrix}$
15:     Solve $A_i \begin{bmatrix} \alpha_i \\ \mu_i \end{bmatrix} = b_i$
16: **end for**
17: $\bar{W} \leftarrow \mu \mathbf{1}^\top + \mathrm{diag}(\alpha)T^{(0)}$
18: **return** $(T^{(0)}, \mu, \alpha, \bar{W})$

---

Define the (row-wise) quantization error

$$\mathcal{L}(\mu, \alpha, \mathbf{t}) \;=\; \left\| \mathbf{w} - \mu \mathbf{1} - \alpha \mathbf{t} \right\|_2^2, \tag{22}$$

where $\mathbf{1} \in \mathbb{R}^m$ is the all-one vector. For a ternary vector $\mathbf{t}$, we denote its support size by

$$s(\mathbf{t}) \;=\; \|\mathbf{t}\|_0 \;=\; \sum_{k=1}^m |t_k| \;=\; \sum_{k=1}^m t_k^2, \tag{23}$$

which corresponds to the number of non-zero ternary entries (i.e., excluding ternary zeros).

B.1.2. Update Rules (Coordinate Descent)

Given $(\mu^\tau, \alpha^\tau, \mathbf{t}^\tau)$ at iteration $\tau$, we perform the following updates:

1. **$\mu$-update.** With $(\alpha^\tau, \mathbf{t}^\tau)$ fixed, update

$$\mu^{\tau+1} \;=\; \mu^\tau + \delta_\mu^\tau, \qquad \delta_\mu^\tau \;=\; \frac{1}{m} \mathbf{1}^\top \left( \mathbf{w} - \mu^\tau \mathbf{1} - \alpha^\tau \mathbf{t}^\tau \right). \tag{24}$$

2. **$\alpha$-update (support-aware).** With $(\mu^{\tau+1}, \mathbf{t}^\tau)$ fixed, update

$$\alpha^{\tau+1} \;=\; \arg\min_\alpha \left\| \mathbf{w} - \mu^{\tau+1}\mathbf{1} - \alpha \mathbf{t}^\tau \right\|_2^2 \;=\; \frac{\sum_{k=1}^m t_k^\tau (w_k - \mu^{\tau+1})}{\sum_{k=1}^m |t_k^\tau|}, \tag{25}$$

    where the denominator excludes ternary zeros ($t_k^\tau = 0$).

3. **$\mathbf{t}$-update (row-wise ternary thresholding).** With $(\mu^{\tau+1}, \alpha^{\tau+1})$ fixed, update $\mathbf{t}$ by row-wise ternary thresholding, which is equivalent to the element-wise minimization

$$t_k^{\tau+1} \in \arg\min_{t \in \{-1,0,1\}} (w_k - \mu^{\tau+1} - \alpha^{\tau+1} t)^2, \qquad k = 1, \ldots, m. \tag{26}$$

---

**Algorithm 4** ILA-AMP: Inter-Layer Aware Activation Mixed Precision

---

**Require:** Quantized-weight model with $L$ layers; bit candidates $\mathcal{B} = \{2, 4, 6, 8\}$; reference $b_{\max} = 8$
**Require:** Validation set $\mathcal{D}_{\mathrm{val}}$; layer weights $\{w_\ell\}$; budget $B$
**Ensure:** Optimal activation bits $\{b_\ell^\star\}_{\ell=1}^L$
    **Build unary and adjacent pairwise costs via NLL**
  1: Set all layers to $b_{\max}$; BaseNLL $\leftarrow v_{\mathrm{NLL}}(\mathbf{b}_{\max})$
  2: **for** $\ell = 1, 2, \ldots, L$ **do**
  3:    **for** each $b \in \mathcal{B}$ **do**
  4:       Set all layers to $b_{\max}$; set layer $\ell$ to $b$
  5:       $C_\ell(b) \leftarrow v_{\mathrm{NLL}}(b_\ell = b,\ \text{others } b_{\max}) - \text{BaseNLL}$
  6:    **end for**
  7: **end for**
  8: **for** $\ell = 2, 3, \ldots, L$ **do**
  9:    **for** each $b' \in \mathcal{B}$ **do**
10:      **for** each $b \in \mathcal{B}$ **do**
11:        Set all layers to $b_{\max}$; set layers $(\ell-1, \ell)$ to $(b', b)$
12:        $K_{\ell-1,\ell}(b', b) \leftarrow v_{\mathrm{NLL}}(\ell-1 = b', \ell = b) - \text{BaseNLL} - C_{\ell-1}(b') - C_\ell(b)$
13:      **end for**
14:    **end for**
15: **end for**
    **Dynamic programming under global budget**
16: Initialize $\mathrm{DP}[\ell][c][b] \leftarrow +\infty$
17: **for** each $b \in \mathcal{B}$ **do**
18:    $c \leftarrow w_1 b$;
19:    **if** $c \le B$ **then**
20:       $\mathrm{DP}[1][c][b] \leftarrow C_1(b)$
21:    **end if**
22: **end for**
23: **for** $\ell = 2, 3, \ldots, L$ **do**
24:    **for** $c = 0, 1, \ldots, B$ **do**
25:      **for** each $b \in \mathcal{B}$ **do**
26:        **if** $c - w_\ell b \ge 0$ **then**
27:          $\mathrm{DP}[\ell][c][b] \leftarrow C_\ell(b) + \min_{b' \in \mathcal{B}} \big( \mathrm{DP}[\ell-1][c - w_\ell b][b'] + K_{\ell-1,\ell}(b', b) \big)$
28:          Record argmin pointer for backtracking
29:        **end if**
30:      **end for**
31:    **end for**
32: **end for**
33: Backtrack from $\min_{c \le B} \min_{b \in \mathcal{B}} \mathrm{DP}[L][c][b]$ to get $\{b_\ell^\star\}$
34: **return** $\{b_\ell^\star\}$

---

## B.1.3. MONOTONIC DECREASE

**Lemma B.1** (Decrease after the $\mu$-update). *Let $\mathbf{r}^\tau = \mathbf{w} - \mu^\tau \mathbf{1} - \alpha^\tau \mathbf{t}^\tau$ be the residual. After updating $\mu$ as in Eq. 24 (with $\alpha^\tau$, $\mathbf{t}^\tau$ fixed), the objective decreases by*

$$\mathcal{L}(\mu^\tau, \alpha^\tau, \mathbf{t}^\tau) - \mathcal{L}(\mu^{\tau+1}, \alpha^\tau, \mathbf{t}^\tau) = m(\delta_\mu^\tau)^2 \ge 0. \tag{27}$$

*Proof.* Since $\mathbf{r}^{\tau+1} = \mathbf{r}^\tau - \delta_\mu^\tau \mathbf{1}$,

$$\|\mathbf{r}^{\tau+1}\|_2^2 = \|\mathbf{r}^\tau\|_2^2 - 2\delta_\mu^\tau \mathbf{1}^\top \mathbf{r}^\tau + m(\delta_\mu^\tau)^2.$$

Using $\delta_\mu^\tau = \frac{1}{m} \mathbf{1}^\top \mathbf{r}^\tau$ yields $\|\mathbf{r}^{\tau+1}\|_2^2 = \|\mathbf{r}^\tau\|_2^2 - m(\delta_\mu^\tau)^2$.    □

---

**Algorithm 5** TWLA: Overall PTQ Pipeline

---

**Require:** LLM with $L$ layers and weights $\{W_\ell\}$; calibration set $\mathcal{D}_{\text{cal}}$; validation set $\mathcal{D}_{\text{val}}$
**Require:** KOTMS hyperparams; E2M-ATQ hyperparams; ILA-AMP hyperparams and budget $B$
**Ensure:** Ternary weights $\{\bar{W}_\ell\}$, rotations $\{(R_{1,\ell}, R_{2,\ell})\}$, activation bits $\{b_\ell^\star\}$

    **Step 1: KOTMS (function-preserving rotation + tri-modal shaping)**
1: **for** $\ell = 1, 2, \ldots, L$ **do**
2:    $(R_{1,\ell}, R_{2,\ell}) \leftarrow \text{KOTMS}(W_\ell)$                                  // Alg. 2
3:    $R_\ell \leftarrow R_{1,\ell} \otimes R_{2,\ell}$
4:    $W_\ell \leftarrow W_\ell R_\ell$; fold/insert $R_\ell^\top$ onto activation pathway
5: **end for**
    **Step 2: E2M-ATQ (ternarize weights with Euclidean-to-Manifold updates)**
6: **for** $\ell = 1, 2, \ldots, L$ **do**
7:    Collect layer inputs $X_\ell$ from $\mathcal{D}_{\text{cal}}$
8:    $(T_\ell^{(0)}, \mu_\ell, \alpha_\ell, \bar{W}_\ell) \leftarrow \text{E2M-ATQ}(W_\ell, X_\ell)$                      // Alg. 3
9:    Replace $W_\ell$ by $\bar{W}_\ell$
10: **end for**
    **Step 3: ILA-AMP (allocate activation bits via DP with adjacent interactions)**
11: $\{b_\ell^\star\}_{\ell=1}^L \leftarrow \text{ILA-AMP}(\mathcal{D}_{\text{val}}, B)$                                // Alg. 4
12: Deploy activation quantizers according to $\{b_\ell^\star\}$
13: **return** Quantized model

---

**Lemma B.2** (Decrease after the support-aware $\alpha$-update). *Fix $(\mu, \mathbf{t})$ and let $\alpha^\star$ be the minimizer in Eq. 25. Then for any $\alpha$,*

$$\mathcal{L}(\mu, \alpha, \mathbf{t}) - \mathcal{L}(\mu, \alpha^\star, \mathbf{t}) = \|\mathbf{t}\|_2^2 (\alpha - \alpha^\star)^2 = s(\mathbf{t})(\alpha - \alpha^\star)^2 \geq 0. \tag{28}$$

*Proof.* Let $\mathbf{u} = \mathbf{w} - \mu \mathbf{1}$. Then $\mathcal{L}(\mu, \alpha, \mathbf{t}) = \|\mathbf{u} - \alpha \mathbf{t}\|_2^2$. This is a 1D least-squares problem with closed-form minimizer $\alpha^\star = \frac{\mathbf{t}^\top \mathbf{u}}{\mathbf{t}^\top \mathbf{t}}$. Expanding $\|\mathbf{u} - \alpha \mathbf{t}\|_2^2$ around $\alpha^\star$ gives $\|\mathbf{u} - \alpha \mathbf{t}\|_2^2 = \|\mathbf{u} - \alpha^\star \mathbf{t}\|_2^2 + (\mathbf{t}^\top \mathbf{t})(\alpha - \alpha^\star)^2$. Since $\mathbf{t}^\top \mathbf{t} = \sum_k t_k^2 = \sum_k |t_k| = s(\mathbf{t})$, we obtain Eq. 28. $\qquad\square$

**Lemma B.3** (Non-increase after the $\mathbf{t}$-update). *Fix $(\mu, \alpha)$. If $\mathbf{t}^\star$ is updated element-wise as in Eq. 26, then*

$$\mathcal{L}(\mu, \alpha, \mathbf{t}^\star) \leq \mathcal{L}(\mu, \alpha, \mathbf{t}). \tag{29}$$

*Proof.* With $(\mu, \alpha)$ fixed, the objective decomposes as $\mathcal{L}(\mu, \alpha, \mathbf{t}) = \sum_{k=1}^m (w_k - \mu - \alpha t_k)^2$. Choosing each $t_k^\star \in \arg\min_{t \in \{-1,0,1\}} (w_k - \mu - \alpha t)^2$ minimizes every summand and therefore cannot increase the sum. $\qquad\square$

**Proposition B.4** (Monotonic decrease and convergence). *Let $\mathcal{L}^\tau = \mathcal{L}(\mu^\tau, \alpha^\tau, \mathbf{t}^\tau)$. The update order $\mu \to \alpha \to \mathbf{t}$ satisfies*

$$\mathcal{L}^{\tau+1} \leq \mathcal{L}^\tau, \qquad \forall \tau \geq 0, \tag{30}$$

*and the sequence $\{\mathcal{L}^\tau\}_{\tau \geq 0}$ converges. Moreover, we have the explicit descent bound*

$$\mathcal{L}^{\tau+1} \leq \mathcal{L}^\tau - m(\delta_\mu^\tau)^2 - s(\mathbf{t}^\tau)(\alpha^\tau - \alpha^{\tau+1})^2, \tag{31}$$

*where the $\mathbf{t}$-update may further decrease the objective.*

*Proof.* Applying Lemma B.1 gives a decrease after the $\mu$-update. Then Lemma B.2 yields a decrease after the $\alpha$-update. Finally, Lemma B.3 shows the $\mathbf{t}$-update is non-increasing. Thus $\mathcal{L}^{\tau+1} \leq \mathcal{L}^\tau$ for all $\tau$. Since $\mathcal{L}^\tau \geq 0$, the monotone sequence converges. Combining Lemma B.1 and Lemma B.2 yields Eq. 31; the $\mathbf{t}$-update can only decrease further. $\qquad\square$

**Extension to the full matrix.** For a matrix $\mathbf{W} \in \mathbb{R}^{n \times m}$ with row-wise parameters $\boldsymbol{\mu} \in \mathbb{R}^n$, $\boldsymbol{\alpha} \in \mathbb{R}^n$, and $\mathbf{T} \in \{-1, 0, 1\}^{n \times m}$, the full Euclidean objective is the sum of row-wise objectives in Eq. 22. Therefore, the same monotonic non-increase holds for the entire update procedure.

## B.2. Row-wise Normal Equations and Numerical Stabilization for Metric-Aware Relocation

This section provides a step-by-step derivation of the row-wise $2 \times 2$ linear system in Eq. 8 by optimizing $(\boldsymbol{\mu}, \boldsymbol{\alpha})$ under the calibration-defined metric induced by $\mathbf{S} \succeq \mathbf{0}$ while fixing $\mathbf{T} = \mathbf{T}^{(0)}$. We also describe practical numerical stabilization used when $\mathbf{S}$ is ill-conditioned or the $2 \times 2$ system becomes nearly singular.

**Row-wise decoupling from the trace form.** Recall $\mathbf{R} = \mathbf{W} - \boldsymbol{\mu}\mathbf{1}^\top - \mathrm{diag}(\boldsymbol{\alpha})\,\mathbf{T}^{(0)}$ and the objective (Eq. **??**)

$$L_2(\boldsymbol{\mu}, \boldsymbol{\alpha}; \mathbf{T}^{(0)}) \;=\; \mathrm{Tr}\big(\mathbf{R}\mathbf{S}\mathbf{R}^\top\big), \qquad \mathbf{S} = \mathbf{S}^\top \succeq \mathbf{0}. \tag{32}$$

Let $\mathbf{r}_i \in \mathbb{R}^{1 \times m}$ be the $i$-th row of $\mathbf{R}$, and let $\mathbf{w}_i, \mathbf{t}_i$ denote the $i$-th rows of $\mathbf{W}$ and $\mathbf{T}^{(0)}$, respectively. Then

$$\mathbf{r}_i = \mathbf{w}_i - \mu_i \mathbf{1}^\top - \alpha_i \mathbf{t}_i. \tag{33}$$

Using the identity $\mathrm{Tr}(\mathbf{A}\mathbf{A}^\top) = \sum_i \mathbf{a}_i \mathbf{a}_i^\top$ for row vectors $\mathbf{a}_i$, we obtain

$$\mathrm{Tr}(\mathbf{R}\mathbf{S}\mathbf{R}^\top) = \sum_{i=1}^{n} \big(\mathbf{R}\mathbf{S}\mathbf{R}^\top\big)_{ii} = \sum_{i=1}^{n} \mathbf{r}_i \mathbf{S} \mathbf{r}_i^\top. \tag{34}$$

Hence the optimization decouples across rows:

$$\min_{\boldsymbol{\mu}, \boldsymbol{\alpha}} \mathrm{Tr}(\mathbf{R}\mathbf{S}\mathbf{R}^\top) \quad \Longleftrightarrow \quad \forall i: \min_{\mu_i, \alpha_i} f_i(\mu_i, \alpha_i), \qquad f_i(\mu_i, \alpha_i) = \mathbf{r}_i \mathbf{S} \mathbf{r}_i^\top. \tag{35}$$

**Step-by-step differentiation (two equivalent derivations).** Fix a row $i$ and define

$$\mathbf{u}_i \triangleq \mathbf{r}_i = \mathbf{w}_i - \mu_i \mathbf{1}^\top - \alpha_i \mathbf{t}_i, \qquad f_i(\mu_i, \alpha_i) = \mathbf{u}_i \mathbf{S} \mathbf{u}_i^\top. \tag{36}$$

We now compute $\partial f_i / \partial \mu_i$ and $\partial f_i / \partial \alpha_i$ in a fully explicit manner.

*(A) Differential (matrix calculus) route.* Since $\mathbf{S} = \mathbf{S}^\top$, the differential of the quadratic form satisfies

$$\begin{aligned}
\mathrm{d}f_i = \mathrm{d}(\mathbf{u}_i \mathbf{S} \mathbf{u}_i^\top) &= (\mathrm{d}\mathbf{u}_i)\mathbf{S}\mathbf{u}_i^\top + \mathbf{u}_i \mathbf{S} (\mathrm{d}\mathbf{u}_i)^\top \\
&= (\mathrm{d}\mathbf{u}_i)\mathbf{S}\mathbf{u}_i^\top + \mathbf{u}_i \mathbf{S}^\top (\mathrm{d}\mathbf{u}_i)^\top = 2(\mathrm{d}\mathbf{u}_i)\mathbf{S}\mathbf{u}_i^\top.
\end{aligned} \tag{37}$$

From Eq. 36, we have

$$\mathrm{d}\mathbf{u}_i = -(\mathrm{d}\mu_i)\mathbf{1}^\top - (\mathrm{d}\alpha_i)\mathbf{t}_i. \tag{38}$$

Plugging Eq. 38 into Eq. 37 yields

$$\begin{aligned}
\mathrm{d}f_i &= 2\Big(-(\mathrm{d}\mu_i)\mathbf{1}^\top - (\mathrm{d}\alpha_i)\mathbf{t}_i\Big)\mathbf{S}\mathbf{u}_i^\top \\
&= -2(\mathrm{d}\mu_i)\underbrace{\mathbf{1}^\top\mathbf{S}\mathbf{u}_i^\top}_{\text{scalar}} -2(\mathrm{d}\alpha_i)\underbrace{\mathbf{t}_i\mathbf{S}\mathbf{u}_i^\top}_{\text{scalar}}.
\end{aligned} \tag{39}$$

By identification of coefficients of $\mathrm{d}\mu_i$ and $\mathrm{d}\alpha_i$, we obtain

$$\frac{\partial f_i}{\partial \mu_i} = -2\,\mathbf{1}^\top\mathbf{S}\mathbf{u}_i^\top, \qquad \frac{\partial f_i}{\partial \alpha_i} = -2\,\mathbf{t}_i\mathbf{S}\mathbf{u}_i^\top. \tag{40}$$

*(B) Fully expanded scalar route (optional but explicit).* Write $f_i = \sum_{p=1}^{m}\sum_{q=1}^{m} u_{ip}S_{pq}u_{iq}$, where $u_{ip} = w_{ip} - \mu_i - \alpha_i t_{ip}$. Then

$$\frac{\partial f_i}{\partial \alpha_i} = \sum_{p,q}\left(\frac{\partial u_{ip}}{\partial \alpha_i}S_{pq}u_{iq} + u_{ip}S_{pq}\frac{\partial u_{iq}}{\partial \alpha_i}\right) = \sum_{p,q}\big(-t_{ip}S_{pq}u_{iq} - u_{ip}S_{pq}t_{iq}\big).$$

Using symmetry $S_{pq} = S_{qp}$ and re-indexing the second term gives $\frac{\partial f_i}{\partial \alpha_i} = -2\sum_{p,q} t_{ip}S_{pq}u_{iq} = -2\,\mathbf{t}_i\mathbf{S}\mathbf{u}_i^\top$, matching Eq. 40. The derivation for $\partial f_i/\partial \mu_i$ is identical with $\partial u_{ip}/\partial \mu_i = -1$, yielding $-2\,\mathbf{1}^\top\mathbf{S}\mathbf{u}_i^\top$.

**From stationarity to the $2 \times 2$ linear system.** Setting Eq. 40 to zero yields the stationarity conditions

$$\mathbf{t}_i \mathbf{S} \mathbf{u}_i^\top = 0, \qquad \mathbf{1}^\top \mathbf{S} \mathbf{u}_i^\top = 0. \tag{41}$$

Substitute $\mathbf{u}_i^\top = \mathbf{w}_i^\top - \mu_i \mathbf{1} - \alpha_i \mathbf{t}_i^\top$ into Eq. 41:

$$\mathbf{t}_i \mathbf{S} \big( \mathbf{w}_i^\top - \mu_i \mathbf{1} - \alpha_i \mathbf{t}_i^\top \big) = 0, \tag{42}$$

$$\mathbf{1}^\top \mathbf{S} \big( \mathbf{w}_i^\top - \mu_i \mathbf{1} - \alpha_i \mathbf{t}_i^\top \big) = 0. \tag{43}$$

Expand each equation and collect coefficients of $\alpha_i$ and $\mu_i$:

$$(\mathbf{t}_i \mathbf{S} \mathbf{t}_i^\top) \alpha_i + (\mathbf{t}_i \mathbf{S} \mathbf{1}) \mu_i = \mathbf{t}_i \mathbf{S} \mathbf{w}_i^\top, \tag{44}$$

$$(\mathbf{1}^\top \mathbf{S} \mathbf{t}_i^\top) \alpha_i + (\mathbf{1}^\top \mathbf{S} \mathbf{1}) \mu_i = \mathbf{1}^\top \mathbf{S} \mathbf{w}_i^\top. \tag{45}$$

Writing Eqs. 44–45 in matrix form gives the $2 \times 2$ normal equations:

$$\begin{bmatrix} \mathbf{t}_i \mathbf{S} \mathbf{t}_i^\top & \mathbf{t}_i \mathbf{S} \mathbf{1} \\ \mathbf{1}^\top \mathbf{S} \mathbf{t}_i^\top & \mathbf{1}^\top \mathbf{S} \mathbf{1} \end{bmatrix} \begin{bmatrix} \alpha_i \\ \mu_i \end{bmatrix} = \begin{bmatrix} \mathbf{t}_i \mathbf{S} \mathbf{w}_i^\top \\ \mathbf{1}^\top \mathbf{S} \mathbf{w}_i^\top \end{bmatrix}, \tag{46}$$

which matches Eq. 8 in the main text.

**Closed-form solution via Cramer's rule.** Define the scalars

$$a_i = \mathbf{t}_i \mathbf{S} \mathbf{t}_i^\top, \quad b_i = \mathbf{t}_i \mathbf{S} \mathbf{1}, \quad c = \mathbf{1}^\top \mathbf{S} \mathbf{1}, \quad d_i = \mathbf{t}_i \mathbf{S} \mathbf{w}_i^\top, \quad e_i = \mathbf{1}^\top \mathbf{S} \mathbf{w}_i^\top. \tag{47}$$

Then Eq. 46 becomes $\begin{bmatrix} a_i & b_i \\ b_i & c \end{bmatrix} \begin{bmatrix} \alpha_i \\ \mu_i \end{bmatrix} = \begin{bmatrix} d_i \\ e_i \end{bmatrix}$. Its determinant is

$$D_i = a_i c - b_i^2. \tag{48}$$

If $D_i \neq 0$, the unique solution is

$$\alpha_i^* = \frac{d_i c - b_i e_i}{D_i}, \qquad \mu_i^* = \frac{a_i e_i - b_i d_i}{D_i}. \tag{49}$$

**Uniqueness and degeneracy (why $D_i$ can be small).** The matrix in Eq. 46 is the Gram matrix of the two vectors $\mathbf{t}_i^\top$ and $\mathbf{1}$ under the $\mathbf{S}$-induced inner product $\langle \mathbf{x}, \mathbf{y} \rangle_\mathbf{S} = \mathbf{x}^\top \mathbf{S} \mathbf{y}$:

$$G_i = \begin{bmatrix} \langle \mathbf{t}_i^\top, \mathbf{t}_i^\top \rangle_\mathbf{S} & \langle \mathbf{t}_i^\top, \mathbf{1} \rangle_\mathbf{S} \\ \langle \mathbf{1}, \mathbf{t}_i^\top \rangle_\mathbf{S} & \langle \mathbf{1}, \mathbf{1} \rangle_\mathbf{S} \end{bmatrix} = \begin{bmatrix} a_i & b_i \\ b_i & c \end{bmatrix}.$$

Since $\mathbf{S} \succeq \mathbf{0}$, we have $a_i \geq 0$, $c \geq 0$, and $D_i = \det(G_i) \geq 0$. Moreover, $D_i = 0$ if and only if $\mathbf{t}_i^\top$ and $\mathbf{1}$ are colinear in the seminorm induced by $\mathbf{S}$ (i.e., linearly dependent modulo the nullspace of $\mathbf{S}$ on $\mathrm{span}\{\mathbf{t}_i^\top, \mathbf{1}\}$). A practically important degenerate case is $\mathbf{t}_i = \mathbf{0}$ (all ternary zeros), which implies $a_i = b_i = d_i = 0$ and hence $D_i = 0$; then $f_i$ does not depend on $\alpha_i$ and the optimal solution reduces to

$$\mu_i^* = \frac{\mathbf{1}^\top \mathbf{S} \mathbf{w}_i^\top}{\mathbf{1}^\top \mathbf{S} \mathbf{1}}, \qquad \alpha_i^* = 0. \tag{50}$$

**Practical numerical stabilization.** With limited calibration samples, $\mathbf{S}$ can be ill-conditioned, making $D_i$ numerically close to zero even when theoretically positive. We apply moment regularization

$$\mathbf{S} \leftarrow \mathbf{S} + \varepsilon \mathbf{I}, \qquad \varepsilon > 0, \tag{51}$$

which improves conditioning and increases the Gram determinant $D_i$ away from machine precision. In our implementation, $\varepsilon$ is chosen as a small fraction of the average diagonal mass, e.g., $\varepsilon = \lambda \cdot \mathrm{Tr}(\mathbf{S})/m$ with $\lambda \in [10^{-6}, 10^{-3}]$.

When $|D_i|$ remains very small for rare rows, we safely solve Eq. 46 using a stable routine (e.g., a direct $2 \times 2$ solve in double precision) and apply a conservative fallback: if $|D_i| < \tau$ (a small threshold), we use the Euclidean warm-start values $(\mu_i^{(0)}, \alpha_i^{(0)})$ rather than the potentially unstable closed form. These safeguards preserve the formulation while improving robustness under extreme activation statistics.

**Reconstruction.** After solving all rows, we aggregate $\boldsymbol{\mu}^*$ and $\boldsymbol{\alpha}^*$ and reconstruct the relocated E2M-ATQ weights as

$$\bar{\mathbf{W}} = \boldsymbol{\mu}^* \mathbf{1}^\top + \operatorname{diag}(\boldsymbol{\alpha}^*)\, \mathbf{T}^{(0)}. \tag{52}$$

## B.3. Storage and Computational Complexity of Kronecker-Structured Orthogonal Transforms

We analyze the storage and computational cost of parameterizing a large auxiliary transform $\mathbf{R} \in \mathbb{R}^{m \times m}$ as a Kronecker product of two smaller orthogonal factors. This provides a practical alternative to learning a dense $m \times m$ transform.

### B.3.1. DENSE TRANSFORM IS PROHIBITIVE AT LLM DIMENSIONS

A dense auxiliary matrix $\mathbf{R} \in \mathbb{R}^{m \times m}$ requires storing $m^2$ parameters. Applying it to a vector $\mathbf{v} \in \mathbb{R}^{1 \times m}$ costs $O(m^2)$ multiply-adds. For LLM hidden sizes $m \in \{4096, 5120, 8192, \ldots\}$, both the $m^2$ storage and $m^2$ compute become prohibitive.

### B.3.2. KRONECKER PARAMETERIZATION

We parameterize the auxiliary transform as

$$\mathbf{R} = \mathbf{R}_1 \otimes \mathbf{R}_2, \qquad \mathbf{R}_1 \in \mathbb{R}^{n_1 \times n_1}, \quad \mathbf{R}_2 \in \mathbb{R}^{n_2 \times n_2}, \quad n_1 n_2 = m, \tag{53}$$

where each factor is orthogonal: $\mathbf{R}_1^\top \mathbf{R}_1 = \mathbf{I}_{n_1}$ and $\mathbf{R}_2^\top \mathbf{R}_2 = \mathbf{I}_{n_2}$, which implies $\mathbf{R}^\top \mathbf{R} = \mathbf{I}_m$.

**Storage.** A dense $\mathbf{R}$ stores $m^2$ entries. In contrast, the Kronecker form stores only $n_1^2 + n_2^2$ entries:

$$\underbrace{m^2}_{\text{dense storage}} \quad \longrightarrow \quad \underbrace{(n_1^2 + n_2^2)}_{\text{Kronecker storage}}. \tag{54}$$

When $n_1 \approx n_2 \approx \sqrt{m}$, we have $n_1^2 + n_2^2 \approx 2m$, i.e., nearly linear storage.

**Concrete example.** For $m = 4096$, a dense matrix requires

$$m^2 = 4096^2 = 16{,}777{,}216 \tag{55}$$

stored entries. Choosing $n_1 = n_2 = 64$ (since $64 \cdot 64 = 4096$) yields only

$$n_1^2 + n_2^2 = 64^2 + 64^2 = 8192 \tag{56}$$

entries. The storage reduction factor is therefore

$$\frac{m^2}{n_1^2 + n_2^2} = \frac{16{,}777{,}216}{8192} = 2048. \tag{57}$$

Thus, the Kronecker parameterization is $2048\times$ smaller in storage than a dense $4096 \times 4096$ transform.

### B.3.3. FAST APPLICATION VIA RESHAPE–MULTIPLY–VECTORIZE

Let $\mathbf{v} \in \mathbb{R}^{1 \times m}$. Reshape it into a matrix $\mathbf{V} \in \mathbb{R}^{n_1 \times n_2}$ such that $\operatorname{vec}(\mathbf{V}) = \mathbf{v}^\top$. Using a standard Kronecker identity, we obtain

$$\mathbf{v}(\mathbf{R}_1 \otimes \mathbf{R}_2) = \operatorname{vec}\!\left(\mathbf{R}_2^\top \mathbf{V} \mathbf{R}_1\right)^\top. \tag{58}$$

Therefore, we can apply $\mathbf{R}$ without forming an $m \times m$ matrix, by computing

$$\mathbf{Y} = \mathbf{V}\mathbf{R}_1 \in \mathbb{R}^{n_1 \times n_2}, \qquad \mathbf{Z} = \mathbf{R}_2^\top \mathbf{Y} \in \mathbb{R}^{n_1 \times n_2}, \qquad \mathbf{v}' = \operatorname{vec}(\mathbf{Z})^\top \in \mathbb{R}^{1 \times m}. \tag{59}$$

**Compute complexity.** The cost of $\mathbf{V}\mathbf{R}_1$ is $O(n_1 n_2 \cdot n_1) = O(n_1^2 n_2) = O(m n_1)$. The cost of $\mathbf{R}_2^\top \mathbf{Y}$ is $O(n_2 n_1 \cdot n_2) = O(n_2^2 n_1) = O(m n_2)$. Hence the total is

$$O(m n_1) + O(m n_2) = O\!\big(m(n_1 + n_2)\big). \tag{60}$$

With $n_1 \approx n_2 \approx \sqrt{m}$, this becomes $O(m^{3/2})$, substantially smaller than $O(m^2)$.

**Concrete compute example ($m = 4096$, $n_1 = n_2 = 64$).** A dense multiplication $\mathbf{vR}$ uses on the order of $m^2 = 16.8$ million multiply-adds. With the Kronecker form, the two multiplies cost on the order of

$$m(n_1 + n_2) = 4096(64 + 64) = 524{,}288, \tag{61}$$

which is about $16{,}777{,}216/524{,}288 \approx 32\times$ fewer operations (up to constant factors). This makes a structured orthogonal auxiliary transform practical at LLM scale.

## B.4. Why Orthogonal Mixing Suppresses Activation Outliers

We provide a step-by-step argument that an orthogonal auxiliary transform can reduce activation outliers by spreading energy across coordinates, while preserving the total energy. Although our method later uses a structured orthogonal matrix, the argument below applies to any sufficiently mixing orthogonal $\mathbf{R} \in \mathbb{R}^{m \times m}$.

### B.4.1. OUTLIERS AND THE PEAK-TO-RMS RATIO

Let $\mathbf{x} \in \mathbb{R}^{1 \times m}$ denote a token-wise activation vector. A common operational notion of "outliers" is that a few coordinates are much larger than the typical scale. We quantify this with the peak-to-RMS ratio

$$\mathrm{RMS}(\mathbf{x}) := \frac{\|\mathbf{x}\|_2}{\sqrt{m}}, \qquad \kappa(\mathbf{x}) := \frac{\|\mathbf{x}\|_\infty}{\mathrm{RMS}(\mathbf{x})} = \frac{\sqrt{m}\,\|\mathbf{x}\|_\infty}{\|\mathbf{x}\|_2}. \tag{62}$$

Large $\kappa(\mathbf{x})$ indicates spiky activations: the maximum coordinate is large relative to the average energy per dimension.

### B.4.2. ORTHOGONAL TRANSFORMS PRESERVE ENERGY

Assume $\mathbf{R}$ is orthogonal: $\mathbf{R}^\top \mathbf{R} = \mathbf{I}_m$. Then, for $\mathbf{y} = \mathbf{xR}$,

$$\|\mathbf{y}\|_2 = \|\mathbf{xR}\|_2 = \|\mathbf{x}\|_2. \tag{63}$$

Thus orthogonal mixing does not change the total activation energy, but can change how energy is distributed across coordinates.

### B.4.3. A PROBABILISTIC BOUND ON THE MAXIMUM COORDINATE AFTER MIXING

We now show that for a random orthogonal $\mathbf{R}$, each coordinate of $\mathbf{y} = \mathbf{xR}$ concentrates around scale $\|\mathbf{x}\|_2/\sqrt{m}$, and the maximum over coordinates is only logarithmically larger.

**Step 1: One coordinate is an inner product with a random unit vector.** Let $\mathbf{R}_j \in \mathbb{R}^m$ be the $j$-th column of $\mathbf{R}$. For orthogonal $\mathbf{R}$ sampled uniformly at random (Haar), $\mathbf{R}_j$ is uniformly distributed on the unit sphere $S^{m-1}$. The $j$-th coordinate of $\mathbf{y}$ is

$$y_j = (\mathbf{xR})_j = \langle \mathbf{x}^\top, \mathbf{R}_j \rangle. \tag{64}$$

**Step 2: Spherical concentration for a fixed vector.** Fix $\mathbf{x}$ and let $\mathbf{u}$ be uniform on $S^{m-1}$. A standard concentration inequality on the sphere gives, for any $t > 0$,

$$\Pr\left( |\langle \mathbf{x}^\top, \mathbf{u} \rangle| \geq t\,\frac{\|\mathbf{x}\|_2}{\sqrt{m}} \right) \leq 2\exp\left( -\frac{(m-2)t^2}{2} \right). \tag{65}$$

Applying Eq. 65 to $\mathbf{u} = \mathbf{R}_j$ yields the same bound for each $y_j$.

**Step 3: Union bound over all coordinates.** Using a union bound over $j = 1, \ldots, m$,

$$\Pr\left( \|\mathbf{y}\|_\infty \geq t\,\frac{\|\mathbf{x}\|_2}{\sqrt{m}} \right) \leq 2m\exp\left( -\frac{(m-2)t^2}{2} \right). \tag{66}$$

Set the right-hand side to $\delta \in (0, 1)$ and solve for $t$:

$$2m\exp\left( -\frac{(m-2)t^2}{2} \right) \leq \delta \quad \implies \quad t \geq \sqrt{\frac{2\log(2m/\delta)}{m-2}}. \tag{67}$$

Therefore, with probability at least $1 - \delta$,

$$\|\mathbf{x}\mathbf{R}\|_\infty = \|\mathbf{y}\|_\infty \leq \|\mathbf{x}\|_2 \sqrt{\frac{2\log(2m/\delta)}{m-2}}. \tag{68}$$

**Step 4: Peak-to-RMS ratio becomes logarithmic.**  Combining Eq. 63 and Eq. 68, we obtain

$$\kappa(\mathbf{x}\mathbf{R}) = \frac{\sqrt{m}\,\|\mathbf{x}\mathbf{R}\|_\infty}{\|\mathbf{x}\mathbf{R}\|_2} \leq \sqrt{m} \cdot \sqrt{\frac{2\log(2m/\delta)}{m-2}} = \sqrt{\frac{2m}{m-2}\log\left(\frac{2m}{\delta}\right)}. \tag{69}$$

For large $m$, this scales as $\kappa(\mathbf{x}\mathbf{R}) \lesssim \sqrt{2\log(2m/\delta)}$, which grows only logarithmically in $m$. In contrast, the worst-case $\kappa(\mathbf{x})$ can be as large as $\sqrt{m}$ when energy concentrates in one coordinate.

**Step 5: Existence (probabilistic method).**  Since a random orthogonal $\mathbf{R}$ satisfies Eq. 68 with probability $1 - \delta$, there must exist at least one orthogonal matrix $\mathbf{R}$ achieving that bound. Hence orthogonal mixing is *theoretically sufficient* to suppress extreme activation peaks.

### B.4.4. IMPLICATION FOR LOW-BIT ACTIVATION QUANTIZATION

Consider symmetric uniform quantization with $b$ bits and per-tensor scale $s(\mathbf{x}) = \|\mathbf{x}\|_\infty$ (a common conservative choice). The quantization step size is $\Delta = 2s(\mathbf{x})/(2^b - 1)$. Under standard high-resolution quantization modeling, the mean squared error per coordinate scales as $O(\Delta^2)$, so reducing $s(\mathbf{x})$ directly reduces quantization error.

Applying Eq. 68 shows that, after orthogonal mixing, $s(\mathbf{x}\mathbf{R}) = \|\mathbf{x}\mathbf{R}\|_\infty$ is upper-bounded by a logarithmic factor times $\|\mathbf{x}\|_2/\sqrt{m}$, while $\|\mathbf{x}\|_2$ is unchanged. Therefore, orthogonal mixing can significantly shrink the required dynamic range (and thus $\Delta$), improving low-bit activation quantization by reducing the dominance of outliers.

### B.5. Zero-Peak Mass Regularization for TriGMM

The TriGMM objective in Eq. 12 is used as a codebook-aligned attraction field rather than a literal density fit. Without additional constraints, directly minimizing the tri-modal negative log-likelihood may admit degenerate configurations, e.g., $c_i \to 0$ (peak collapse) or an imbalanced assignment where a single component dominates, which undermines the intended ternary-aligned tri-modal shaping.

**Posterior responsibilities.**  For channel $i$ and entry $z_{ij}$, define the component means $\mu_{i+} = +c_i$, $\mu_{i0} = 0$, and $\mu_{i-} = -c_i$. Let $\phi(\cdot; \mu, \sigma^2)$ denote the Gaussian density and let the mixture weights satisfy $\pi_+ = \pi_- = (1 - \pi_0)/2$ with $\pi_0 \in (0, 1)$. The posterior responsibility of component $k \in \{+, 0, -\}$ is

$$r_{ijk} = \frac{\pi_k\,\phi(z_{ij}; \mu_{ik}, \sigma_i^2)}{\sum_{\ell \in \{+, 0, -\}} \pi_\ell\,\phi(z_{ij}; \mu_{i\ell}, \sigma_i^2)}. \tag{70}$$

We measure the *average zero-peak responsibility* in channel $i$ as

$$\bar{r}_{i0} = \frac{1}{m}\sum_{j=1}^m r_{ij0}. \tag{71}$$

**Zero-peak mass constraint.**  To explicitly control the mass assigned to the ternary zero attractor and prevent collapse, we steer $\bar{r}_{i0}$ toward a target sparsity ratio $\rho \in (0, 1)$ via

$$L_{\text{zero}} = \frac{1}{n}\sum_{i=1}^n \left(\bar{r}_{i0} - \rho\right)^2, \qquad L_{\text{shape}} = L_{\text{TriGMM}} + \beta\,L_{\text{zero}}, \tag{72}$$

where $\beta > 0$ controls the strength of the regularizer. This constraint makes the zero-peak mass an explicit and controllable structural prior, consistent with the support-set semantics of ternary quantization (i.e., entries attracted to the zero peak are encouraged to map to the 0 state), while empirically preventing degenerate tri-modal shaping such as $c_i \to 0$.

### B.6. Small-Variance Limit: TriGMM as a Soft Projection

This section justifies the interpretation that minimizing $L_{\text{TriGMM}}$ induces a soft projection of each scalar $z_{ij}$ onto the ternary-aligned attractor set $\{-c_i, 0, +c_i\}$.

**Start from the per-entry negative log-mixture.** Fix a channel $i$ and a scalar $z = z_{ij}$. Consider the negative log-mixture term

$$\ell_i(z) := -\log\Big(\pi_+\phi(z; +c_i, \sigma_i^2) + \pi_0\phi(z; 0, \sigma_i^2) + \pi_-\phi(z; -c_i, \sigma_i^2)\Big). \tag{73}$$

Using $\phi(z; \mu, \sigma^2) = \frac{1}{\sqrt{2\pi\sigma^2}} \exp\big(-\frac{(z-\mu)^2}{2\sigma^2}\big)$, we can rewrite the mixture as

$$\pi_+\phi(z; +c_i, \sigma_i^2) + \pi_0\phi(z; 0, \sigma_i^2) + \pi_-\phi(z; -c_i, \sigma_i^2) = \frac{1}{\sqrt{2\pi\sigma_i^2}} \sum_{s \in \{-c_i, 0, +c_i\}} \pi_s \exp\Big(-\frac{(z-s)^2}{2\sigma_i^2}\Big), \tag{74}$$

where we use $\pi_{+c_i} = \pi_+$, $\pi_0 = \pi_0$, and $\pi_{-c_i} = \pi_-$ for notational convenience. Substituting Eq. 74 into Eq. 73 yields

$$\ell_i(z) = \frac{1}{2} \log(2\pi\sigma_i^2) - \log\left(\sum_{s \in \{-c_i, 0, +c_i\}} \pi_s \exp\Big(-\frac{(z-s)^2}{2\sigma_i^2}\Big)\right). \tag{75}$$

**Log-sum-exp domination in the small-variance regime.** When $\sigma_i^2$ is small, the sum inside the logarithm is dominated by the largest exponential term, i.e., the attractor $s$ closest to $z$ in squared distance. Formally, applying the standard log-sum-exp approximation gives

$$-\log\left(\sum_s \pi_s \exp\Big(-\frac{(z-s)^2}{2\sigma_i^2}\Big)\right) \approx \min_{s \in \{-c_i, 0, +c_i\}} \left\{\frac{(z-s)^2}{2\sigma_i^2} - \log\pi_s\right\}. \tag{76}$$

If the mixture weights are fixed and not extreme (e.g., symmetric side peaks $\pi_+ = \pi_-$), the term $-\log\pi_s$ contributes only an additive bias compared to the dominant quadratic term. Combining Eqs. 75–76, we obtain

$$\ell_i(z) \approx \frac{1}{2\sigma_i^2} \min_{s \in \{-c_i, 0, +c_i\}} (z-s)^2 + \text{const.} \tag{77}$$

**Implication for TriGMM minimization.** Summing Eq. 77 over all entries $(i, j)$ shows that minimizing $L_{\text{TriGMM}}$ approximately minimizes the squared distance of each $z_{ij}$ to the ternary-aligned attractor set $\{-c_i, 0, +c_i\}$, i.e., it performs a differentiable *soft projection* prior to the subsequent hard ternary assignment. This explains why TriGMM shaping is geometrically consistent with the hard ternary projection step used by our quantizer.

### B.7. Why Modeling Only Adjacent-Layer Interactions is Sufficient

This subsection provides a theoretical and practical rationale for restricting second-order interaction terms to adjacent layers. Under mild stability and smoothness assumptions around the reference configuration $\mathbf{b}_{\max}$, we show that pairwise couplings between non-adjacent layers admit an explicit upper bound that decays with layer distance. We further clarify the optimization implications: while higher-order adjacent interactions (e.g., triplets) still preserve a chain structure and remain exactly solvable by dynamic programming (DP) with an augmented state, introducing interactions across arbitrary layer pairs yields a dense factor graph with high treewidth, which precludes the efficient chain-structured DP used in our solver. Finally, estimating dense interactions is significantly more expensive and less statistically stable under limited calibration data.

**Notation.** Consider a network with $L$ layers and candidate activation bitwidths $\mathcal{B} = \{2, 4, 6, 8\}$. A layer-wise configuration is denoted by $\mathbf{b} = (b_1, \ldots, b_L) \in \mathcal{B}^L$. Let $b_{\max} = 8$ and $\mathbf{b}_{\max} = (b_{\max}, \ldots, b_{\max})$ be the reference configuration. We adopt the validation average per-token negative log-likelihood (NLL) as the value function:

$$v_{\text{NLL}}(\mathbf{b}) := \mathbb{E}_u\big[\ell(x_L(u; \mathbf{b}))\big], \tag{78}$$

where $u$ indexes validation sequences, $x_L(u; \mathbf{b})$ is the final hidden state, and $\ell(\cdot)$ is the per-token NLL. We analyze the increment relative to the reference model:

$$\Delta v(\mathbf{b}) := v_{\mathrm{NLL}}(\mathbf{b}) - v_{\mathrm{NLL}}(\mathbf{b}_{\mathrm{max}}). \tag{79}$$

Recall the first-order cost:

$$C_\ell(b) = v_{\mathrm{NLL}}\Big(b_\ell = b, \ b_{k \neq \ell} = b_{\mathrm{max}}\Big) - v_{\mathrm{NLL}}(\mathbf{b}_{\mathrm{max}}), \tag{80}$$

and the adjacent interaction cost:

$$K_{\ell-1,\ell}(b', b) = v_{\mathrm{NLL}}\Big(b_{\ell-1} = b', \ b_\ell = b, \ b_{k \notin \{\ell-1,\ell\}} = b_{\mathrm{max}}\Big) - v_{\mathrm{NLL}}(\mathbf{b}_{\mathrm{max}}) - C_{\ell-1}(b') - C_\ell(b). \tag{81}$$

For analysis, define the general (not necessarily adjacent) pairwise interaction term for $1 \leq i < j \leq L$:

$$K_{i,j}(b_i, b_j) = v_{\mathrm{NLL}}\Big(b_i = b_i, \ b_j = b_j, \ b_{k \notin \{i,j\}} = b_{\mathrm{max}}\Big) - v_{\mathrm{NLL}}(\mathbf{b}_{\mathrm{max}}) - C_i(b_i) - C_j(b_j). \tag{82}$$

**Perturbation model around $\mathbf{b}_{\mathrm{max}}$.** Let $x_\ell$ denote the input to layer $\ell$ along the forward pass. Along the reference trajectory $\mathbf{b}_{\mathrm{max}}$:

$$x_{\ell+1}^0 = F_\ell(x_\ell^0), \qquad \ell = 1, \ldots, L-1. \tag{83}$$

Lowering the activation precision at layer $\ell$ from $b_{\mathrm{max}}$ to $b_\ell$ perturbs the layer mapping, modeled as:

$$x_{\ell+1} = F_\ell(x_\ell) + e_\ell(x_\ell; b_\ell), \qquad e_\ell(\cdot; b_{\mathrm{max}}) \equiv 0, \tag{84}$$

and define the deviation from the reference trajectory as $\delta_\ell := x_\ell - x_\ell^0$.

**Assumptions (effective stability and smoothness).** We state assumptions in a form that does not require every individual layer to be contractive; instead, we assume an effective bound on Jacobian products along the reference trajectory.

**(A1) Bounded Jacobian products.** Let $J_\ell(x) := \partial F_\ell(x)/\partial x$ and define the Jacobian product along the reference trajectory:

$$A_{p \to q} := J_{q-1}(x_{q-1}^0) \, J_{q-2}(x_{q-2}^0) \cdots J_p(x_p^0), \qquad 1 \leq p < q \leq L. \tag{85}$$

Assume there exists $\rho \in (0, 1)$ such that:

$$\|A_{p \to q}\| \leq \rho^{q-p} \quad \text{for all } 1 \leq p < q \leq L, \tag{86}$$

where $\| \cdot \|$ is the operator norm.

**(A2) Bounded quantization perturbation.** For any $b \in \mathcal{B}$ and for $x$ in the relevant neighborhood:

$$\|e_\ell(x; b)\| \leq \varepsilon_\ell(b). \tag{87}$$

**(A3) Second- and third-order smoothness of the loss.** Assume $\ell(x_L)$ is three-times differentiable in $x_L$, and there exist constants $M$ and $T$ such that:

$$\|\nabla^2 \ell(x_L)\| \leq M, \qquad \|\nabla^3 \ell(x_L)\| \leq T \tag{88}$$

throughout the same neighborhood.

**Lemma 1 (Propagation of perturbations).** Under Eq. 84 and Eq. 87, the deviation obeys:

$$\|\delta_{\ell+1}\| \leq \|J_\ell(\xi_\ell)\| \cdot \|\delta_\ell\| + \varepsilon_\ell(b_\ell), \tag{89}$$

for some $\xi_\ell$ between $x_\ell^0$ and $x_\ell$. In particular, if only layer $i$ is perturbed (all other layers at $b_{\mathrm{max}}$), then:

$$\|\delta_j^{(i)}\| \leq \|A_{i \to j}\| \cdot \varepsilon_i(b_i) \leq \rho^{j-i} \varepsilon_i(b_i). \tag{90}$$

**Lemma 2 (Decay of non-adjacent pairwise interactions).** Fix $1 \leq i < j \leq L$ and consider the four configurations in Eq. 82: the baseline $\mathbf{b}_{\max}$, the two single-layer perturbations at $i$ and $j$, and the joint perturbation at $(i,j)$. Let $\delta x_L^{(i)}$ and $\delta x_L^{(j)}$ denote the baseline-referenced changes in the final hidden state when perturbing only layer $i$ or only layer $j$, respectively. Under the linearization around the reference trajectory:

$$\delta x_L^{(i)} \approx A_{i \to L}\, e_i(x_i^0; b_i), \qquad \delta x_L^{(j)} \approx A_{j \to L}\, e_j(x_j^0; b_j). \tag{91}$$

The definition in Eq. 82 cancels the baseline and the two single-layer effects, leaving the cross term at second order. A third-order Taylor expansion of $\ell(\cdot)$ around $x_L^0$ yields:

$$K_{i,j}(b_i, b_j) = \mathbb{E}_u\left[(\delta x_L^{(i)})^\top \nabla^2 \ell(\tilde{x}_u)\,(\delta x_L^{(j)})\right] + R_{i,j}, \tag{92}$$

where $\tilde{x}_u$ lies between the reference and the joint-perturbed final state, and $R_{i,j}$ collects third-order (and higher) remainder terms.

Using Eq. 88 and Eq. 86 to bound the linearized perturbations, we have:

$$\left|K_{i,j}(b_i, b_j)\right| \leq \frac{M}{2}\, \|\delta x_L^{(i)}\|\, \|\delta x_L^{(j)}\| + \left|R_{i,j}\right|. \tag{93}$$

By splitting $A_{i \to L} = A_{j \to L} A_{i \to j}$:

$$\|\delta x_L^{(i)}\| \leq \|A_{j \to L}\|\, \|A_{i \to j}\|\, \varepsilon_i(b_i), \qquad \|\delta x_L^{(j)}\| \leq \|A_{j \to L}\|\, \varepsilon_j(b_j). \tag{94}$$

Substituting into Eq. 93 yields the distance-dependent upper bound:

$$\left|K_{i,j}(b_i, b_j)\right| \leq \frac{M}{2}\, \|A_{j \to L}\|^2\, \|A_{i \to j}\|\, \varepsilon_i(b_i)\, \varepsilon_j(b_j) + \left|R_{i,j}\right| \leq \frac{M}{2}\, \rho^{2(L-j)}\, \rho^{j-i}\, \varepsilon_i(b_i)\, \varepsilon_j(b_j) + \left|R_{i,j}\right|. \tag{95}$$

Moreover, the remainder can be explicitly controlled using $\|\nabla^3 \ell\| \leq T$:

$$\left|R_{i,j}\right| \leq \frac{T}{6}\, \mathbb{E}_u\left[\left(\|\delta x_L^{(i)}\| + \|\delta x_L^{(j)}\|\right)^3\right], \tag{96}$$

which is cubic in the perturbation magnitudes and negligible in the small-perturbation regime around $\mathbf{b}_{\max}$.

**Implication (adjacent interactions dominate).** Eq. 95 shows that under the effective stability condition in Eq. 86, the leading second-order interaction between layers $i$ and $j$ decays geometrically with their distance $(j - i)$ through the factor $\rho^{j-i}$, in addition to attenuation to the output through $\rho^{2(L-j)}$. Hence, adjacent pairs $(j = i + 1)$ capture the dominant second-order effects, while longer-range pairwise couplings are attenuated and can be approximated by accumulating local effects along the chain. This provides a theoretical basis for retaining only adjacent $K_{\ell-1,\ell}$ terms in the surrogate objective.

**Estimation and optimization implications.** **Estimating dense pairwise interactions is expensive and noisy.** With $|\mathcal{B}|$ candidate bitwidths, estimating adjacent pairwise tables requires $(L-1)|\mathcal{B}|^2$ interaction configurations, whereas estimating all pairs requires $\binom{L}{2}|\mathcal{B}|^2$ configurations, which is quadratic in $L$ and substantially increases calibration time and estimation variance under limited calibration data.

**Chain-structured DP is preserved by local factors, but not by dense pairwise graphs.** With adjacent pairwise interactions only, the energy has a chain form:

$$E(\mathbf{b}) = \sum_{\ell=1}^{L} C_\ell(b_\ell) + \sum_{\ell=2}^{L} K_{\ell-1,\ell}(b_{\ell-1}, b_\ell), \tag{97}$$

which supports efficient DP (including the budget dimension used in Appendix B.8). If one adds interactions for all layer pairs:

$$E_{\text{full}}(\mathbf{b}) = \sum_{\ell=1}^{L} C_\ell(b_\ell) + \sum_{1 \leq i < j \leq L} K_{i,j}(b_i, b_j), \tag{98}$$

the resulting interaction graph is dense and has high treewidth, so exact optimization no longer admits the linear-time chain DP structure. Finally, we note that incorporating higher-order adjacent interactions, such as triplets $T_{\ell-2,\ell-1,\ell}(b_{\ell-2}, b_{\ell-1}, b_\ell)$, still preserves a chain factorization and remains exactly solvable by DP after augmenting the state to retain the two most recent assignments, but it increases both calibration cost (from $|\mathcal{B}|^2$ to $|\mathcal{B}|^3$ per local factor) and solver complexity accordingly. For these reasons, we adopt adjacent pairwise interactions as a stable and computationally efficient surrogate.

## B.8. Dynamic Programming for Chain-Structured Mixed-Precision Allocation

This subsection presents a detailed derivation and implementation-ready dynamic programming (DP) solver for the chain-structured surrogate objective used in the main text. We restate the objective, show why it admits an exact DP solution, and then provide the state definition, initialization, recursion, optimality proof sketch, and backtracking procedure.

**Problem statement.** Consider a network with $L$ layers. Each layer $\ell$ is assigned an activation bitwidth $b_\ell \in \mathcal{B}$, where $\mathcal{B} = \{2, 4, 6, 8\}$, and the layer-wise configuration is $\mathbf{b} = (b_1, \ldots, b_L)$. Let $B$ be the global bit budget. Since we allocate bitwidths uniformly across layers (i.e., each layer has equal budget weight), the total budget consumption is

$$\sum_{\ell=1}^{L} b_\ell \leq B. \tag{99}$$

We minimize a chain-structured surrogate objective consisting of unary (per-layer) costs and pairwise (adjacent-layer) interaction costs:

$$\min_{\mathbf{b} \in \mathcal{B}^L} \sum_{\ell=1}^{L} C_\ell(b_\ell) + \sum_{\ell=2}^{L} K_{\ell-1,\ell}(b_{\ell-1}, b_\ell), \qquad \text{s.t.} \sum_{\ell=1}^{L} b_\ell \leq B. \tag{100}$$

**Why dynamic programming applies (chain optimal substructure).** The objective in Eq. 100 is a sum of local terms on a chain: each $b_\ell$ appears only in $C_\ell(b_\ell)$ and in at most two pairwise terms $K_{\ell-1,\ell}(b_{\ell-1}, b_\ell)$ and $K_{\ell,\ell+1}(b_\ell, b_{\ell+1})$. Therefore, if we fix the bitwidth at layer $\ell$ and fix the cumulative budget spent up to $\ell$, the remaining choices for layers $1{:}\ell - 1$ are independent of layers $\ell + 1{:}L$ except through $b_\ell$. This yields the standard optimal substructure property needed for exact DP.

**DP state definition.** For $\ell \in \{1, \ldots, L\}$, cumulative budget $c \in \{0, 1, \ldots, B\}$, and current-layer bitwidth $b \in \mathcal{B}$, define

$$\mathrm{DP}[\ell][c][b] = \min_{\substack{b_1, \ldots, b_{\ell-1} \in \mathcal{B} \\ \sum_{k=1}^{\ell} b_k = c, \ b_\ell = b}} \left( \sum_{k=1}^{\ell} C_k(b_k) + \sum_{k=2}^{\ell} K_{k-1,k}(b_{k-1}, b_k) \right), \tag{101}$$

i.e., the minimum surrogate objective value among all assignments up to layer $\ell$ that (i) spend exactly $c$ budget and (ii) assign bitwidth $b$ to layer $\ell$.

**Initialization.** At $\ell = 1$, there is no pairwise term. For each $b \in \mathcal{B}$, the budget is $c = b$ and

$$\mathrm{DP}[1][c][b] = C_1(b), \qquad c = b. \tag{102}$$

All other states are infeasible and are set to $+\infty$.

**Deriving the recursion.** For any $\ell \geq 2$, consider a feasible partial assignment ending at layer $\ell$ with $b_\ell = b$ and total budget $c$. Let the previous layer choose $b_{\ell-1} = b'$. Then the budget spent on layers $1{:}\ell - 1$ must be $c' = c - b$, and feasibility requires $c' \geq 0$. The surrogate objective decomposes as

$$\sum_{k=1}^{\ell} C_k(b_k) + \sum_{k=2}^{\ell} K_{k-1,k}(b_{k-1}, b_k)$$

$$= \underbrace{\left( \sum_{k=1}^{\ell-1} C_k(b_k) + \sum_{k=2}^{\ell-1} K_{k-1,k}(b_{k-1}, b_k) \right)}_{\text{objective on } 1:\ell-1} + C_\ell(b) + K_{\ell-1,\ell}(b', b). \tag{103}$$

The bracketed part is exactly the quantity minimized by $\mathrm{DP}[\ell - 1][c'][b']$ under the constraints $b_{\ell-1} = b'$ and budget $c'$. Hence, minimizing over all possible predecessor choices $b' \in \mathcal{B}$ yields

$$\mathrm{DP}[\ell][c][b] = C_\ell(b) + \min_{b' \in \mathcal{B}} \left\{ \mathrm{DP}[\ell - 1][c - b][b'] + K_{\ell-1,\ell}(b', b) \right\}, \tag{104}$$

subject to the feasibility constraint $c - b \geq 0$; if $c - b < 0$, we set $\mathrm{DP}[\ell][c][b] = +\infty$.

**Optimal value under a budget $B$.** The DP state uses an exact budget $c$. To enforce the global constraint $\sum_\ell b_\ell \le B$, we take the best value among all terminal states with $c \le B$:

$$\min_{c \le B} \min_{b \in \mathcal{B}} \mathrm{DP}[L][c][b]. \tag{105}$$

**Backtracking to recover $\{b_\ell^*\}$.** During the forward DP, we store the argmin predecessor

$$\mathrm{Prev}[\ell][c][b] \in \arg\min_{b' \in \mathcal{B}} \left\{ \mathrm{DP}[\ell-1][c-b][b'] + K_{\ell-1,\ell}(b',b) \right\}. \tag{106}$$

After selecting the terminal pair $(c^*, b_L^*)$ that attains Eq. 105, we recover the full assignment by iterating backward for $\ell = L, L-1, \dots, 2$:

$$b_{\ell-1}^* \leftarrow \mathrm{Prev}[\ell][c^*][b_\ell^*], \qquad c^* \leftarrow c^* - b_\ell^*. \tag{107}$$

This yields the optimal layer-wise bitwidth assignment $\{b_\ell^*\}_{\ell=1}^L$.

**Correctness (principle of optimality).** Eq. 104 follows from the optimal substructure of the chain objective: any optimal solution for layers $1{:}\ell$ ending at $(c,b)$ must contain an optimal solution for layers $1{:}\ell-1$ ending at $(c-b, b')$ for some predecessor $b'$, otherwise we could replace the prefix by a better one and strictly reduce the total objective, contradicting optimality. Thus the DP computes the exact optimum.

**Complexity.** Let $|\mathcal{B}|$ be the number of candidate bitwidths. The recursion in Eq. 104 evaluates a $\min$ over $|\mathcal{B}|$ predecessors for each $(\ell, c, b)$, giving time complexity $O\big(L \cdot B \cdot |\mathcal{B}|^2\big)$ and memory $O(L \cdot B \cdot |\mathcal{B}|)$. In practice, memory can be reduced to $O(B \cdot |\mathcal{B}|)$ by keeping only the previous layer's DP table, while storing backpointers separately (or recomputing them on demand).

## C. Additional Experimental Results

### C.1. More Detailed Results

In this appendix, we provide the full expanded results corresponding to Table 1 in the main paper.

**A16 (full-precision activations).** When activations remain in full precision (A16), the detailed breakdown in Table 4 shows that classical weight-only PTQ baselines such as GPTQ and QuaRot incur substantial average-accuracy losses across model families and scales, with particularly pronounced drops on reasoning-heavy benchmarks (e.g., ARC-C and HellaSwag). In contrast, TWLA consistently delivers the strongest ternary results at the lowest average weight precision (1.58 bits). For example, on LLaMA3-8B, TWLA improves Avg[7] from 39.04 (PT[2]-LLM) to 62.98, a +23.94-point gain (relative +61.3%), retaining 86.9% of FP16 performance. On Qwen3-14B, TWLA increases Avg[7] from 48.10 (SliM-LLM) to 68.48 (+20.38 points, +42.4%), reaching 94.5% of FP16 (72.47). On LLaMA2-70B, TWLA attains 73.60 Avg[7] versus 68.79 for QuaRot (+4.81 points, +7.0%), and remains only 3.11 points below FP16 (76.71), i.e., a 4.05% gap.

**A6 (6-bit activations).** Under 6-bit activations, Table 5 shows that performance gaps widen substantially: GPTQ/QuaRot often stay in low-accuracy regimes, and even stronger baselines (SliM-LLM and PB-LLM) exhibit clear degradation relative to A16. In contrast, TWLA remains the top performer across all evaluated models and is notably stable as activation precision is reduced. Concretely, on LLaMA2-13B, TWLA improves Avg[7] from 56.49 (ResQ) to 66.88 (+10.39 points, +18.4%). On Qwen3-32B, TWLA raises Avg[7] from 51.45 (PB-LLM) to 68.13 (+16.68 points, +32.4%). The collapse of conventional methods is particularly evident on LLaMA2-70B, where GPTQ attains 35.89 Avg[7] while TWLA reaches 73.45 (+37.56 points, +104.6%). Importantly, TWLA's own drop from A16 to A6 is minimal on large models (e.g., LLaMA2-70B: $73.60 \to 73.45$, only a 0.20% relative decrease).

**A4 (4-bit activations).** The A4 setting represents an extreme low-precision activation regime. As shown in Table 6, most baselines exhibit severe failure modes, with pronounced degradation on LAMBADA (LO/LS) and Avg[7] approaching chance-level on several models. By contrast, TWLA remains markedly resilient at A4 and preserves strong margins, especially at scale. For LLaMA2-70B, TWLA improves Avg[7] from 53.64 (QuaRot) to 71.10 (+17.46 points, +32.6%); on LAMBADA-standard, TWLA reaches 77.33 versus 21.50 for QuaRot (+55.83 points, +259.7%). On Qwen3-32B, TWLA increases Avg[7] from 26.36 (PB-LLM) to 65.25 (+38.89 points, +147.5%), and from 24.85 (GPTQ) to 65.25 (+40.40 points, +162.6%). Overall, these results highlight TWLA's strong robustness and high quantization quality under 4-bit activations.

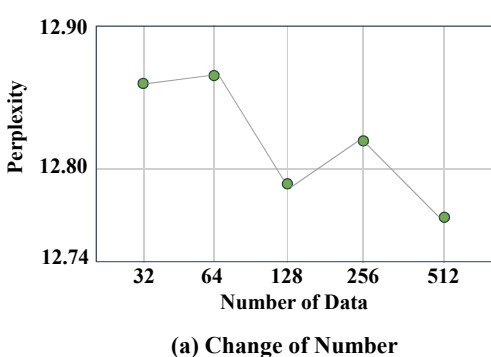 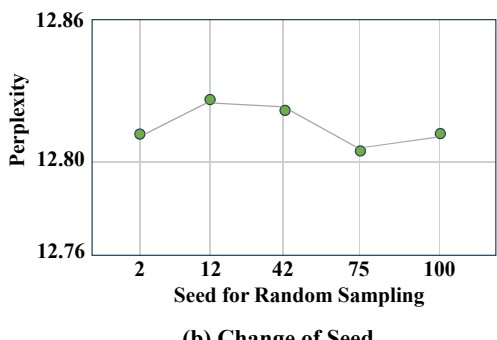

(a) Change of Number          (b) Change of Seed

*Figure 5.* Perplexity of LLaMA3-8B (W1.58A4) using calibration data sampled with different number or seeds from WikiText2.

**Summary.** Overall, the expanded tables consistently show that TWLA offers superior robustness and scalability as activation precision decreases from 16 to 6 and further to 4 bits. While existing PTQ methods rapidly lose accuracy (and often collapse) under low-bit activations, TWLA maintains strong and stable performance across model families and sizes. This stability is evident from the small performance drift as activations are reduced on large models; for example, on Qwen3-32B, Avg[7] changes only slightly from 69.54 (A16) to 68.13 (A6, $-2.03\%$) and remains high at 65.25 under A4 ($-4.23\%$ from A6; $-6.17\%$ from A16). These results support TWLA as a practical PTQ solution for extreme low-precision inference with ternary weights and low-bit activations.

### C.2. Ablation study on Calibration Data

We conducted an ablation on calibration data choice using three corpora (WikiText2, C4, and PTB) to examine how the calibration set affects TWLA quantization. As summarized in Table 7, calibrating on WikiText2 or C4 yields comparable Avg. accuracy across all tested models, whereas PTB consistently underperforms by a clear margin. One possible factor is the domain and diversity mismatch of PTB relative to the evaluation suites, which may yield less representative activation statistics; we leave a controlled domain-matching test for future work. In addition, calibrating on WikiText2 (resp. C4) improves perplexity on the same in-domain benchmark (WikiText2 / C4), highlighting a pronounced domain-matching advantage: the calibration corpus not only impacts downstream accuracy but also directly benefits the perplexity of the corresponding evaluation domain.

We further investigated the impact of calibration data on TWLA. Specifically, when binarizing LLaMA3-8B on WikiText-2, we fixed all quantization hyperparameters and varied the calibration set by (i) subsampling different numbers of calibration samples and (ii) resampling with multiple random seeds that control both sample selection and token order. For each setting, we re-quantized once and evaluated perplexity on a held-out split. Across all sizes and seeds, the resulting perplexity fluctuates by $<1\%$ relative to the mean. As shown in Fig. 5, the curves remain nearly flat as calibration size increases, and seed-wise traces largely overlap, indicating that even our smallest tested calibration subsets perform on par with larger ones. This stability suggests that TWLA does not rely on large-scale calibration corpora to obtain reliable activation statistics; instead, a small but representative subset is sufficient to reach near-saturated performance, which substantially reduces the calibration overhead in practical deployments. These observations confirm that TWLA is highly robust to calibration data selection.

### C.3. Ablation study on ILA-AMP

**Benefit–cost trade-off across interaction orders.** To characterize the calibration overhead induced by different interaction orders, we measure the cost by the wall-clock time required to collect validation NLL statistics. Let $|\mathcal{B}|$ denote the number of candidate bitwidths (with $|\mathcal{B}| = 4$ in our setting) and $L$ the number of layers. When interactions are restricted to local adjacent neighborhoods, the calibration cost (in terms of the number of NLL-evaluation configurations, which typically translates into wall-clock time) grows approximately as

$$\text{Cost}(k) \propto N_k, \qquad N_1 = L\,|\mathcal{B}|, \quad N_2 = (L-1)\,|\mathcal{B}|^2, \quad N_3 = (L-2)\,|\mathcal{B}|^3, \tag{108}$$

where $k \in \{1, 2, 3\}$ denotes the interaction order. The 1st-order variant uses only unary layer sensitivities (i.e., single-layer NLL perturbations) and thus implicitly assumes that quantization effects are independent across layers. The 2nd-order

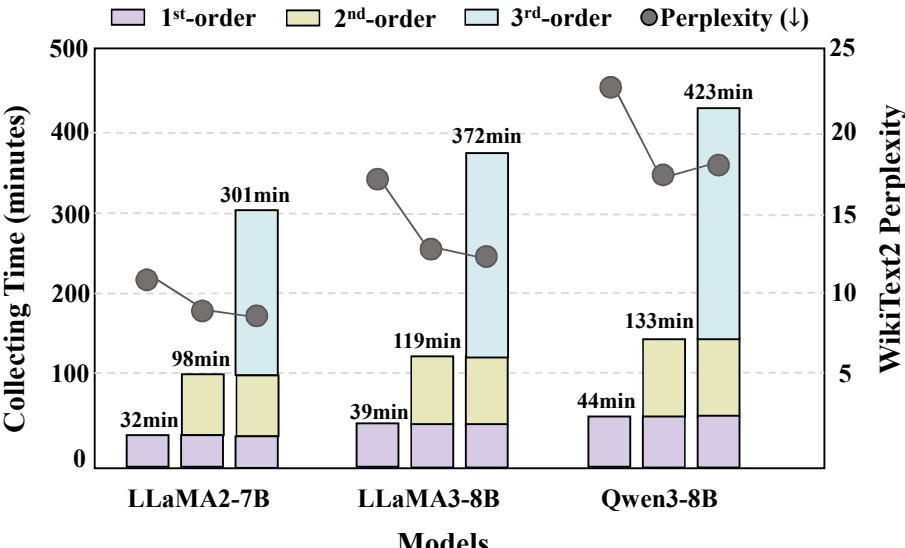

*Figure 6.* Calibration cost versus quantization quality under different interaction orders. Bars report the wall-clock time (minutes) spent collecting validation NLL statistics for 1st-, 2nd-, and 3rd-order interaction modeling, and dots report WikiText2 perplexity (lower is better).

variant (ILA-AMP) additionally models adjacent-layer pairwise couplings to capture error propagation across neighboring layers. The 3rd-order variant further incorporates adjacent triplet interactions. As shown in Fig. 6, we compare 1st-, 2nd-, and 3rd-order designs on three representative models, LLaMA2-7B, LLaMA3-8B, and QWEN3-8B, reporting both the NLL-collection time and the WikiText2 perplexity (lower is better). Moving from 1st-order to 2nd-order (ILA-AMP) yields a clear and consistent perplexity reduction at an acceptable increase in calibration time, indicating that adjacent pairwise couplings capture the dominant non-additive effects caused by cross-layer distribution shifts. In contrast, extending to 3rd-order incurs a substantial additional time overhead while providing little to no further perplexity improvement, and it may even slightly degrade perplexity. For example, on QWEN3-8B, upgrading from 2nd-order to 3rd-order increases the NLL-collection time by nearly 300 minutes (from 133 minutes to 423 minutes), yet the perplexity slightly increases. Overall, these results empirically justify our choice of a 2nd-order adjacent-interaction surrogate: it delivers most of the accuracy benefit with a significantly better cost–effectiveness profile than higher-order alternatives.

**Layer-wise activation bit allocation under a 4-bit budget.**   We visualize the layer-wise activation bit assignments produced by ILA-AMP under an average activation budget of 4 on the WikiText2 calibration set (as shown in Fig. 7). Both LLaMA2-7B and LLaMA3-8B exhibit a sparse adjustment pattern: most layers stay at 4-bit, while a few critical layers are assigned higher precision and a few less sensitive layers are reduced to 2-bit to satisfy the global budget. Concretely, LLaMA2-7B allocates a single early layer to 8-bit, keeps the vast majority of layers at 4-bit, and compensates with two late layers at 2-bit. LLaMA3-8B assigns 6-bit to the first three layers, maintains 4-bit for most layers, and uses three late 2-bit layers as budget offsets. This structure suggests that early layers are more influential under activation quantization: noise injected there perturbs the activation distribution that feeds all subsequent layers, and the resulting distribution shifts can accumulate and amplify along the stack. In contrast, some late layers are comparatively less sensitive because the remaining propagation path is shorter, leaving less room for amplification; thus, they can tolerate more aggressive quantization with limited accuracy loss. Overall, the observed "higher precision early, lower precision late, sparse deviations from 4-bit" allocation is consistent with ILA-AMP's adjacent-interaction modeling, which prioritizes precision where cross-layer error propagation is most consequential.

**Ablation on layer-sensitivity metrics.**   We further investigate how different layer-sensitivity metrics affect activation mixed-precision allocation under the same average activation-bit budget. Specifically, we compare four representative metrics against ILA-AMP on three models (LLaMA2-7B, LLaMA3-8B, and QWEN3-8B), using WikiText2 for calibration and evaluating quantization quality by WikiText2 perplexity (PPL; lower is better).

- **LIM** (Layer Input Modification) measures layer importance via the negative cosine similarity between the layer's input

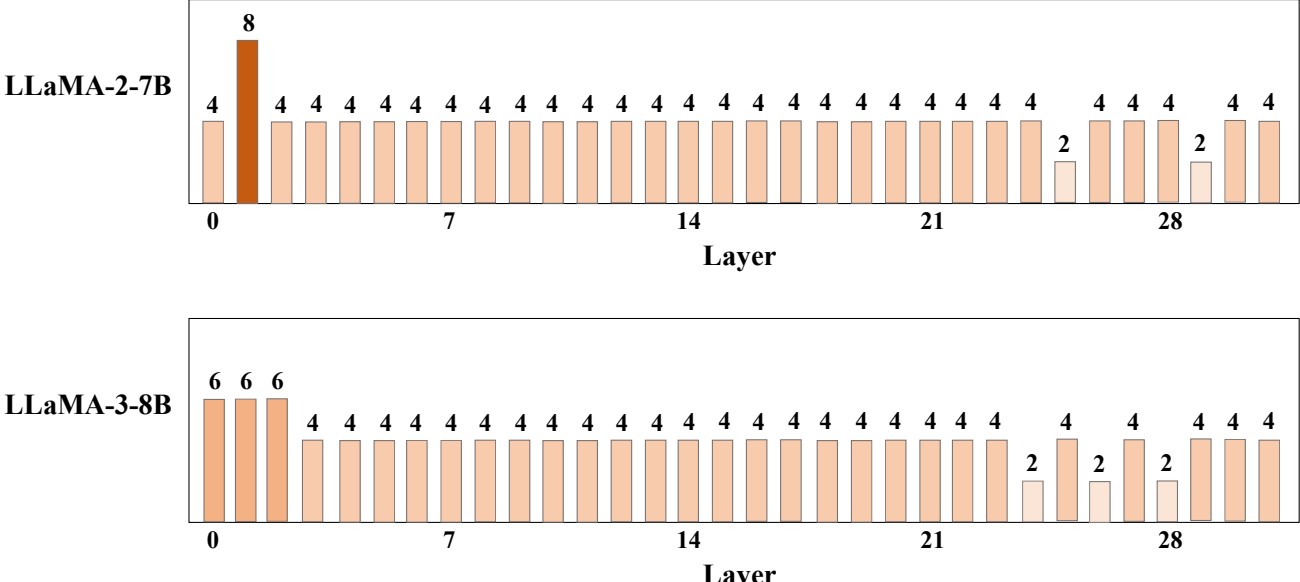

*Figure 7.* Layer-wise activation bit allocation produced by ILA-AMP under an average activation bit budget of 4 on the WikiText2 calibration set for LLaMA2-7B and LLaMA3-8B. Numbers above bars denote the assigned per-layer activation bit-width.

and output embeddings: larger input–output changes indicate higher importance, and the metric requires calibration data.

- **ZD** (Z-score Distribution) quantifies importance by the fraction of outlier weights in the target layer; it does not require calibration data, and a higher outlier ratio typically implies greater sensitivity.

- **Activation-based scoring** computes the Frobenius norm of layer activations, assuming that layers with larger activation energy process more critical information.

- **NLL Increase (NLL)**, which defines sensitivity as the increase in validation negative log-likelihood (NLL) when quantizing a single layer while keeping all other layers at a higher-precision reference, thereby directly reflecting per-layer NLL perturbations.

For each metric, we generate a layer-wise bit assignment under the same average bits(A)=4 constraint and report the resulting perplexity. As shown in Table 8, ILA-AMP consistently achieves the lowest perplexity across all models (8.31/12.83/16.15), substantially outperforming heuristic metrics. Compared to the strongest single-metric baseline NLI (which can be viewed as the first-order variant of ILA-AMP), ILA-AMP reduces PPL by 30.3% on LLAMA2-7B (11.92→8.31), 14.6% on LLAMA3-8B (15.03→12.83), and 25.9% on QWEN3-8B (21.79→16.15). These results indicate that under tight bit budgets, layer-wise heuristics alone are often insufficient, and explicitly modeling cross-layer error propagation and adjacent-layer coupling (ILA-AMP) is crucial for stable and high-quality mixed-precision allocation.

## D. Distribution Visualizations

In this section, we visualize the effect of KOTMS from two complementary perspectives: a single-layer mechanistic view and a cross-layer heterogeneity view. First, we focus on a representative layer of QWEN3-8B (Layer 12) and compare the statistical distributions of all weight matrices and the corresponding input activations before and after applying KOTMS within TWLA. This single-layer analysis is designed to provide an intuitive understanding of how KOTMS improves *distribution alignment* and *outlier suppression*. Second, we extend the analysis across layers by visualizing the pre-/post-KOTMS activation distributions at different layers of QWEN3-8B. This reveals that the quantizability gains are heterogeneous across layers, motivating the necessity of ILA-AMP to allocate activation precision adaptively rather than uniformly.

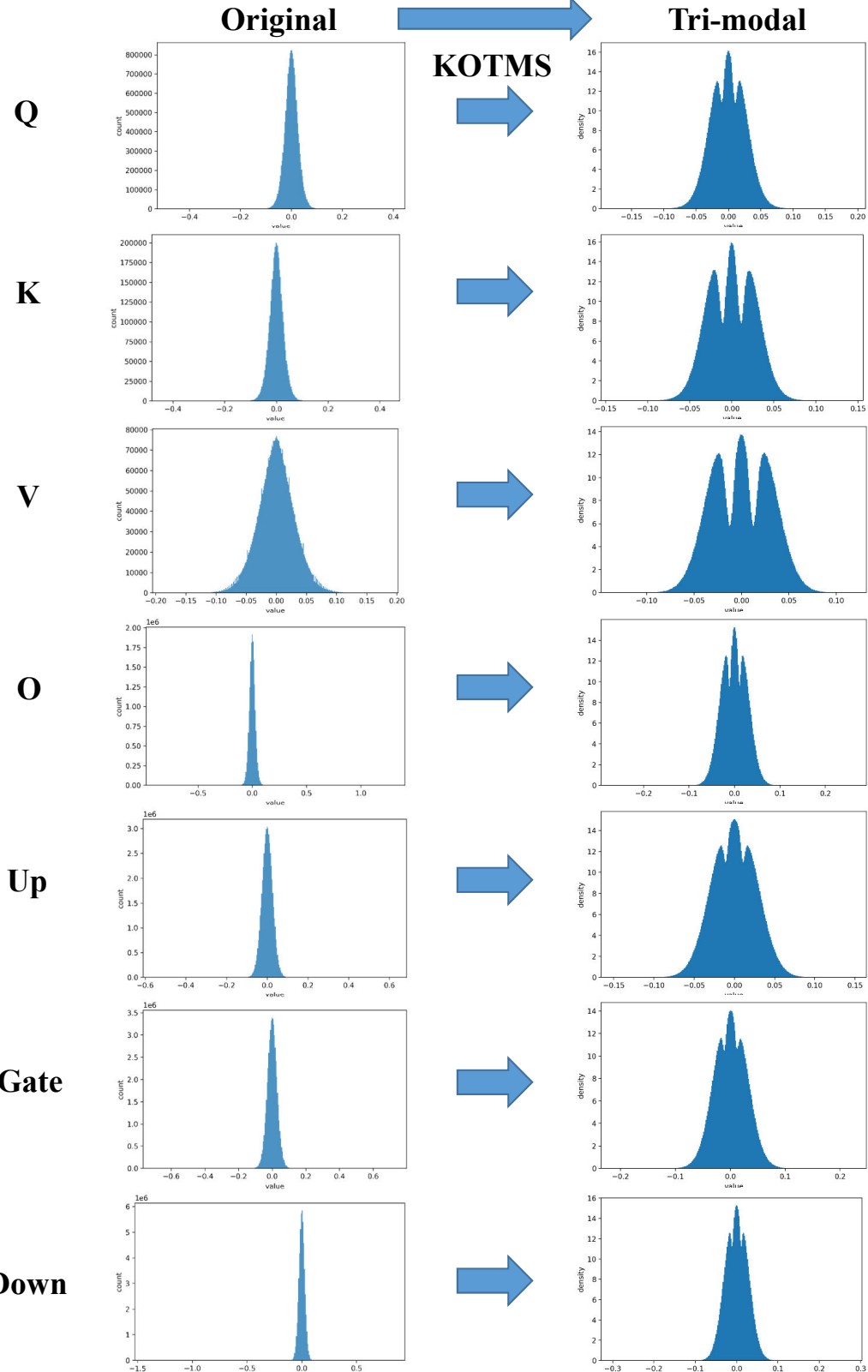

*Figure 8.* The weight distribution of the 12th layer in Qwen3-8B before and after TWLA.

## D.1. Single-layer visualization of KOTMS

Fig. 8 visualizes all weight matrices in Layer 12 of QWEN3-8B before and after KOTMS. Before applying KOTMS, the weights exhibit a Gaussian-like unimodal distribution, where most values concentrate near zero and decay smoothly toward both tails. Such a continuous unimodal shape is inherently misaligned with ternarization: when weights do not form separable clusters, mapping them to a discrete ternary codebook (e.g., $\{-1, 0, +1\}$) typically incurs substantial quantization error, especially under extreme low-bit regimes. Moreover, we observe pronounced outliers in several submodules (e.g., down-projection related matrices), manifested as heavy tails and a few extreme values. These outliers inflate the effective dynamic range and dominate the quantization scale, which compresses the resolution allocated to the bulk of weights and makes stable ternarization more difficult.

After applying KOTMS, the weight distributions change noticeably. The originally unimodal profiles are reshaped into a more structured multi-cluster pattern, which is qualitatively more compatible with the clustered structure desired by ternary quantization. More importantly, KOTMS substantially suppresses extreme outliers and tightens the tails, leading to a markedly reduced dynamic range. As a result, the quantization scale is less dominated by a small number of abnormal values, improving the statistical alignment between pretrained weights and the ternary codebook and providing a more stable foundation for high-accuracy ternarization.

A consistent trend is observed for the corresponding input activations. Prior to KOTMS, the activation distributions exhibit clear long tails and extreme values, which are known to trigger scale inflation and saturation under low-bit activation quantization, thereby amplifying quantization errors. After KOTMS projection, the activations become more concentrated with visibly tightened tails, and both the number and magnitude of extreme values are significantly reduced. This indicates that KOTMS not only reshapes weight statistics toward ternarization-friendly forms, but also improves the activation-side quantization conditions by mitigating outlier interference.

Fig. 9 further corroborates these observations from a quantile perspective. Before processing, the high-quantile region rises sharply, indicating extreme tail values that deviate from the main density mass and suggesting that the original activation statistics are unfavorable for low-bit quantization. After KOTMS, the high-quantile region drops substantially, the quantile curves become more concentrated and smoother, and the tails are noticeably thinner. In addition, variations across tokens become more continuous with far fewer localized anomalies. Together, these visualizations provide direct evidence that KOTMS reorganizes the activation geometry via orthogonal projection, suppresses outliers, and improves distribution alignment for low-bit quantization.

## D.2. Cross-layer heterogeneity in activation quantizability after KOTMS

To further reveal how KOTMS affects activation quantizability across layers, we visualize quantile plots of the Q-projection (**Q** matrix) input activations in QWEN3-8B at Layers 4, 12, 24, and 36, comparing the original activations against those after KOTMS projection (Fig. 10). Since KOTMS constructs an orthogonal auxiliary matrix through *weight-side* optimization, applying this orthogonal projection to activations effectively rotates and reorganizes the representation geometry, which can alleviate anisotropy and suppress extreme values, thereby mitigating scale inflation and saturation that typically hinder low-bit activation quantization. Importantly, the outlier-smoothing benefit is clearly *layer-dependent*, exhibiting pronounced cross-layer heterogeneity. As shown in Fig. 10, after KOTMS, Layer 4 still displays noticeably more high-quantile spikes and residual extreme fluctuations, whereas Layer 24 exhibits a substantially tighter quantile band with much weaker high-quantile excursions, indicating a significantly larger activation-side quantizability gain at Layer 24 than at Layer 4. This systematic heterogeneity implies that a uniform activation quantization setting across layers (e.g., the same bitwidth or the same quantization strength) is suboptimal: it either incurs excessive quantization error in layers where outliers remain prominent, or wastes precision budget in layers that have already become quantization-friendly after KOTMS. Therefore, these cross-layer quantile visualizations provide strong empirical evidence for the necessity of ILA-AMP, which allocates activation precision adaptively according to layer-wise quantization difficulty and marginal benefit, yielding a more robust and cost-effective end-to-end quantization outcome.

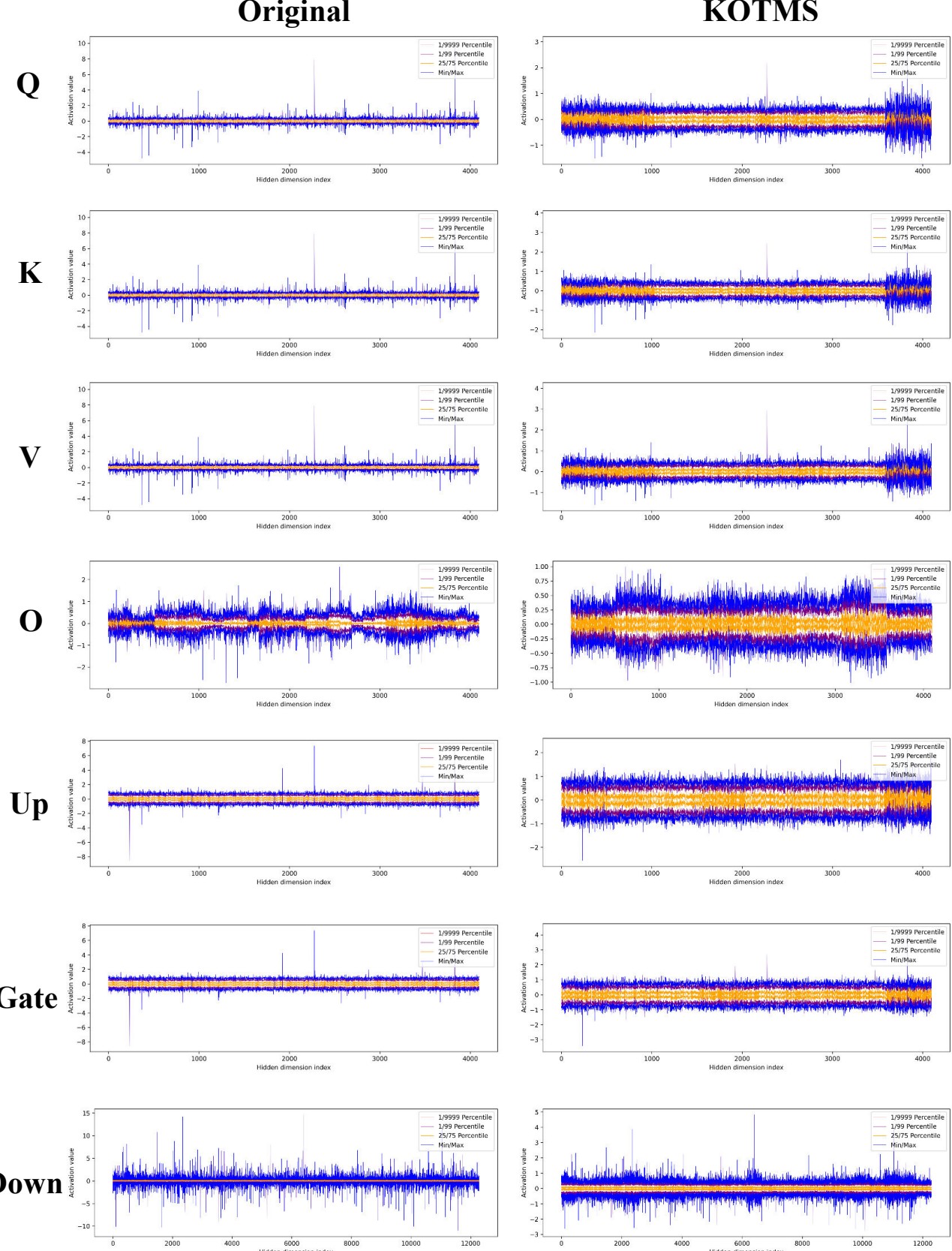

*Figure 9.* The activation distribution of the 12th layer in Qwen3-8B before and after TWLA.

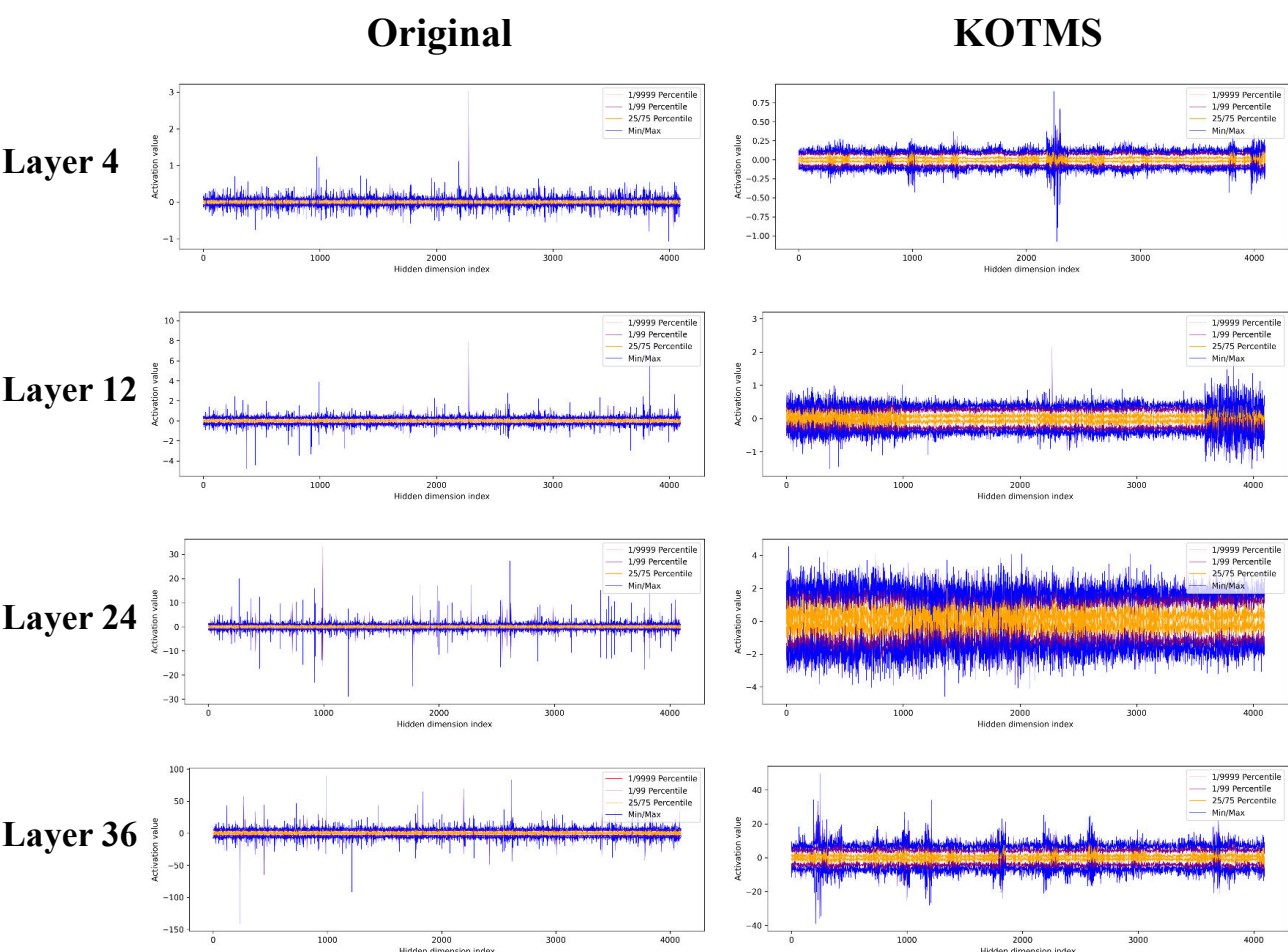

*Figure 10.* Cross-layer heterogeneity of activation outlier suppression after KOTMS on QWEN3-8B. Quantile plots (Min/Max, 1/99%, and 25/75%) of the **Q**-projection input activations are shown for Layers 4, 12, 24, and 36, comparing the original activations (left) with those after KOTMS projection (right).

*Table 4.* Zero-shot accuracy on Arc-Challenge (AC), Arc-Easy (AE), HellaSwag (HS), LAMBADA-openai (LO), LAMBADA-standard (LS), PIQA (PQ), and WinoGrande (WG) under a **16-bit activation** quantization setting.

| Model | Method | #Bits(W) | #Bits(A) | AE↑ | AC↑ | HS↑ | LO↑ | LS↑ | PQ↑ | WG↑ | Avg.[7]↑ |
|---|---|---|---|---|---|---|---|---|---|---|---|
| LLaMA2-7B | FP16 | 16 | | 74.66 | 46.25 | 75.96 | 73.45 | 68.19 | 78.73 | 69.22 | 69.49 |
| | GPTQ | 2 | | 32.45 | 22.61 | 30.53 | 7.72 | 4.08 | 55.44 | 50.83 | 29.09 |
| | QuaRot | 2 | | 40.95 | 24.83 | 35.77 | 23.87 | 16.94 | 58.60 | 53.12 | 36.30 |
| | SliM-LLM | 2MP | 16 | 50.44 | 25.59 | 52.21 | 52.33 | 39.77 | 62.45 | 59.95 | 48.96 |
| | PB-LLM | 1.7 | | 29.29 | 23.04 | 28.33 | 8.77 | 10.15 | 53.37 | 50.12 | 29.01 |
| | PT²-LLM | 1.58 | | 47.01 | 25.08 | 46.82 | 32.35 | 28.75 | 62.95 | 56.75 | 42.82 |
| | **TWLA** | 1.58 | | **65.87** | **37.63** | **65.75** | **70.33** | **60.94** | **74.65** | **65.19** | **62.91** |
| LLaMA2-13B | FP16 | 16 | | 77.57 | 49.15 | 79.39 | 76.71 | 70.06 | 80.47 | 71.98 | 72.19 |
| | GPTQ | 2 | | 26.39 | 27.39 | 28.15 | 3.98 | 4.68 | 50.05 | 49.80 | 27.21 |
| | QuaRot | 2 | | 51.98 | 29.61 | 49.63 | 51.25 | 38.68 | 65.34 | 58.64 | 49.30 |
| | SliM-LLM | 2MP | 16 | 56.71 | 30.46 | 51.76 | 55.17 | 43.21 | 66.87 | 59.83 | 52.00 |
| | PB-LLM | 1.7 | | 27.82 | 24.74 | 27.35 | 2.19 | 2.41 | 51.58 | 47.91 | 26.29 |
| | PT²-LLM | 1.58 | | 62.85 | 39.67 | 55.22 | 59.20 | 45.21 | 70.77 | 62.89 | 56.54 |
| | **TWLA** | 1.58 | | **70.24** | **42.41** | **72.59** | **76.40** | **66.82** | **76.71** | **68.75** | **67.70** |
| LLaMA2-70B | FP16 | 16 | | 81.02 | 57.34 | 83.78 | 79.57 | 74.67 | 82.70 | 77.90 | 76.71 |
| | GPTQ | 2 | | 49.83 | 29.10 | 44.20 | 44.21 | 31.48 | 63.71 | 56.59 | 45.59 |
| | QuaRot | 2 | | 75.50 | 45.60 | 65.20 | 77.30 | 70.60 | 76.00 | 71.30 | 68.79 |
| | SliM-LLM | 2MP | 16 | – | – | – | – | – | – | – | – |
| | PB-LLM | 1.7 | | 50.05 | 30.38 | 50.65 | 54.30 | 45.00 | 63.82 | 61.96 | 50.88 |
| | PT²-LLM | 1.58 | | 71.00 | 37.71 | 66.17 | 62.35 | 55.73 | 72.96 | 71.35 | 62.47 |
| | **TWLA** | 1.58 | | **80.73** | **53.75** | **71.87** | **78.40** | **73.67** | **79.40** | **77.35** | **73.60** |
| LLaMA3-8B | FP16 | 16 | | 77.90 | 52.82 | 79.07 | 75.63 | 68.58 | 80.63 | 72.93 | 72.51 |
| | GPTQ | 2 | | 28.28 | 23.29 | 28.46 | 2.17 | 0.60 | 52.07 | 49.80 | 26.38 |
| | QuaRot | 2 | | 40.87 | 24.15 | 36.34 | 17.50 | 14.85 | 60.34 | 55.56 | 35.66 |
| | SliM-LLM | 2MP | 16 | 31.82 | 21.58 | 29.42 | 3.15 | 5.51 | 52.50 | 48.46 | 27.49 |
| | PB-LLM | 1.7 | | 31.52 | 19.88 | 29.73 | 11.66 | 10.89 | 53.86 | 49.72 | 29.32 |
| | PT²-LLM | 1.58 | | 34.22 | 22.43 | 43.86 | 35.95 | 26.70 | 56.86 | 53.28 | 39.04 |
| | **TWLA** | 1.58 | | **67.30** | **39.42** | **66.90** | **64.93** | **59.21** | **74.76** | **68.35** | **62.98** |
| Qwen3-8B | FP16 | 16 | | 80.93 | 56.74 | 74.98 | 64.18 | 61.11 | 77.37 | 68.35 | 69.09 |
| | GPTQ | 2 | | 30.35 | 21.93 | 31.46 | 10.79 | 7.51 | 52.99 | 49.72 | 29.25 |
| | QuaRot | 2 | | – | – | – | – | – | – | – | – |
| | SliM-LLM | 2MP | 16 | 42.45 | 28.92 | 42.40 | 23.00 | 20.85 | 60.94 | 55.33 | 39.13 |
| | PB-LLM | 1.7 | | 41.08 | 25.09 | 38.15 | 28.53 | 24.57 | 59.25 | 52.33 | 38.43 |
| | PT²-LLM | 1.58 | | – | – | – | – | – | – | – | – |
| | **TWLA** | 1.58 | | **71.38** | **45.99** | **63.09** | **60.29** | **55.13** | **73.78** | **64.72** | **62.05** |
| Qwen3-14B | FP16 | 16 | | 83.08 | 60.49 | 78.82 | 67.84 | 64.47 | 79.76 | 72.85 | 72.47 |
| | GPTQ | 2 | | 32.49 | 22.35 | 29.01 | 7.37 | 6.31 | 54.03 | 48.70 | 28.61 |
| | QuaRot | 2 | | – | – | – | – | – | – | – | – |
| | SliM-LLM | 2MP | 16 | 52.70 | 33.96 | 53.75 | 35.10 | 34.80 | 66.65 | 59.75 | 48.10 |
| | PB-LLM | 1.7 | | 46.21 | 29.78 | 43.15 | 40.07 | 34.87 | 62.24 | 57.30 | 44.80 |
| | PT²-LLM | 1.58 | | 53.03 | 25.63 | 50.65 | 37.22 | 35.19 | 62.95 | 59.75 | 46.35 |
| | **TWLA** | 1.58 | | **76.26** | **52.30** | **70.98** | **69.05** | **63.95** | **76.55** | **70.64** | **68.48** |
| Qwen3-32B | FP16 | 16 | | 83.21 | 61.09 | 82.60 | 67.24 | 58.04 | 81.99 | 72.77 | 72.13 |
| | GPTQ | 2 | | 38.72 | 30.63 | 45.48 | 22.22 | 17.87 | 59.58 | 52.80 | 38.18 |
| | QuaRot | 2 | | – | – | – | – | – | – | – | – |
| | SliM-LLM | 2MP | 16 | 74.03 | 52.21 | 68.98 | 52.62 | 55.80 | 78.99 | 70.88 | 64.79 |
| | PB-LLM | 1.7 | | 71.04 | 48.72 | 63.18 | 65.94 | 57.38 | 72.14 | 69.06 | 63.92 |
| | PT²-LLM | 1.58 | | – | – | – | – | – | – | – | – |
| | **TWLA** | 1.58 | | **79.53** | **55.33** | **76.89** | **67.41** | **57.33** | **78.99** | **71.27** | **69.54** |

*Table 5.* Zero-shot accuracy on Arc-Challenge (AC), Arc-Easy (AE), HellaSwag (HS), LAMBADA-openai (LO), LAMBADA-standard (LS), PIQA (PQ), and WinoGrande (WG) under a **6-bit activation** quantization setting.

| Model | Method | #Bits(W) | #Bits(A) | AE↑ | AC↑ | HS↑ | LO↑ | LS↑ | PQ↑ | WG↑ | Avg.[7]↑ |
|---|---|---|---|---|---|---|---|---|---|---|---|
| | GPTQ | 2 | 6 | 29.34 | 23.46 | 29.22 | 7.06 | 3.16 | 52.45 | 50.91 | 27.94 |
| | QuaRot | 2 | 6 | 40.70 | 23.72 | 35.21 | 21.41 | 15.58 | 58.00 | 51.93 | 35.22 |
| | ResQ | 2.3MP | 6.1MP | 55.70 | 29.52 | 52.30 | 53.55 | 39.85 | 67.90 | 59.43 | 51.18 |
| LLaMA2-7B | SliM-LLM | 2MP | 6 | 38.75 | 24.15 | 42.60 | 25.15 | 17.30 | 58.05 | 53.43 | 37.06 |
| | PB-LLM | 1.7 | 6 | 30.35 | 21.67 | 33.75 | 11.70 | 12.15 | 52.56 | 52.01 | 30.60 |
| | PT$^2$-LLM | 1.58 | 6 | 44.82 | 25.79 | 42.97 | 22.07 | 17.98 | 60.97 | 55.67 | 38.61 |
| | **TWLA** | **1.58** | **6MP** | **64.56** | **36.69** | **64.92** | **70.13** | **59.23** | **74.70** | **63.61** | **61.98** |
| | GPTQ | 2 | 6 | 26.39 | 28.16 | 26.69 | 0.17 | 0.58 | 50.92 | 48.38 | 25.90 |
| | QuaRot | 2 | 6 | 49.45 | 27.82 | 48.74 | 49.56 | 37.18 | 65.29 | 57.62 | 47.95 |
| | ResQ | 2.3MP | 6.1MP | 60.40 | 34.64 | 56.75 | 60.45 | 48.55 | 71.11 | 63.54 | 56.49 |
| LLaMA2-13B | SliM-LLM | 2MP | 6 | 48.20 | 27.30 | 48.50 | 31.65 | 29.20 | 63.38 | 55.01 | 43.32 |
| | PB-LLM | 1.7 | 6 | 28.05 | 24.06 | 28.10 | 0.70 | 0.55 | 51.90 | 52.88 | 26.61 |
| | PT$^2$-LLM | 1.58 | 6 | 58.40 | 36.23 | 52.58 | 44.94 | 37.30 | 67.31 | 59.01 | 51.82 |
| | **TWLA** | **1.58** | **6MP** | **69.23** | **42.49** | **71.02** | **75.98** | **65.28** | **76.28** | **67.88** | **66.88** |
| | GPTQ | 2 | 6 | 39.81 | 25.43 | 33.98 | 28.31 | 18.36 | 58.22 | 47.12 | 35.89 |
| | QuaRot | 2 | 6 | 74.10 | 45.00 | 64.70 | 76.20 | 68.50 | 75.60 | 71.70 | 67.97 |
| | ResQ | 2.3MP | 6.1MP | 37.90 | 32.60 | 64.80 | 75.70 | 68.10 | 53.50 | 72.10 | 57.81 |
| LLaMA2-70B | SliM-LLM | 2MP | 6 | – | – | – | – | – | – | – | – |
| | PB-LLM | 1.7 | 6 | 39.10 | 26.79 | 42.15 | 38.95 | 29.70 | 55.71 | 53.83 | 40.89 |
| | PT$^2$-LLM | 1.58 | 6 | 58.70 | 32.96 | 51.75 | 38.10 | 38.80 | 64.65 | 57.75 | 48.96 |
| | **TWLA** | **1.58** | **6MP** | **80.07** | **53.50** | **71.60** | **77.60** | **73.60** | **79.93** | **77.82** | **73.45** |
| | GPTQ | 2 | 6 | 25.84 | 26.11 | 27.17 | 0.54 | 0.10 | 49.56 | 50.51 | 25.69 |
| | QuaRot | 2 | 6 | 41.31 | 24.57 | 35.50 | 18.20 | 12.15 | 58.16 | 58.67 | 35.51 |
| | ResQ | 2.3MP | 6.1MP | 51.00 | 29.52 | 50.75 | 39.90 | 37.20 | 64.85 | 59.75 | 47.57 |
| LLaMA3-8B | SliM-LLM | 2MP | 6 | 29.15 | 23.72 | 31.50 | 2.15 | 2.30 | 51.36 | 49.88 | 27.15 |
| | PB-LLM | 1.7 | 6 | 32.10 | 19.54 | 35.65 | 12.25 | 10.00 | 52.99 | 49.41 | 30.28 |
| | PT$^2$-LLM | 1.58 | 6 | 29.93 | 21.79 | 35.79 | 14.22 | 12.90 | 52.99 | 48.02 | 30.81 |
| | **TWLA** | **1.58** | **6MP** | **65.15** | **35.86** | **65.84** | **63.87** | **58.20** | **73.83** | **67.32** | **61.44** |
| | GPTQ | 2 | 6 | 26.81 | 24.06 | 28.03 | 2.87 | 1.94 | 51.52 | 51.30 | 26.65 |
| | QuaRot | 2 | 6 | – | – | – | – | – | – | – | – |
| | ResQ | 2.3MP | 6.1MP | – | – | – | – | – | – | – | – |
| Qwen3-8B | SliM-LLM | 2MP | 6 | 33.90 | 25.00 | 36.15 | 12.15 | 9.70 | 54.95 | 47.99 | 31.41 |
| | PB-LLM | 1.7 | 6 | 30.90 | 22.70 | 36.75 | 14.90 | 13.90 | 53.92 | 48.62 | 31.67 |
| | PT$^2$-LLM | 1.58 | 6 | – | – | – | – | – | – | – | – |
| | **TWLA** | **1.58** | **6MP** | **69.88** | **45.27** | **56.85** | **54.97** | **51.73** | **72.77** | **61.73** | **59.03** |
| | GPTQ | 2 | 6 | 29.50 | 22.27 | 27.56 | 4.62 | 4.09 | 51.14 | 48.15 | 26.76 |
| | QuaRot | 2 | 6 | – | – | – | – | – | – | – | – |
| | ResQ | 2.3MP | 6.1MP | – | – | – | – | – | – | – | – |
| Qwen3-14B | SliM-LLM | 2MP | 6 | 44.25 | 28.75 | 44.70 | 15.15 | 14.10 | 60.12 | 55.17 | 37.46 |
| | PB-LLM | 1.7 | 6 | 40.80 | 24.74 | 42.05 | 23.25 | 19.40 | 58.32 | 52.41 | 37.28 |
| | PT$^2$-LLM | 1.58 | 6 | 42.21 | 27.72 | 43.20 | 14.33 | 12.12 | 58.13 | 54.19 | 36.00 |
| | **TWLA** | **1.58** | **6MP** | **74.41** | **50.85** | **69.89** | **67.88** | **62.25** | **75.57** | **70.17** | **67.29** |
| | GPTQ | 2 | 6 | 32.28 | 26.71 | 36.34 | 10.64 | 8.67 | 54.41 | 49.01 | 31.15 |
| | QuaRot | 2 | 6 | – | – | – | – | – | – | – | – |
| | ResQ | 2.3MP | 6.1MP | – | – | – | – | – | – | – | – |
| Qwen3-32B | SliM-LLM | 2MP | 6 | 61.70 | 39.93 | 51.40 | 37.45 | 40.90 | 65.89 | 62.90 | 52.43 |
| | PB-LLM | 1.7 | 6 | 61.70 | 39.93 | 51.40 | 37.45 | 40.90 | 65.89 | 62.90 | 51.45 |
| | PT$^2$-LLM | 1.58 | 6 | – | – | – | – | – | – | – | – |
| | **TWLA** | **1.58** | **6MP** | **77.49** | **53.18** | **75.33** | **66.33** | **56.20** | **77.27** | **71.11** | **68.13** |

*Table 6.* Zero-shot accuracy on Arc-Challenge (AC), Arc-Easy (AE), HellaSwag (HS), LAMBADA-openai (LO), LAMBADA-standard (LS), PIQA (PQ), and WinoGrande (WG) under a **4-bit activation** quantization setting.

| Model | Method | #Bits(W) | #Bits(A) | AE↑ | AC↑ | HS↑ | LO↑ | LS↑ | PQ↑ | WG↑ | Avg.[7]↑ |
|---|---|---|---|---|---|---|---|---|---|---|---|
| LLaMA2-7B | GPTQ | 2 | 4 | 26.26 | 27.73 | 26.35 | 0.00 | 0.00 | 49.89 | 50.36 | 25.80 |
| | QuaRot | 2 | 4 | 35.02 | 22.87 | 31.34 | 10.27 | 8.48 | 56.75 | 51.07 | 30.83 |
| | ResQ | 2.3MP | 4.2MP | 46.65 | 28.50 | 48.05 | 45.30 | 32.10 | 63.32 | 52.72 | 45.23 |
| | SliM-LLM | 2MP | 4 | 26.80 | 25.43 | 28.10 | 0.15 | 0.15 | 49.24 | 49.80 | 25.67 |
| | PB-LLM | 1.7 | 4 | 28.20 | 25.26 | 29.15 | 0.70 | 0.35 | 50.11 | 51.46 | 26.46 |
| | PT$^2$-LLM | 1.58 | 4 | 29.20 | 26.21 | 30.22 | 0.90 | 0.65 | 51.23 | 52.77 | 27.31 |
| | **TWLA** | 1.58 | 4MP | **58.88** | **34.73** | **60.81** | **64.45** | **53.76** | **71.27** | **62.12** | **58.00** |
| LLaMA2-13B | GPTQ | 2 | 4 | 26.39 | 27.82 | 25.93 | 0.00 | 0.00 | 48.53 | 48.78 | 25.35 |
| | QuaRot | 2 | 4 | 39.56 | 25.77 | 39.94 | 32.08 | 23.37 | 60.12 | 55.09 | 39.42 |
| | ResQ | 2.3MP | 4.2MP | 56.75 | 32.94 | 54.80 | 50.10 | 38.70 | 68.66 | 60.93 | 51.84 |
| | SliM-LLM | 2MP | 4 | 27.15 | 25.17 | 30.45 | 0.40 | 0.00 | 50.87 | 51.07 | 26.44 |
| | PB-LLM | 1.7 | 4 | 26.10 | 26.54 | 26.95 | 0.00 | 0.00 | 48.69 | 50.51 | 25.54 |
| | PT$^2$-LLM | 1.58 | 4 | 27.10 | 26.54 | 28.95 | 0.00 | 0.00 | 49.69 | 51.22 | 26.19 |
| | **TWLA** | 1.58 | 4MP | **66.58** | **39.85** | **68.20** | **73.24** | **60.82** | **74.27** | **67.17** | **64.30** |
| LLaMA2-70B | GPTQ | 2 | 4 | 26.39 | 26.88 | 26.14 | 0.00 | 0.00 | 49.46 | 52.57 | 25.92 |
| | QuaRot | 2 | 4 | 63.50 | 34.20 | 55.20 | 57.50 | 35.20 | 68.60 | 61.30 | 53.64 |
| | ResQ | 2.3MP | 4.2MP | 42.60 | 34.60 | 62.40 | 70.00 | 60.90 | 55.10 | 69.40 | 56.43 |
| | SliM-LLM | 2MP | 4 | – | – | – | – | – | – | – | – |
| | PB-LLM | 1.7 | 4 | 26.95 | 24.40 | 29.50 | 0.55 | 0.45 | 50.44 | 50.75 | 26.18 |
| | PT$^2$-LLM | 1.58 | 4 | 26.95 | 25.40 | 29.59 | 0.35 | 0.15 | 50.22 | 50.66 | 26.19 |
| | **TWLA** | 1.58 | 4MP | **78.60** | **51.88** | **70.00** | **75.93** | **69.07** | **77.33** | **74.90** | **71.10** |
| LLaMA3-8B | GPTQ | 2 | 4 | 26.56 | 25.00 | 26.46 | 0.00 | 0.00 | 50.38 | 51.07 | 25.64 |
| | QuaRot | 2 | 4 | 31.61 | 21.42 | 29.26 | 4.81 | 4.97 | 53.43 | 48.30 | 27.69 |
| | ResQ | 2.3MP | 4.2MP | 42.90 | 26.96 | 43.85 | 23.35 | 21.75 | 59.64 | 55.38 | 39.12 |
| | SliM-LLM | 2MP | 4 | 27.15 | 25.68 | 28.05 | 0.05 | 0.00 | 48.69 | 47.99 | 25.37 |
| | PB-LLM | 1.7 | 4 | 29.70 | 22.78 | 30.30 | 0.85 | 0.50 | 51.03 | 48.86 | 26.29 |
| | PT$^2$-LLM | 1.58 | 4 | 28.71 | 22.02 | 30.10 | 0.75 | 0.30 | 50.83 | 47.99 | 25.81 |
| | **TWLA** | 1.58 | 4MP | **60.23** | **34.56** | **59.83** | **51.87** | **45.47** | **71.38** | **63.30** | **55.23** |
| Qwen3-8B | GPTQ | 2 | 4 | 25.17 | 26.37 | 25.47 | 0.00 | 0.00 | 50.38 | 49.64 | 25.29 |
| | QuaRot | 2 | 4 | – | – | – | – | – | – | – | – |
| | ResQ | 2.3MP | 4.2MP | – | – | – | – | – | – | – | – |
| | SliM-LLM | 2MP | 4 | 26.35 | 26.28 | 27.10 | 0.00 | 0.00 | 48.75 | 50.75 | 25.60 |
| | PB-LLM | 1.7 | 4 | 28.85 | 23.81 | 28.65 | 0.15 | 0.05 | 52.18 | 50.12 | 26.26 |
| | PT$^2$-LLM | 1.58 | 4 | – | – | – | – | – | – | – | – |
| | **TWLA** | 1.58 | 4MP | **57.19** | **35.83** | **51.82** | **40.49** | **39.97** | **68.22** | **59.45** | **50.42** |
| Qwen3-14B | GPTQ | 2 | 4 | 26.60 | 25.94 | 25.74 | 0.00 | 0.00 | 52.07 | 50.20 | 25.79 |
| | QuaRot | 2 | 4 | – | – | – | – | – | – | – | – |
| | ResQ | 2.3MP | 4.2MP | – | – | – | – | – | – | – | – |
| | SliM-LLM | 2MP | 4 | 26.35 | 25.77 | 27.40 | 0.00 | 0.00 | 50.27 | 49.57 | 25.62 |
| | PB-LLM | 1.7 | 4 | 28.05 | 23.63 | 31.90 | 0.80 | 0.65 | 50.60 | 49.01 | 29.23 |
| | PT$^2$-LLM | 1.58 | 4 | – | – | – | – | – | – | – | – |
| | **TWLA** | 1.58 | 4MP | **68.39** | **45.05** | **65.29** | **61.48** | **57.17** | **72.91** | **63.61** | **62.00** |
| Qwen3-32B | GPTQ | 2 | 4 | 24.62 | 25.60 | 26.13 | 0.00 | 0.00 | 49.29 | 48.30 | 24.85 |
| | QuaRot | 2 | 4 | – | – | – | – | – | – | – | – |
| | ResQ | 2.3MP | 4.2MP | – | – | – | – | – | – | – | – |
| | SliM-LLM | 2MP | 4 | 26.10 | 24.77 | 29.80 | 0.85 | 0.55 | 50.89 | 51.22 | 26.31 |
| | PB-LLM | 1.7 | 4 | 27.10 | 25.77 | 28.80 | 0.30 | 0.40 | 51.09 | 51.07 | 26.36 |
| | PT$^2$-LLM | 1.58 | 4 | – | – | – | – | – | – | – | – |
| | **TWLA** | 1.58 | 4MP | **73.34** | **51.83** | **67.80** | **65.90** | **55.50** | **74.29** | **68.10** | **65.25** |

*Table 7.* Ablation study on calibration set type.

| Model | Calibration Data Type | Wikitext2↓ | C4↓ | PTB↓ | Avg.[7]↑ |
|---|---|---|---|---|---|
| | Wikitext2 | 8.31 | 12.47 | 58.12 | 58.00 |
| LLaMA-2-7B | C4 | 11.76 | 10.77 | 60.10 | 57.39 |
| | PTB | 13.10 | 19.29 | 54.70 | 56.09 |
| | Wikitext2 | 12.83 | 19.31 | 21.93 | 55.23 |
| LLaMA-3-8B | C4 | 16.37 | 15.57 | 21.78 | 54.96 |
| | PTB | 16.59 | 16.21 | 18.49 | 53.39 |
| | Wikitext2 | 16.26 | 22.72 | 28.65 | 50.42 |
| Qwen3-8B | C4 | 21.37 | 17.33 | 29.14 | 49.16 |
| | PTB | 21.95 | 23.08 | 23.37 | 47.73 |

*Table 8.* WikiText2 perplexity under activation mixed precision with average bits(A)=4, using different layer-sensitivity metrics.

| Model | Avg. Bits(A) | MP metric (PPL↓) | | | | |
|---|---|---|---|---|---|---|
| | | LIM | ZD | Act. | NLL | ILA-AMP |
| LLaMA-2-7B | 4 | 17.49 | 18.01 | 16.33 | 11.92 | **8.31** |
| LLaMA-3-8B | 4 | 23.21 | 23.37 | 20.05 | 15.03 | **12.83** |
| Qwen3-8B | 4 | 33.17 | 29.76 | 24.64 | 21.79 | **16.15** |

