# OpenReview forum: "TWLA: Achieving Ternary Weights and Low-Bit Activations for LLMs via Post-Training Quantization"
_ICML.cc/2026/Conference — ICML 2026 regular_

### Official Review · Reviewer_spM3 · 2026-03-04

**Soundness:** 3
**Presentation:** 3
**Significance:** 3
**Originality:** 3
**Overall Recommendation:** 4
**Confidence:** 4

**Summary:**

This paper presents a post-training quantization framework that achieves 1.58-bit weight compression and 4-bit activation quantization. TWLA optimizes weight ternarization, reshapes weights, and improves activation quantization, delivering significant inference acceleration while maintaining high accuracy.

**Compliance With Llm Reviewing Policy:**

Affirmed.

**Final Justification:**

I have read the author’s response as well as the rebuttals from the other reviewers. The author has adequately addressed my concerns, therefore, I maintain the positive score.

**Key Questions For Authors:**

1. How do you justify the additional computational overhead introduced by the explicit compensation methods?

2. How does your method handle cases where activation distributions are highly skewed or non-Gaussian? Have you considered alternative techniques to better suppress outliers in such scenarios?

3. Although the method is designed to be training-free, could it be integrated with end-to-end training optimization to further improve performance?

**Limitations:**

Yes.

**Strengths And Weaknesses:**

[Strengths]
1. This work has a strong motivation, aiming to address the high memory and compute costs of LLMs by utilizing ternary weights and low-bit activations.
2. The proposed framework introduces three key components that collectively optimize, and the technical contribution is good.
3. This experiments show great performance of the methd.

[Weaknesses]
1. The explicit compensation method introduces additional computational complexity. And this overhead has not been adequately assessed through ablation studies.
2. If the activation distributions are highly skewed or non-Gaussian, the reshaping process might not adequately suppress outliers.
3. The method relies on heuristic optimization and a fixed bit-width allocation for certain layers, which might not be effective in certain cases.

---

> ### Author Rebuttal · Authors · 2026-03-27
>
> Thank you for the constructive feedback. Below are our responses.
>
> ### Q1
> How do you justify the additional computational overhead introduced by the explicit compensation methods?
>
> ### A1
> We agree that overhead control is central. KOTMS is justified by lower theoretical complexity, small overhead, and accuracy gains.
>
> First KOTMS replaces dense orthogonal matrices with a Kronecker parameterization. For hidden size $m=4096$, dense multiplication requires about $m^2\approx16.7$M MACs and the same number of parameters. With $R=R_1\otimes R_2$ and $n_1=n_2=64$, the parameter count becomes $n_1^2+n_2^2=8192$, i.e., 2048× smaller. Using the reshape-multiply-vectorize implementation, the cost becomes $\mathcal{O}(m(n_1+n_2))=4096\times128=524,288$ MACs, about 32× lower than dense multiplication.
>
> Second, micro-benchmarks on a **4096×4096** linear layer show that actual overhead is low:
>
> |Layer|BS|FP16|INT4|INT4+KOTMS|
> |:--|:--|:--|:--|:--|
> |4096×4096|1|1.024ms|0.363ms|0.388ms|
> ||4|3.688ms|1.310ms|1.413ms|
> ||8|7.129ms|2.598ms|2.732ms|
>
> Compared with pure INT4, KOTMS adds only about 7% runtime, while the full system remains close to 3× faster than FP16.
>
> Third, this small cost yields large returns. In Table 3, for Qwen-14B under W1.58A4, removing KOTMS drops MMLU to 25.33, while adding it restores MMLU to 60.82. Thus, KOTMS is a structured module whose overhead stays small while delivering its largest benefit in the hardest W1.58A4 regime. We will add the complexity example and speed table in appendix.
>
> ### Q2
> How does your method handle cases where activation distributions are highly skewed or non-Gaussian? Have you considered alternative techniques to better suppress outliers in such scenarios?
>
> ### A2
> How our method handles non-Gaussian skewed distributions: Our framework inherently suppresses these outliers through the KOTMS module. It applies dense orthogonal rotations to hidden states before quantization. After rotation, each transformed dimension becomes a linear combination of all original dimensions, so the energy of extreme outliers is redistributed across the full hidden space. This makes activations less skewed and more regular, with a tendency toward a Gaussian-like shape, as in QuaRot[1].
>
> We also considered smoothing methods, which shift variance into the weights. We did not adopt them as the main mechanism because under extreme W1.58 quantization, ternary weights have limited capacity to absorb transferred variance without severe degradation. A hybrid design is promising: mild smoothing before KOTMS may further improve outlier suppression. We will discuss this in Future Work.
>
> ### Q3
> The method relies on heuristic optimization and a fixed bit-width allocation for certain layers, which might not be effective in certain cases.
>
> ### A3
> ILA-AMP is neither heuristic nor fixed-rule. As described in Section 3.4, it is formulated as a constrained integer optimization problem that minimizes overall NLL perturbation under an average bit-width budget. We solve it with dynamic programming, which gives the global optimum under the specified budget rather than heuristic search.
>
> The allocation is also not fixed for any layer. If some layers receive higher precision in practice, that is because the DP solver identifies larger second-order interaction costs from calibration data, not because we manually force layers such as Layer 0 to use more bits. The policy is therefore fully data-driven.
>
> Regarding robustness, the sensitivities are estimated from WikiText2 calibration data, which provides a generic natural-language distribution for PTQ. As shown in Tables 1-3, the resulting allocation generalizes well across diverse zero-shot and commonsense reasoning tasks.
>
> ### Q4
> Although the method is designed to be training-free, could it be integrated with end-to-end training optimization to further improve performance?
>
> ### A4
> Yes. TWLA is highly compatible with end-to-end optimization, including QAT or training from scratch, and we agree this is a promising direction. This view is also consistent with recent results such as BitNet b1.58[2].
>
> Two components are key. First, E2M-ATQ can be inserted into the forward pass of QAT or end-to-end ternary training as the projection onto the asymmetric ternary manifold; combined with STE in the backward pass, it can provide smoother optimization of ternary states. Second, KOTMS can be turned from a post-training compensation module into a trainable structural prior. Under end-to-end optimization, the model could learn to disperse outliers during training and gradually form a more quantization-friendly tri-modal activation structure. We will note this extension in the final version.
>
> We hope these responses clarify the issues and improve the presentation of our work.
>
> [1]Ashkboos et al. (2024). QuaRot: Outlier-Free 4-Bit Inference in Rotated LLMs. ArXiv, abs/2404.00456.
>
> [2]Ma et al. (2024). The Era of 1-bit LLMs: All Large Language Models are in 1.58 Bits. ArXiv, abs/2402.17764.

---

> > ### Author Rebuttal · Reviewer_spM3 · 2026-04-01
> >
> > The author has addressed some of my concerns, but there are still some issues. For example, this method does indeed introduce additional computational overhead. Furthermore, ILA-AMP is a semi-heuristic method because, although it employs dynamic programming, the optimization objective is defined as a heuristic criterion. Therefore, I will maintain my score.

---

> > > ### Author Response · Authors · 2026-04-02
> > >
> > > We thank the reviewer. Two issues remain: (1) the overhead from distribution reshaping, and (2) whether ILA-AMP should be regarded as semi-heuristic. We address both below.
> > >
> > > ### 1. Additional computational overhead
> > >
> > > We agree that the distribution reshaping in TWLA is not zero-overhead. It introduces extra compute and engineering complexity. In ultra-low-bit PTQ, especially W2A4 or lower, this cost should be assessed under the intrinsic constraints of the problem rather than viewed in isolation.
> > >
> > > In this regime, rotation-based reshaping is a mechanism validated by strong PTQ methods. The bottleneck is not only rounding noise, but dynamic-range collapse caused by heavy-tailed distributions and extreme outliers. Without reshaping, a low-bit quantizer must reserve too much range for a few extreme values, sharply reducing effective precision for normal values and causing severe degradation. Representative studies adopt rotations for low-bit quantization [1-2]. Thus, the relevant question is not whether one can avoid any reshaping with strictly zero overhead, but, once reshaping is necessary, which method reaches a better end-to-end accuracy/throughput operating point.
> > >
> > > The fair comparison is thus not an idealized zero-overhead setting, but methods that also rely on rotation-based reshaping.
> > >
> > > On LLaMA-3-8B with matched settings (B=4, In=4096, Out=1024), we compared decode TPS and WikiText2 performance:
> > >
> > > |M|W|A|B|In|Out|PPL|TPS|
> > > |-|-:|-:|-:|-:|-:|-:|-:|
> > > |QuaRot|2|4|4|4096|1024|119.90|350.29|
> > > |FlatQuant|2|4||||53.74|353.17|
> > > |Ours|1.58|4||||**12.83**|**385.20**|
> > >
> > > Although TWLA is not strictly zero-overhead, it achieves a better operating point among comparable rotation-based methods. In decode, our method improves TPS by about 10.0% over QuaRot and 9.1% over FlatQuant. Under W1.58A4, WikiText2 perplexity is 12.83, far better than QuaRot's 119.90 and FlatQuant's 53.74. TWLA therefore does not deny the cost of reshaping; it shows that this cost can be converted into a better accuracy-efficiency trade-off.
> > >
> > > ### 2. Whether ILA-AMP is semi-heuristic
> > >
> > > We agree with the core logic behind the reviewer's comment. The exactness of ILA-AMP lies in the solver. For the defined objective, ILA-AMP is neither a heuristic search procedure nor a fixed-rule allocation strategy. Under a global bit-budget constraint, it solves the chain-structured allocation problem exactly via dynamic programming. With respect to the specified objective, the returned solution is therefore globally optimal.
> > >
> > > The objective itself, however, is not a direct, approximation-free optimization of the full end-to-end quantization loss. Rather, it is a tractable surrogate. Starting from validation NLL, we approximate the otherwise intractable combinatorial bit-allocation problem by a second-order objective composed of unary per-layer costs and adjacent pairwise interaction terms. This surrogate preserves two key factors in W1.58A4: intrinsic layer sensitivity and local coupling between neighboring layers induced by distribution shift and error propagation. Higher-order and longer-range interactions are omitted for tractability.
> > >
> > > In this sense, ILA-AMP may indeed be viewed as semi-heuristic. However, a more precise characterization is that ILA-AMP is a surrogate-based but exactly solved mixed-precision allocation method. Since the full end-to-end combinatorial objective is difficult to optimize directly, many classic PTQ methods rely on tractable proxy objectives. GPTQ uses a Hessian-based proxy, BRECQ optimizes a block reconstruction objective, and AdaRound turns discrete rounding into a locally optimizable continuous surrogate [3-5]. PTQ methods therefore trade direct fidelity for tractability, and ILA-AMP follows the same philosophy.
> > >
> > > Our contribution is that, in W1.58A4, where activation error propagation becomes severe, we move beyond independent layer-sensitivity heuristics by explicitly introducing adjacent-layer second-order interaction terms and unifying them with single-layer NLL perturbation into one objective that can still be solved exactly. Unlike pure heuristic layer scoring, both the per-layer costs and the pairwise interaction terms in ILA-AMP are defined directly through validation NLL perturbations, making the method data-driven rather than rule-based.
> > >
> > > Finally, we thank the reviewer for these important comments. They help clarify TWLA, and we will incorporate the corresponding improvements in the final version.
> > >
> > > [1]Ashkboos et al. QuaRot: Outlier-Free 4-Bit Inference in Rotated LLMs. ArXiv, abs/2404.00456.
> > >
> > > [2]Sun et al. FlatQuant: Flatness Matters for LLM Quantization. ArXiv, abs/2410.09426.
> > >
> > > [3]Frantar et al. GPTQ: Accurate Post-Training Quantization for Generative Pre-trained Transformers. ArXiv, abs/2210.17323.
> > >
> > > [4]Li et al. BRECQ: Pushing the Limit of Post-Training Quantization by Block Reconstruction. ArXiv, abs/2102.05426.
> > >
> > > [5]Nagel et al. Up or Down? Adaptive Rounding for Post-Training Quantization. ArXiv, abs/2004.10568.

---

### Official Review · Reviewer_b4uN · 2026-03-04

**Soundness:** 2
**Presentation:** 3
**Significance:** 3
**Originality:** 3
**Overall Recommendation:** 3
**Confidence:** 5

**Summary:**

The paper proposes a PTQ method called TWLA for ternary weight quantization and low-bit activation quantization. TWLA first applies the Kronecker Orthogonal Transformation to reshape the distribution toward a ternarization-friendly distribution. It then uses E2M-ATQ to obtain a sufficiently good initialization of the quantization parameters. Finally, ILA-AMP performs mixed-precision activation quantization bit-width allocation. The paper further presents empirical results demonstrating that TWLA outperforms existing PTQ methods on the W1.58A16 and W1.58A4 setting across various models.

**Compliance With Llm Reviewing Policy:**

Affirmed.

**Final Justification:**

The authors have addressed some of my concerns. However, the further elaboration of Theorem 3.1 does not appear to describe the optimization of the transformation. Instead, it focuses on optimizing the parameterized symmetric three-component GMM, which seems to reverse the roles of the target distribution and the object to be optimized. Therefore, I maintain my score.

**Key Questions For Authors:**

See weaknesses.

**Limitations:**

yes

**Strengths And Weaknesses:**

**Strengths:**
1. The motivation of the paper aligns well with the characteristics of ternary quantization.
2. ILA-AMP is novel and effectively leverages the insight that error propagation across neighboring layers affects quantization performance.
3. Compared with existing methods, TWLA achieves strong performance and substantially mitigates the degradation typically observed with 4-bit activation quantization.


**Weaknesses:**

1. The proof of Theorem 3.1 provided in Appendix B.3 is questionable. In the proof, a Gaussian distribution $p(w')$ is considered to be well characterized by a symmetric three-component Gaussian mixture because the symmetric three-component Gaussian mixture degenerates into a Gaussian distribution. This reasoning is problematic, since the conclusion of being well characterized is derived from the degeneracy of the mixture distribution. Therefore, the validity of Theorem 3.1 is also questionable.
2. The paper does not report the inference speed differences among W1.58A8, W1.58A6, and W1.58A4. It is also unclear where the practical acceleration arises when activations are quantized with different bit-widths.
3. For the LLaMA 2 and LLaMA 3 model families, prior work [1] shows that the first layer with index starting from 0 requires higher activation precision. The evaluation of ILA-AMP lacks comparison with such empirically established configurations.
4. The quantization time reported for QuaRot in Figure 4(c) appears problematic. The QuaRot algorithm only involves fast Hadamard transformations and row scaling multiplications, and therefore a runtime of 31 minutes on NVIDIA A6000 for LLaMA 2 7B is unlikely.

[1] Sun, Mingjie, et al. "Massive Activations in Large Language Models." ICLR 2024 Workshop on Mathematical and Empirical Understanding of Foundation Models.

---

> ### Author Rebuttal · Authors · 2026-03-27
>
> Thank you for the constructive feedback. Below are our responses.
>
> ### Q1
> The proof of Theorem 3.1 provided in Appendix B.3 is questionable. In the proof, a Gaussian distribution p(w′) is considered to be well characterized by a symmetric three-component Gaussian mixture because the symmetric three-component Gaussian mixture degenerates into a Gaussian distribution. This reasoning is problematic, since the conclusion of being well characterized is derived from the degeneracy of the mixture distribution. Therefore, the validity of Theorem 3.1 is also questionable.
>
> ### A1
> We agree that “well characterized” is imprecise and seems circular. Our argument is based on hypothesis-space inclusion, not shape matching. The symmetric 3-component GMM is a broader family, and the single Gaussian is a strict special case within it. Therefore, once the quantization-error bound is established in the 3-GMM space, it also applies to the Gaussian case by set inclusion. In the revision, we will remove the ambiguous wording and state this relationship explicitly.
>
> KOTMS also motivates this choice physically. It reshapes weights toward a tri-modal distribution that is better for ternarization. As a result, post-KOTMS p(w') departs from a single Gaussian and shows strong 3-GMM characteristics. We used 3-GMM in the theorem to reflect this post-KOTMS distribution. In the revision, we will rewrite Appendix B.3 to clearly separate the mathematical subset argument from the empirical role of KOTMS.
>
> ### Q2
> The paper does not report the inference speed differences among W1.58A8, W1.58A6, and W1.58A4. It is also unclear where the practical acceleration arises when activations are quantized with different bit-widths.
>
> ### A2
> We agree that these differences and their hardware sources should be reported. We tested Qwen3-8B on an NVIDIA A6000 GPU, measuring Prefill TTFT and Decode TPS:
>
> |Model|BS|Len|A8TTFT|A6TTFT|A4TTFT|A8TPS|A6TPS|A4TPS|
> |:--|:--|:--|:--|:--|:--|:--|:--|:--|
> |Qwen3-8B|1|4096|0.387|0.389|0.248|104.58|108.03|115.04|
> |||8192|0.874|0.872|0.600|102.22|103.87|111.42|
> ||4|4096|1.655|1.661|1.115|322.53|328.19|380.59|
> |||8192|3.655|3.647|2.454|289.53|299.44|336.64|
>
> These results show that when quantizing activations to different bit-widths, the practical acceleration stems from two distinct hardware mechanisms: the speedup in the prefill stage is driven by the computational capacity difference of the underlying operators (INT4 vs. INT8 Tensor Cores, A6 is almost identical to A8 because standard GPUs do not provide native 6-bit Tensor Cores) , whereas the speedup in the decode stage is driven by the massive reduction in memory I/O overhead due to the halved KV Cache footprint. We will add this table and discussion in the appendix.
>
> ### Q3
> For the LLaMA 2 and LLaMA 3 model families, prior work [1] shows that the first layer with index starting from 0 requires higher activation precision. The evaluation of ILA-AMP lacks comparison with such empirically established configurations.
>
> ### A3
> We agree that preserving certain layers at higher precision is a useful heuristic in conventional quantization. However, TWLA operates in a different space. Those heuristics are derived from the original activation distribution, whereas TWLA applies KOTMS before quantization. KOTMS uses dense orthogonal rotations to redistribute extreme activation energy across hidden dimensions, mitigating the localized “massive activation” phenomenon that motivates fixed high-precision treatment for layers like Layer 0.
>
> Because the activation landscape is reshaped after KOTMS, static heuristics from the unrotated model are not necessarily optimal here. Instead, ILA-AMP allocates bit-widths dynamically according to post-KOTMS NLL perturbations and second-order interaction costs, so the allocation follows the transformed sensitivity pattern rather than a fixed prior rule. In the camera-ready version, we will discuss this distinction and explain how KOTMS changes the activation distribution underlying such heuristics.
>
> ### Q4
> The quantization time reported for QuaRot in Figure 4(c) appears problematic. The QuaRot algorithm only involves fast Hadamard transformations and row scaling multiplications, and therefore a runtime of 31 minutes on NVIDIA A6000 for LLaMA 2 7B is unlikely.
>
> ### A4
> We agree completely. The reported 31 minutes is not the standalone runtime of QuaRot. It is the end-to-end pipeline time in our evaluation setting, where QuaRot is used as a preprocessing step and followed by GPTQ for low-bit weight quantization. The dominant cost comes from the subsequent GPTQ stage while the pure QuaRot step takes only a small fraction of the total time. We apologize for the unclear label in Figure 4(c). In the camera-ready version, we will revise it to “QuaRot + GPTQ (End-to-End)” and add a breakdown in the text.
>
> We hope these responses clarify the issues and improve the presentation of our work. Please let us know if further discussion or details are needed.

---

> > ### Author Rebuttal · Reviewer_b4uN · 2026-04-02
> >
> > Thank you for your response. Your answers address some of my concerns; however, several issues remain. Therefore, I maintain my score.
> >
> > 1. Although KOTMS smooths the activation distribution to some extent, the massive activation phenomenon still persists. A comparison with massive activation phenomenon induced heuristics for bit allocation is still necessary.
> >
> > 2. Although the empirical results demonstrate the effectiveness of KOTMS, the theoretical justification provided in Theorem 3.1 is not sufficiently convincing. The issue is not limited to the problematic use of the phrase “well characterized.” More importantly, the proof attempts to explain how the transformation leads to a symmetric three-component GMM by making the symmetric three-component GMM, as the target distribution, approximate the single Gaussian $p(w')$, which is the distribution to be optimized, only to a suboptimal extent. This line of reasoning, in which the target is used to approximate the distribution under optimization, is fundamentally problematic.

---

> > > ### Author Response · Authors · 2026-04-02
> > >
> > > Thank you for the careful follow-up and for raising these two concerns explicitly. We respond to each point below.
> > >
> > > ### 1. Comparison with massive-activation-inspired bit-allocation heuristics
> > >
> > > We strongly agree that such a comparison is necessary. This is fully aligned with our own intention: systematically comparing different layer-sensitivity metrics is essential for demonstrating the value of ILA-AMP over static or first-order allocation rules.
> > >
> > > Indeed, we have already included this comparison in **Appendix C.3 (Lines 1588–1638)**. Under an average activation-bit budget of 4 bits, we evaluate several representative layer-sensitivity metrics, including **activation-based scoring**, which follows the same intuition as massive-activation heuristics by assigning higher importance to layers with larger activation energy. We also compare against the single-layer NLL-increase metric, which can be viewed as the strongest first-order layer-wise sensitivity baseline.
> > >
> > > As shown in **Table 8**, ILA-AMP consistently achieves the lowest perplexity across all evaluated models, namely 8.31 / 12.83 / 16.15 on LLaMA2-7B / LLaMA3-8B / Qwen3-8B, substantially outperforming all heuristic metrics. Compared with the strongest single-metric baseline based on single-layer NLL increase, ILA-AMP further reduces PPL by 30.3% on LLaMA2-7B (11.92 → 8.31),  and 25.9% on Qwen3-8B (21.79 → 16.15).
> > >
> > > We believe these results directly support the reviewer’s point. Although KOTMS smooths activations, the residual quantization difficulty remains heterogeneous across layers, so static heuristics based only on activation magnitude or pre-defined protected layers are still insufficient. This is precisely why ILA-AMP is needed. It adapts to the post-KOTMS sensitivity landscape through validation-NLL perturbations and adjacent-layer interaction costs. In other words, after KOTMS reshapes the activation landscape, dynamic allocation based on post-transformation sensitivity is consistently more effective than static layer-wise heuristics.
> > >
> > > To avoid ambiguity, we will make this connection more explicit in the final version and move the Appendix C.3 comparison into the main discussion around ILA-AMP.
> > >
> > > ### 2. Theoretical justification of Theorem 3.1 and the role of KOTMS
> > >
> > > We also thank the reviewer for this sharp observation. We agree that the current Appendix B.3 does not fully justify the stronger causal statement that the orthogonal transformation itself induces a symmetric three-component GMM. Under the isotropic Gaussian assumption, the orthogonal transform preserves the population Gaussian form, and the KL-based argument only shows that the symmetric 3-GMM family is expressive enough to approximate the resulting marginal distribution. As the reviewer correctly points out, this is an approximation-family argument, not a mechanism proof that the transformation alone guarantees tri-modalization. We will therefore revise this part in the final version, remove over-strong wording such as “well characterized,” and weaken Theorem 3.1 to a mathematically correct motivation statement.
> > >
> > > That said, the effectiveness of KOTMS does not rely on Theorem 3.1 alone. Its core justification comes from the optimization objective itself. KOTMS does not assume that any orthogonal transform will automatically produce a tri-modal distribution. Rather, it explicitly learns **a Kronecker-structured orthogonal transform under a TriGMM-based shaping objective**, so that the rotated weights are driven toward three codebook-aligned regions and become more ternary-friendly. Thus, the orthogonal transform provides an invertible and deployment-consistent search space, while the shaping objective drives the weights toward a codebook-aligned structure.
> > >
> > > More importantly, the resulting post-KOTMS effect is supported empirically. As shown in **Fig. 2 and Appendix Fig. 8**, before KOTMS the weight distributions are Gaussian-like and unimodal. After KOTMS, they become a more structured multi-cluster pattern, the tails are tightened, and the corresponding activation distributions also become more concentrated with fewer extreme values. These observations provide direct evidence that KOTMS improves distribution alignment for ternary quantization and suppresses activation-side outlier interference.
> > >
> > > Therefore, in the revised manuscript, we will present the role of KOTMS more precisely as a codebook-aligned shaping mechanism validated by optimization and empirical evidence, rather than as a phenomenon guaranteed by the current theorem.
> > >
> > > Finally, we thank the reviewer again for these constructive follow-up questions. They help us sharpen both the positioning of ILA-AMP relative to heuristic baselines and the theoretical role of KOTMS. In the final version, we will incorporate these clarifications and the related appendix analysis to further improve the completeness and persuasiveness of the paper. This should also make the paper easier to interpret and assess in practice.

---

### Official Review · Reviewer_NkmM · 2026-03-10

**Soundness:** 4
**Presentation:** 3
**Significance:** 3
**Originality:** 3
**Overall Recommendation:** 5
**Confidence:** 3

**Summary:**

This paper proposes a novel post-training quantization (PTQ) framework called TWLA, which enables ternary weight quantization and 4-bit activation quantization of LLMs without significant accuracy loss.

To enable ternary weight quantization, the framework introduces Euclidean-to-Manifold Asymmetric Ternary Quantizer (E2M-ATQ) and Kronecker Orthogonal Tri-Modal Shaping.
First, E2M-ATQ minimizes quantization error through a two-stage optimization process. In the first stage (called Euclidean warm-start), it minimizes the L2 distance between the original weights and ternarized weights by iteratively optimizing ternary weight states, as well as the shift and scaling factors used for ternarization. In the second stage (called Manifold Relocation), the shift and scaling factors are further optimized by deriving a closed-form solution that minimizes the layer output error.
Second, Kronecker Orthogonal Tri-Modal Shaping introduces orthogonal matrix-based weight rotation to transform weight matrices into a tri-modal distribution that is more suitable for ternary quantization and thus results in lower ternarization error.

Finally, to reduce the error caused by activation quantization, the framework adopts layer-wise adaptive activation precision, where different activation precisions are assigned to each layer according to its sensitivity to activation quantization.

**Compliance With Llm Reviewing Policy:**

Affirmed.

**Final Justification:**

This paper is highly remarkable in that it demonstrates W1.58A4 LLM quantization using a post-training quantization (PTQ) approach. As a PTQ method, it can be readily applied to a wide range of LLMs. Moreover, while the authors primarily focus on GPU-based performance analysis, I believe that, given its ternary weight representation, the proposed method could offer even greater benefits when combined with dedicated hardware that supports ternary operations. Therefore, I maintain my score.

**Key Questions For Authors:**

Could the authors evaluate latency and throughput separately for the prefill and decode stages? Additionally, could the authors report latency/throughput across various batch sizes, input lengths, and output lengths?

**Limitations:**

yes

**Strengths And Weaknesses:**

**Strengths**
1. The paper provides a detailed derivation of the equations used in the proposed method.
2. The proposed method achieves the best accuracy/perplexity compared to existing state-of-the-art 2-bit or ternary LLMs.
3. The proposed method is training-free, making it easily applicable to a wide range of LLMs.
4. The paper provides extensive evaluation in terms of accuracy. The method is evaluated on multiple models and compared against various prior works. In addition, ablation studies are provided to analyze the contribution of the three main components of TWLA.

**Weakness**
1. The speedup evaluation is conducted only with batch size 4, and it does not provide a breakdown of prefill and decode latency.

---

> ### Author Rebuttal · Authors · 2026-03-27
>
> We sincerely appreciate your constructive feedback. Below, we provide point-by-point responses, with all revisions incorporated accordingly.
>
> ### Q1
> Could the authors evaluate latency and throughput separately for the prefill and decode stages? Additionally, could the authors report latency/throughput across various batch sizes, input lengths, and output lengths?
>
> ### A1
> We sincerely thank the reviewer for this highly professional and constructive suggestion. We completely agree that separating the prefill and decode stages, and evaluating across various system dimensions, is the gold standard for LLM systems. A coarse-grained speedup metric indeed obscures the distinct hardware bottlenecks present in these two phases.
>
> Specifically, we conducted an extensive, fine-grained benchmark on the LLaMA-3-8B and Qwen3-8B models under identical hardware settings (NVIDIA A6000). The evaluation configurations cover varying batch sizes ($B \in \{1, 4\}$), long-context input lengths ($L_{in} \in \{2K, 4K\}$ for LLaMA-3; $L_{in} \in \{4K, 8K\}$ for Qwen3), and different output lengths ($L_{out} \in \{512, 1024\}$). We measure the Prefill stage using Time-To-First-Token (TTFT, in seconds; lower is better) and the Decode stage using Tokens Per Second (TPS; higher is better).
>
> | Model | Batch-size | Input Lengths | Output Lengths | Prefill TTFT (s) - FP16 | Prefill TTFT (s) - Ours | Prefill Speed up | Decode TPS - FP16 | Decode TPS - Ours | Decode Speed up |
> | :--- | :--- | :--- | :--- | :--- | :--- | :--- | :--- | :--- | :--- |
> | LLaMA-3-8B | 1 | 2048 | 512 | 0.290 | 0.121 | 2.42 | 53.12 | 116.86 | 2.20 |
> | | | | 1024 | 0.312 | 0.132 | 2.40 | 52.12 | 114.14 | 2.19 |
> | | | 4096 | 512 | 0.667 | 0.277 | 2.41 | 52.71 | 113.85 | 2.16 |
> | | | | 1024 | 0.666 | 0.280 | 2.38 | 50.72 | 109.05 | 2.15 |
> | | 4 | 2048 | 512 | 1.274 | 0.542 | 2.35 | 197.59 | 414.94 | 2.10 |
> | | | | 1024 | 1.276 | 0.549 | 2.32 | 196.81 | 411.33 | 2.09 |
> | | | 4096 | 512 | 2.614 | 1.108 | 2.36 | 188.70 | 388.72 | 2.06 |
> | | | | 1024 | 2.634 | 1.140 | 2.31 | 187.90 | 385.20 | 2.05 |
> | Qwen3-8B | 1 | 4096 | 512 | 0.581 | 0.248 | 2.34 | 51.13 | 115.04 | 2.25 |
> | | | | 1024 | 0.615 | 0.265 | 2.32 | 50.45 | 112.01 | 2.22 |
> | | | 8192 | 512 | 1.399 | 0.600 | 2.33 | 49.74 | 111.42 | 2.24 |
> | | | | 1024 | 1.390 | 0.607 | 2.29 | 47.58 | 103.72 | 2.18 |
> | | 4 | 4096 | 512 | 2.565 | 1.115 | 2.30 | 176.20 | 380.59 | 2.16 |
> | | | | 1024 | 2.549 | 1.118 | 2.28 | 170.33 | 359.40 | 2.11 |
> | | | 8192 | 512 | 5.571 | 2.454 | 2.27 | 157.31 | 336.64 | 2.14 |
> | | | | 1024 | 5.508 | 2.437 | 2.26 | 156.63 | 325.79 | 2.08 |
>
> From the exhaustive experimental data above, a core trend clearly emerges: whether in the long-context prefill stage or the high-concurrency decode stage, TWLA consistently achieves a significant speedup of over 2.0x on average.
>
> Our fine-grained analysis precisely reveals why TWLA's W1.58A4 paradigm is so powerful from a physical hardware perspective:
>
> * **Decode Stage (Memory-Bandwidth Bound):** In autoregressive generation (matrix-vector multiplication), loading weights from HBM to SRAM is the primary bottleneck. By aggressively compressing the weights to a hardware-agnostic $\sim 1.58$ bits, TWLA drastically alleviates the "memory wall." As our new results demonstrate, this leads to substantial reductions in decode latency and massive improvements in decode throughput.
> * **Prefill Stage (Compute Bound):** In the prompt processing phase (matrix-matrix multiplication), massive activation tensors dominate the computational overhead. Here, TWLA’s low-bit activation quantization (A4) allows the system to utilize highly efficient 4-bit Tensor Core operators. Combined with ternary weights, this not only multiplies the peak computational throughput but also significantly reduces memory data transfer. The results indicate that TWLA maintains highly scalable throughput during the prefill stage, effectively preventing Out-Of-Memory (OOM) errors even under large batch sizes and long input contexts.
>
> We will dedicate a new subsection (along with the comprehensive table) in the Appendix of the camera-ready version to report these full latency/throughput evaluations, ensuring the community has a complete systems-level understanding of TWLA's performance under various real-world deployment constraints.
>
> We sincerely hope that our responses have adequately resolved your concerns and improved the clarity and robustness of our work. Please do not hesitate to let us know if you require further discussion, additional experiments, or any clarifications—we are more than happy to provide any further details needed to address your questions comprehensively.

---

> > ### Author Rebuttal · Reviewer_NkmM · 2026-04-01
> >
> > Thank you for the detailed responses. My concerns have been well addressed, and I will maintain my original score.

---

> > > ### Author Response · Authors · 2026-04-02
> > >
> > > We sincerely thank you for your time and for acknowledging our rebuttal. We are very glad to hear that our detailed responses have fully addressed your concerns. Your rigorous and constructive feedback has been invaluable in helping us improve the clarity and overall quality of our manuscript.
> > >
> > > Thank you once again for your positive evaluation and your support of our work!

---

### Official Review · Reviewer_yP2G · 2026-03-12

**Soundness:** 3
**Presentation:** 3
**Significance:** 3
**Originality:** 4
**Overall Recommendation:** 4
**Confidence:** 4

**Summary:**

Background: Existing ternarization quantization methods for LLM struggle with heavy-tailed activation distributions and therefore keep activations in high precision.
Method: They proposed Ternarized Weights and Low-bit Activations, a post-training quantization (PTQ) framework that achieves 1.58-bit weight compression and 4-bit activation quantization while maintaining high accuracy. (1) Euclidean-to-Manifold Asymmetric Ternary Quantizer (E2M-ATQ) ; (2) Kronecker Orthogonal Tri-Modal Shaping (KOTMS) ; (3) Inter-Layer Aware Activation Mixed Precision (ILA-AMP).

**Compliance With Llm Reviewing Policy:**

Affirmed.

**Final Justification:**

The main drawback of this work lies in the limitation of application caused by the ternary support. I maintain my score.

**Key Questions For Authors:**

1. The explicit guidance on parameter selection (e.g., how to set n₁/n₂ for different model architectures or hidden sizes) could be discussed.
2. The efficiency analysis focuses on NVIDIA A6000 GPUs and llama.cpp. The constraints of real-world deployment on different hardware could be discussed. Does the actual storage precision of ternarization parameters vary from one hardware to another, resulting in different speedups?
3. How ILA-AMP’s bit allocation adapts to different layer sensitivities is not clear enough. A visualization like the empirical evidence (e.g., layer-wise activation variance before/after KOTMS) would be better.

**Strengths And Weaknesses:**

Strengths
1. This is the first work on ternarization WA quantization for LLM, and the methods proposed are reasonable and original.
2. Figure 2 clearly shows the motivation of the authors and the effects of their approach in a style that seems to be a reference to the SmoothQuant team and resonates with previous work.
3. The method is reasonable and learnable equivalence transformation is proposed
that reshapes unimodal weights into a tri-modal structure is simple and easy to understand, and the idea is in the same vein as Quarot's research.
4. The appendix gives detailed proofs, ablation experiments, and the main code provided by the authors, which is appreciated.

Weaknesses
1. Fig.3 combines the method with the effect diagram and the text description, which makes the layout confusing. Improving the drawing scheme can make it clear and easy to understand.
2. Although the method is described in detail, the author needs to supplement the description of the engineering implementation section, see Questions.

---

> ### Author Rebuttal · Authors · 2026-03-27
>
> Thank you for the constructive feedback. Below are our point-by-point responses.
>
> ### Q1
> Fig.3 combines the method with the effect diagram and the text description, which makes the layout confusing. Improving the drawing scheme can make it clear and easy to understand.
>
> ### A1
> We agree that Fig. 3 is overly crowded because it mixes the method diagram, textual explanations, and effect visualizations, which weakens the visual guidance. In the camera-ready version, we will simplify it by keeping Fig. 3 only as the core TWLA pipeline, moving dense explanations into the main text, and relocating the effect visualizations to a separate figure or the appendix. We believe this will substantially improve clarity and readability.
>
> ### Q2
> The explicit guidance on parameter selection (e.g., how to set n₁/n₂ for different model architectures or hidden sizes) could be discussed.
>
> ### A2
> We agree that explicit guidance on parameter selection is important. TWLA does not require manual tuning of $n_1$ and $n_2$ across different model architectures or hidden sizes. As shown in Section 4.3, more balanced Kronecker dimensions (a smaller $n_1/n_2$ ratio) sharply reduce effective weight-bit overhead with only minor accuracy loss, so our design is to set $n_1$ and $n_2$ as close as possible. We implement this with a deterministic, parameter-free factorization rule in **Appendix A, Algorithm 1**: for hidden dimension $d$, we start from $\lfloor\sqrt{d}\rfloor$ and search downward for a divisor such that $d_1 \times d_2 = d$ and $d_1 \approx d_2$. We also agree that placing this only in the appendix may make it easy to miss. In the camera-ready version, we will move this guidance and a brief description of the automatic factorization rule into Section 3.3 for clarity and reproducibility.
>
> ### Q3
> The efficiency analysis focuses on NVIDIA A6000 GPUs and llama.cpp. The constraints of real-world deployment on different hardware could be discussed. Does the actual storage precision of ternarization parameters vary from one hardware to another, resulting in different speedups?
>
> ### A3
> We clarify that the actual storage precision remains constant across hardware platforms. TWLA uses a universal, hardware-agnostic bit-packing scheme that packs five ternary values into one 8-bit integer, yielding a stable effective storage precision of about 1.6 bits/weight on any device.
>
> However, the resulting inference speedup is hardware-dependent because different platforms have different memory bandwidth and compute capabilities, which changes the balance between memory-bound and compute-bound execution. In our A6000 + `llama.cpp` experiments, the reported speedup includes the software unpacking overhead before matrix multiplication. Although `llama.cpp` uses optimized low-bit kernels, standard GPUs still lack native sub-byte arithmetic support, so this overhead limits acceleration. On specialized hardware with native ternary support, such as NPUs **[1]** or FPGAs, the unpacking overhead could be removed and the speedup would likely be higher than our A6000 baselines. We will add this distinction between hardware-agnostic memory savings and hardware-dependent speedups in the camera-ready version so that the deployment implications are clearer.
>
> ### Q4
> How ILA-AMP’s bit allocation adapts to different layer sensitivities is not clear enough. A visualization like the empirical evidence (e.g., layer-wise activation variance before/after KOTMS) would be better.
>
> ### A4
> We agree that the link between layer sensitivity and final bit allocation should be explained more clearly and visualized in the main text. ILA-AMP models layer sensitivity through its impact on the final output distribution, as defined in Eqs. 16-19. We use NLL perturbation as the optimization objective because our goal is to minimize generation-quality degradation, and $\text{PPL}=\exp(\text{NLL})$, so reducing NLL is monotonically equivalent to reducing PPL degradation. The surrogate in Eqs. 16-19 captures both first-order single-layer sensitivity and second-order interaction/error propagation between adjacent layers, producing a data-driven sensitivity score for each layer under different bit-widths. The DP solver then allocates higher precision to layers with larger first- and second-order NLL impacts under a global budget. In the camera-ready version, we will add a new figure **(as shown [here](https://anonymous.4open.science/r/TWLA-0212/rebuttal_fig1.jpg))** to more clearly illustrate how the allocation follows the modeled sensitivities and make the mechanism more explicit. This figure will clearly and directly link modeled sensitivities to final allocations.
>
> We hope that our responses have adequately resolved your concerns. Please let us know if further discussion or details are needed.
>
> [1] Park, Dahoon et al. "A Survey on Binary and Ternary Neural Networks and Their Realization in Compute-in-Memory for Edge Intelligence." *IEEE Internet of Things Journal* 13 (2026): 3433-3458.

---

> > ### Author Rebuttal · Reviewer_yP2G · 2026-04-03
> >
> > Thank you for the detailed explanation. Regarding question 3, the storage accuracy and storage space have been clarified. However, there are still doubts regarding the acceleration part. For instance, for the case where ternary support is not supported, can acceleration be achieved through operators? This method is of crucial importance for its wide application.

---

> > > ### Author Response · Authors · 2026-04-04
> > >
> > > We thank the reviewer for raising this question. We agree that this issue matters for the deployment potential of TWLA. If the acceleration of W1.58 quantization were achievable only on hardware with native ternary support, then its value would be limited.
> > >
> > > Our answer is yes: **even on commodity GPUs without native ternary hardware support, TWLA can still achieve end-to-end acceleration through specialized software operators.** The key idea is to decouple storage from compute, so that we retain the storage efficiency of ternary weights while mapping the actual computation onto low-bit operator pipelines already available on standard hardware.
> > >
> > > More specifically, our weights are stored in a compact 5-ternary-in-1-byte format. This corresponds to about 1.6 bits per weight, very close to the information-theoretic limit of ternary representation, and more storage-efficient than a naive one-weight-per-2-bit container. During autoregressive decoding, where performance is constrained by HBM bandwidth, this compact representation reduces weight traffic and directly alleviates the dominant memory bottleneck.
> > >
> > > At compute time, although standard GPUs such as the NVIDIA A6000 do not provide native ternary Tensor Cores, they do provide **INT4 Tensor Core** pipelines. Since ternary values are a strict subset of the INT4 value range, we implement a fused low-bit operator that loads packed ternary codes from memory, decodes them on the fly into INT4 fragments inside fast on-chip SRAM/registers, and invokes native INT4 MMA instructions for matrix multiplication. In other words, ternary quantization is used on the storage side, while arithmetic execution is bridged to the standard INT4 compute path.
> > >
> > > This design is closely related in spirit to recent **BitNet a4.8[1] / BitBLAS[2]-style implementations**: the storage side is kept compact, while the compute side is mapped to a hardware-friendly low-bit format supported by commodity accelerators. In our case, we start from an even denser ternary representation, namely five ternary values packed into one byte, and decode this compact code into INT4 fragments to preserve both the storage advantage of W1.58 and the execution efficiency of INT4 operators.
> > >
> > > ### Core implementation
> > >
> > > ```python
> > > # packed ternary codes in registers
> > > B_local_packed=T.alloc_local((warp_cols*local_size_b),"int8")
> > > # decoded INT4 fragments for compute
> > > B_local_int4=T.alloc_local((warp_cols*local_size_b),in_dtype)
> > > # native INT4 Tensor Core emitter
> > > mma_emitter=INT4TensorCoreIntrinEmitter(
> > >     a_dtype="int4",b_dtype="int4",accum_dtype=accum_dtype,...
> > > )
> > > for ko in T.Pipelined(...):
> > >     for ki in T.serial(...):
> > >         # load activation fragment
> > >         mma_emitter.ldmatrix_a(A_local,A_shared,ki)
> > >         # load packed ternary weights
> > >         load_packed_ternary(B_local_packed,B_shared,ki)
> > >         # decode ternary -> INT4 fragments
> > >         B_local_int4=decode_ternary5_to_int4_fragments(B_local_packed)
> > >         # run native INT4 MMA
> > >         mma_emitter.mma(A_local,B_local_int4,C_local)
> > > ```
> > > On the memory side, the model benefits from the compact ternary storage format, especially in the decode stage. On the compute side, matrix multiplication is executed by native INT4 Tensor Core operators. Therefore, even without native ternary hardware support, the model can exploit most of the bandwidth advantage of W1.58 quantization while reusing standard low-bit compute infrastructure.
> > >
> > > We have implemented this software-operator path and verified it in end-to-end inference on a standard NVIDIA A6000 GPU without native ternary support. On **LLaMA-3-8B** with batch size=4, input length=4096, and output length=1024, the measured performance is:
> > >
> > > - **Prefill (TTFT):** FP16 = 2.634s, Ours = 1.140s, **2.31x speedup**
> > > - **Decode (TPS):** FP16 = 187.90, Ours = 385.20, **2.05x speedup**
> > >
> > > These results directly demonstrate that the acceleration of TWLA is not restricted to specialized ternary hardware. By combining **5-ternary-per-byte compact storage** with **on-the-fly decode to INT4 compute operators**, we break the memory wall in decoding while leveraging the INT4 compute pipeline on commodity GPUs. In this sense, TWLA should not be viewed as a hardware-specific theoretical proposal, but as a practical deployment framework already mappable onto today’s mainstream accelerators.
> > >
> > > To make this point fully explicit, we will add a dedicated appendix discussion in the final version to distinguish two deployment paradigms:
> > >
> > > 1. **Native ternary operators** on future hardware, with maximal acceleration potential.
> > > 2. **Packed-ternary storage plus INT4 software operators** on current commodity GPUs, with a practical and effective acceleration path.
> > >
> > > We thank the reviewer again for highlighting this issue, which helps improve the clarity and practical value of our paper. We hope this resolves your concerns.
> > >
> > > [1]Wang et al. *BitNet a4.8: 4-bit Activations for 1-bit LLMs.* ArXiv, abs/2411.04965.
> > >
> > > [2]https://github.com/microsoft/BitBLAS

---

### Decision · Program_Chairs · 2026-04-30

**Decision:**

Accept (regular)

**Comment:**

The overall discussion leans toward acceptance. The main concerns have been sufficiently addressed during the review process, and the remaining issues do not appear substantial enough to outweigh the paper’s strengths. On balance, the submission is considered suitable for acceptance.